# Spectrin coordinates cell shape and signaling essential for epidermal differentiation

Arad Soffer[1]* , Aishwarya Bhosale[2,3]* , Roohallah Ghodrat[2,3] , Marc Peskoller[2,3,4] , Takeshi Matsui[5] , Carien M. Niessen[2,3,4]** , Chen Luxenburg[1]** , and Matthias Rübsam[2,3,4]**

**Cell shape and fate are tightly linked, yet how the cortical cytoskeleton integrates regulation of shape and fate remains unclear. Using the multilayered epidermis as a paradigm for cell shape–guided changes in differentiation, we identify spectrin as an essential organizer of the actomyosin cortex to integrate transitions in cell shape with spatial organization of signaling. Loss of αII-spectrin (*Sptan1*) in mouse epidermis altered cell shape in all layers and impaired differentiation and barrier formation. High-resolution imaging and laser ablation revealed that E-cadherin organizes gradients of cortical actin and spectrin into layer-specific submembranous networks with discrete structural and mechanical properties that coordinate cell shape and fate. This layer-specific organization dissipates tension and, in upper layers, retains activated growth factor receptor EGFR and the calcium channel TRPV3 at the membrane to induce terminal differentiation. Together, these findings reveal how polarized organization of the cortical cytoskeleton directs transitions in cell shape and cell fate at the tissue scale necessary to establish epithelial barriers.**

## Introduction

Cells in our body exhibit a wide range of shapes, sizes, and functions, from giant, multinucleated osteoclasts and contractile skeletal muscle cells to small, disc-shaped red blood cells (Luxenburg et al., 2007; Calderón et al., 2014; Elgsaeter et al., 1986). Defects in cell shape often correlate with tissue malfunction and disease phenotypes (Lee et al., 2016; Kalluri and Weinberg, 2009; Delaporte et al., 1990; Emery, 2002; PAULING and ITANO, 1949). For example, in common skin diseases such as psoriasis, atopic dermatitis, or squamous cell carcinoma skin, keratinocytes exhibit abnormal cell shape associated with impaired differentiation and compromised epidermal barrier function (Koegel et al., 2009; Van Der Kammen et al., 2017). However, whether cell shape directly regulates cell differentiation and tissue function remains an open question.

At the body's surface, the interfollicular epidermis (hereafter, epidermis) is a stratified, multilayered epithelium that provides a barrier to protect organisms from water loss and external challenges. The epidermis exhibits progressive cell shape changes that are linked to the differentiation status of keratinocytes. Basal cuboidal stem cells, when initiating differentiation, move upward into the suprabasal spinous layer (SS) to begin flattening and increase in size. Upon entering the first of the three granular layers, SG3, keratinocyte flattening is substantially increased to take on the shape of Kelvin's flattened tetrakaidecahedrons in SG2, which is essential to maintain the tight junctional (TJ) barrier in this layer (Kubo et al., 2009; Yoshida et al., 2013; Yokouchi et al., 2016; Rübsam et al., 2017). While passing through the SG1 cells take on their final squamous shape that coincides with terminal differentiation. This process involves enucleation and transglutaminase (TGM)-mediated cross-linking of structural proteins and barrier lipids necessary to form a functional stratum corneum barrier that physically separates the organism from the environment (Simpson et al., 2011; Fuchs, 2007). The mammalian epidermis is thus an ideal model to address the relationship between cell position, shape, and differentiation.

A key determinant of cell shape is the actomyosin cytoskeleton and its associated adhesion receptors. The actomyosin cytoskeleton forms a contractile network made up of actin fibers (F-actin), myosin II motor proteins, and dozens of actin-binding proteins that regulate its organization and dynamics to control the contractile mechanical state of cells (Pollard, 2016). By engaging integrin-based cell–matrix adhesions and cadherin-based adherens junctions (AJs), actomyosin-generated forces

[1]Gray School of Medical Sciences, Gray Faculty of Medical and Health Sciences, Tel Aviv University, Tel Aviv, Israel;   [2]Department Cell Biology of the Skin, University Hospital Cologne, University of Cologne, Cologne, Germany;   [3]Cologne Excellence Cluster for Stress Responses in Ageing-associated Diseases (CECAD), University Hospital Cologne, University of Cologne, Cologne, Germany;   [4]Center for Molecular Medicine Cologne (CMMC), University Hospital Cologne, University of Cologne, Cologne, Germany;   [5]School of Bioscience and Biotechnology, Tokyo University of Technology, Tokyo, Japan.

*A. Soffer and A. Bhosale contributed equally to this paper;   **C.M. Niessen, C. Luxenburg, and M. Rübsam shared corresponding authorship.   Chen Luxenburg: lux@tauex.tau.ac.il;   Carien M. Niessen: carien.niessen@uni-koeln.de.

are transmitted across cells and tissues (Noordstra et al., 2023; Saraswathibhatla et al., 2023). Moreover, actomyosin activity also modifies signaling cascades involved in fate regulation (Luxenburg and Zaidel-Bar, 2019; Pollard, 2016; Zaidel-Bar et al., 2015). In the epidermis, an increase in cortical F-actin levels accompanies differentiation and the gradual formation of squamous shapes, a process regulated by E-cadherin (Rübsam et al., 2017). Yet, how cortical actomyosin networks are spatially patterned across layers and how these changes are linked to differentiation remain poorly understood.

Spectrins are tetramers consisting of two α- and two β-spectrins. Vertebrates have two α-spectrin (αI and αII) and five β-spectrin (βI–βV) variants encoded by different genes. While αI-spectrin is expressed only in erythrocytes, αII-spectrin (encoded by *Sptan1*) is the only α-spectrin in non-erythrocyte cells that can interact with each of the five β-spectrins (Bennett and Healy, 2009; Bennett and Lorenzo, 2016; Teliska and Rasband, 2021) to form antiparallel tetramers that integrate into the cortical actomyosin network in diverse cell types, including erythrocytes, neurons, and fibroblasts (Leterrier and Pullarkat, 2022). These tetramers display tensile, spring-like behavior that in combination with a dynamic interplay with myosin enable spectrin–actin networks to absorb shocks, sense and generate forces, and regulate their elastic properties that together control cell shape (Leterrier and Pullarkat, 2022; Lorenzo et al., 2023; Ghisleni et al., 2020). Moreover, the organization of spectrin-actomyosin networks promotes microdomain formation of transmembrane proteins, e.g., ion channels, in the plasma membrane (Lorenzo et al., 2023).

The function of spectrin in epithelia remains less well defined. As in erythrocytes, spectrin controls cortical actomyosin organization and cell shape in *Drosophila* follicular epithelium and in *Caenorhabditis elegans* epidermis (Praitis et al., 2005; Ng et al., 2016). In mice, *Sptan1* knockout is embryonic lethal, characterized by neural tube, cardiac, and craniofacial defects as well as abnormal growth (Stankewich et al., 2011). In cultured keratinocytes, spectrin was implicated in regulating early keratinocyte differentiation (Zhao et al., 2011, 2013; Wu et al., 2015), but whether spectrins control in vivo epithelial cell shapes and regulate epithelial differentiation and barrier formation is not known.

In the present study, we identify spectrin as an essential component of the epidermal actomyosin cytoskeleton. We show that distinct actin and spectrin gradients drive changes in the conformation, mechanics, and signaling properties of this submembranous network to guide transitions in cell shape and fate when cells move into a new layer. E-cadherin orchestrates this layer-specific organization of the spectrin-actomyosin cortex to control tension states and membrane nanoscale organization, resulting in the spatial activation of EGF receptor (EGFR) and the calcium channel TRPV3 in the uppermost epidermal layers to promote terminal differentiation and thus epidermal barrier formation. These observations provide novel insights into how dynamic changes in cortical organization integrate mechanics and signaling to coordinate cell fate and cell shape in a physiologically relevant mammalian system.

## Results

### αII-spectrin determines epidermal cell shape

Although it is well established that keratinocytes adopt an increasingly squamous morphology during stratification, their exact in vivo 3D shapes and volumes have not been defined. To this end, we used a K14-Cre–driven rainbow transgene (Snippert et al., 2010) to fluorescently label single cells in vivo, followed by 3D rendering of the cytoplasmic YFP signal to determine cell shape, size, and volume (Fig. 1, a and b; and Fig. S1 a). This analysis showed that initially cuboidal basal keratinocytes double in both size and volume twice, first when transiting into the spinous layer and again when moving into the granular layer during stratification. (Fig. 1, a–c and Fig. S1 b). Only upon transitioning into fully squamous corneocytes, the size-to-volume ratio is increased due to a further expansion of surface area (1.41-fold). Thus, keratinocytes undergo highly robust and highly stereotypic changes in cell size, shape, and volume.

We next asked whether epidermal cell shape depends on intercellular contacts, as AJs are key determinants of cell shape (Luxenburg and Zaidel-Bar, 2019; Niessen et al., 2024). To this end, we first separated the SS and the SG3 layer from the late-differentiated (SG2/SG1) layers (Matsui et al., 2021) to then isolate individual cells. While SS/SG3 cells rounded up, SG2/SG1 cells maintained their characteristic flattened tetrakaidecahedron shape also upon loss of cell–cell contacts (Fig. 1, d and e; and Fig. S1 a) (Yokouchi et al., 2016).

The ability of single SG1/SG2 cells to maintain their in vivo shape is reminiscent of the characteristic red blood cell shape for which the spectrin cortical cytoskeleton is essential (Leterrier and Pullarkat, 2022). Immunostaining for non-erythrocyte αII-spectrin and F-actin in epidermal whole mounts and sections from newborn mice revealed that spectrin partially colocalized with F-actin to form cortical micro honeycomb-like lattices in suprabasal layers (Fig. 1, f and g; and Fig. S1 c). Whereas αII-spectrin cortical enrichment was highest in the SG3 layer (Fig. 1 h) to then drop again, F-actin localization at the cortex became increasingly more intense in upper layers (Rübsam et al., 2017), peaking in SG1 cells (Fig. 1 i), suggesting that distinct spectrin–actin conformations determine layer-specific cell shapes. In agreement, first suprabasal αII-spectrin enrichment correlated with the initial flattening of suprabasal cells at embryonic (E) day 15 (Fig. S1 d, arrow) and precedes stratum corneum formation (Rübsam et al., 2017). Thus, spectrin is an integral part of the keratinocyte actin network, and stratification-induced cell shape transitions go along with changes in cortical F-actin spectrin ratios.

We thus asked directly whether αII-spectrin regulates epidermal cell shape and depleted αII-spectrin during epidermal development using lentiviruses encoding two different *Sptan1* short hairpins *(shSptan1-0595* or *shSptan1-9753)* or a control *shRNA (shScr)* together with a GFP-tagged histone 2B reporter (H2B-GFP) to identify transduced cells (Fig. S1, e–g) (Beronja et al., 2010). As a readout for cell shape, we then quantified the cell area and the cell shape index (perimeter [P]/√area [A]) (Sahu et al., 2020) of each layer. Depletion of αII-spectrin resulted in less-flattened suprabasal cells, as indicated by an increased cell area and a decrease in shape index (Fig. 1, h and i).

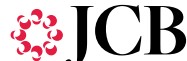

Figure 1. **αII-spectrin determines epidermal cell shape. (a)** Representative examples of 3D-rendered keratinocytes in the different epidermal layers of newborn epidermal confetti mice using cytoplasmic YFP expression. **(b)** Schematic illustration and denotation of the epidermal layers and the distance of the border of each layer relative to the stratum corneum surface. **(c)** Quantification of cell shape parameters from rendered cells. 71 rendered cells from three mice have been analyzed. R-squared values for the linear regression analysis of the accumulated values are shown. **(d)** Newborn epidermal whole-mount immu-nofluorescence analysis for phalloidin (F-actin) revealing top-view cell shapes in the spinous (SS cells) and granular layer (SG cells) within tissue (in vivo). Right column: Phalloidin staining of single cells isolated from the spinous or granular layer showing deformation upon isolation only in spinous cells. **(e)** Quantification of cell top-view area/shape of granular and spinous cells in the tissue and after isolation using stainings as shown in d. *P < 0.05, ****P < 0.0001; >100 cells from three mice (isolated) or six mice (in tissue) with Kolmogorov–Smirnov per layer. **(f)** Newborn epidermal whole-mount immunofluorescence analysis for phalloidin (F-actin) and αII-spectrin. Max. projections of the spinous (SS) and granular layer 3 (SG3) are shown. d and f representative images of N ≥ 3 biological replicates. **(g)** Immunofluorescence analysis for phalloidin (F-actin) and αII-spectrin on newborn skin cryosections. Representative image of n > 3 biological replicates. **(h and i)** Quantification of cortical αII-Spectrin and F-actin intensity across the epidermal layers. Mean values (lines) of pooled cells (dots) from three mice. ****P < 0.0001, ***P = 0.0001, **P < 0.007; n ≥ 157 cells with Kruskal–Wallis, Dunn's multiple comparison test. **(j)** Skin sections from shCtr and *shSptan1 0595*-transduced E17.5 embryos immunolabeled for E-cadherin. Upper Insets show transduced cells (H2B–GFP+). Dashed lines indicate the dermal-epidermal border. **(k)** Quantification of cell sagittal area and cell shape (perimeter/√area) from data shown in j. Mean (lines) ± SD from ~200 individual cells (dots) from n = 3 embryos per condition. ****P < 0.0001 by Kolmogorov–Smirnov test for sagittal cell area. ***P = 0.0002 basal *shSptan1 0595*; *P = 0.0181 for basal *shSptan1 9753*, ****P < 0.0001 by Kolmogorov–Smirnov test for cell shape index.

Basal spectrin-depleted cells also increased in size and became even more spherical. Inactivation of αII-spectrin in all epidermal cells using keratin14-Cre (*Sptan1^epi−/−*) (Fig. S1, h and i) confirmed changes in cell shape in newborn epidermis (Fig. S1, j and k). Thus, αII-spectrin regulates keratinocyte shape in all layers of the epidermis.

### E-cadherin controls cell shape upstream of spectrin

We next asked how spectrin is recruited to the cortex in keratinocytes. One candidate is ankyrin, a spectrin-binding partner that enables its assembly into networks (Bennett and Healy, 2009; Bennett and Lorenzo, 2016). Ankyrin also interacts with E-cadherin (Kong et al., 2023; Kizhatil et al., 2007; Bennett and Lorenzo, 2013). Immunostaining of E17.5 embryos for ankyrin G (encoded by *Ank3*) that is expressed in the developing epidermis (Sennett et al., 2015) revealed cortical enrichment in the granular layer, which is lost upon spectrin depletion (Fig. 2, a, c, and d). In contrast, *Ank3* depletion in E17.5 embryos (Fig. S2, a–c) did not alter spectrin localization or levels (Fig. 2, a and b), with no changes in F-actin localization and cell shape (Fig. S2, d–g).

E-cadherin–based AJ can recruit spectrin to the cortex (Pradhan et al., 2001). Epidermal inactivation of E-cadherin (*Ecad^epi−/−*) (Tunggal et al., 2005) resulted in loss of αII-spectrin from the cortex of suprabasal layers (Fig. 2 e), whereas F-actin remained cortical but with increased intensity in the spinous layer (Rübsam et al., 2017). In contrast, E-cadherin maintained its localization in αII-spectrin–depleted epidermis (Fig. 1 j). Consistently, αII-spectrin was also mislocalized in cultured E-cadherin depleted keratinocytes (Fig. S2 h), whereas depletion of αII-spectrin did not affect AJ assembly in vitro (Fig. S2 i).

We then asked whether cortical recruitment of spectrin is necessary to control cell shape. Like spectrin depletion, epidermal loss of E-cadherin altered cell shapes in all layers with a reduced flattening of stratum granular (SG) cells (Fig. 2, f and g). Isolated *E-cad^−/−* SG1/SG2 cells were also smaller but maintained their flattened tetrakaidecahedron cell shape, even when treated with latrunculin B to depolymerize F-actin (Fig. 2, h and i; and Fig. S2, j and k), indicating that once the shape of SG2 cells is set this shape becomes independent of AJ or F-actin. In contrast to spectrin depletion, *Ecad^−/−* spinous layer cells showed increased flattening, perhaps due to the increased cortical actin in this layer. Together, these data indicate that E-cadherin, through polarized organization of the spectrin-actomyosin cortex, regulates transitions in cell shape that guide terminal differentiation when cells move up through the epidermis.

### Cortical F-actin and spectrin organization are mutually dependent

We next asked whether F-actin and spectrin coordinate the organization of the epidermal cortex into a honeycomb lattice. Treatment of E17.5 embryos with low levels of latrunculin B to reduce F-actin but not disturb E-cadherin adhesion (Fig. S3 a) reduced cortical αII-spectrin levels in all layers of the epidermis (Fig. 3, a and b). Conversely, embryonic KD of αII-spectrin reduced suprabasal cortical F-actin levels in E17.5 embryos (Fig. 3, c and d). High-resolution imaging of newborn epidermal whole mounts revealed a striking loss of F-actin honeycomb lattices in

*Sptan1^epi−/−* mice, with over 60% of suprabasal cells now showing a more streak- or spot-like organization (Fig. 3, e and f, arrow), indicating that spectrin not only recruits but is essential to properly organize F-actin at the cortex.

We then asked how AJ assembly initiates the organization of the actin-spectrin network by switching primary keratinocytes to high Ca²⁺ (1.8 mM) to induce contact formation and stratification. Upon initial E-cadherin engagement (6 h Ca²⁺), only F-actin but not spectrin was recruited to early AJ, so-called AJ zippers (Fig. 3, g and h), to form peri-junctional rings (Vasioukhin et al., 2000) (Fig. 3, g and h; and Fig. S3 b). Consistently, αII-spectrin knockdown did not affect initial E-cadherin engagement or actin recruitment (6 h Ca²⁺, Fig. S2 i).

Stratification of keratinocytes into a multilayer (48 h Ca²⁺) upregulated spectrin protein levels (Fig. S3 c), in agreement with the in vivo suprabasal increase in cortical spectrin (Fig. 1, g and h). As seen in vivo, spectrin co-localized with F-actin and E-cadherin to form micro honeycomb-like cortical F-actin-spectrin lattices (Fig. 3 g, arrow, h; Fig. S3, b and d). Spectrin was also localized to the F-actin cortical ring that supported barrier-forming TJs in the uppermost apical cell layer, albeit to a much lesser extent than F-actin (Fig. 3, g and h, arrowhead; Fig. S3 b) (Rübsam et al., 2017).

Low levels of latrunculin B reduced cortical αII-spectrin recruitment, similar to in vivo, and disturbed the honeycomb organization (Fig. 3 i and Fig. S3 e). Strikingly, depletion of αII-spectrin also resulted in a loss of cortical honeycomb lattices, with F-actin now showing a linear fiber organization reminiscent of F-actin stress fibers (Fig. 3 j). In contrast, αII-spectrin KD did not prevent formation of the apical TJ-associated F-actin cortical ring even though F-actin levels were reduced (Fig. S3, h and i).

Thus, F-actin and spectrin spatiotemporally coordinate the organization of the cortex during stratification. Upon initial adhesion, AJs first recruit F-actin and then reorganize the F-actin cortex (Vaezi et al., 2002) to allow recruitment of αII-spectrin necessary to further restructure the cortex into F-actin-spectrin lattices in suprabasal layers.

### Spectrin stabilizes contractile cortical actomyosin networks

The observed reorganization of F-actin into stress fiber-like structures upon loss of αII-spectrin suggested a change in myosin recruitment and activity. The cortex of keratinocytes is highly enriched for the actin-binding protein myosin II that regulates contractile states of the actin cytoskeleton (Dor-On et al., 2017; Sumigray et al., 2012). In *Drosophila*, loss of either β- or α-spectrin increased junctional myosin levels and activity (Deng et al., 2015; Forest et al., 2018; Ibar et al., 2023). We thus examined where at the cortex myosin II was recruited. Whole-mount imaging for non-muscle myosin heavy chain IIa (myosin-IIa) revealed an enrichment of myosin-IIa mostly at the lateral junctions in the spinous layer (Fig. 4 a, SS). Staining increased in the granular layer with myosin-IIa, now mostly located to spots that decorated the actin-spectrin honeycomb lattices (Fig. 4 a, SG3/2), thus indicating that cells increase their tensile state when moving from the spinous to the granular layer. Loss of spectrin increased myosin intensity and spot size, suggesting an

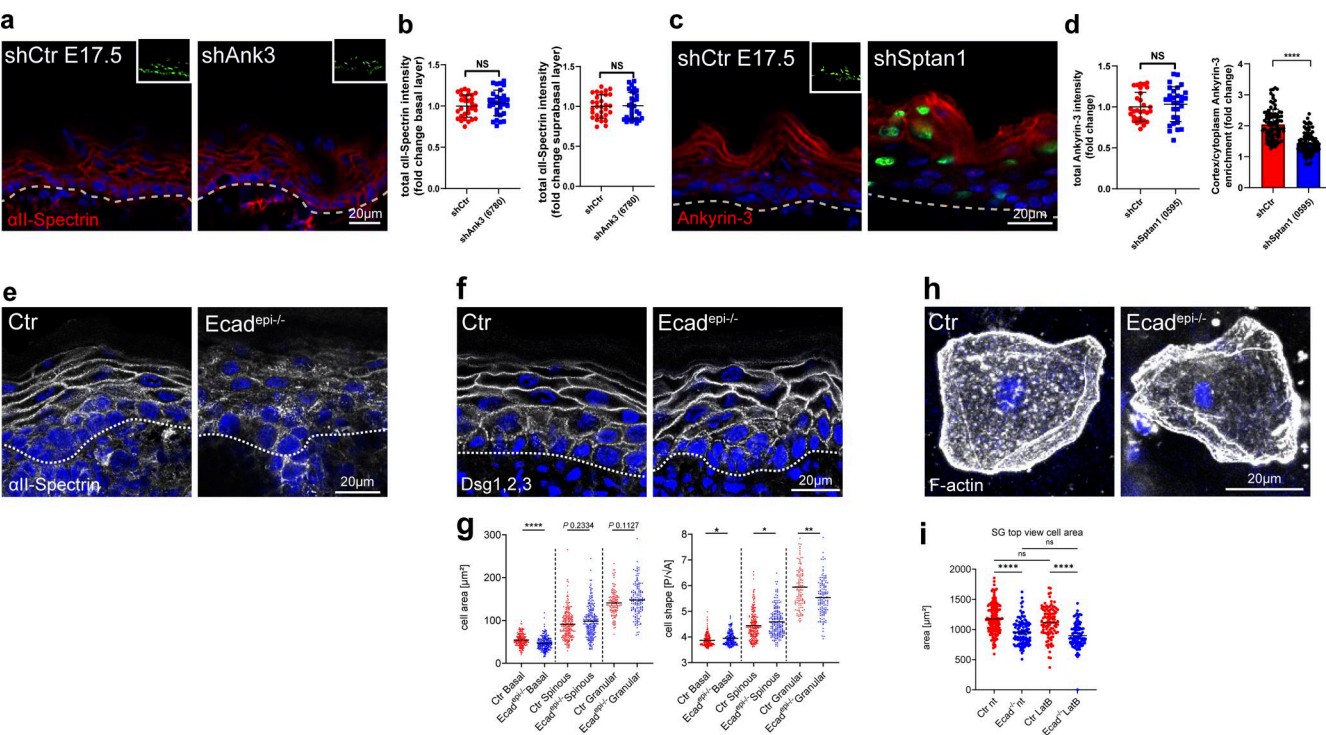

Figure 2. **E-cadherin controls cell shape upstream of spectrin. (a)** Sagittal views of dorsal skin sections from shCtr and *shAnk3* 6780-transduced E17.5 embryos immunolabeled for αII-spectrin. Insets show the transduced cells (H2B–GFP+). **(b)** Quantification of basal (left) and suprabasal (right) layer αII-spectrin intensity from data shown in a. Mean ± SD of 30 ROI from *n* = 3 embryos per condition. Bars: mean normalized intensity; dots: microscopy fields. NS: P = 0.586 (basal), NS: P = 0.9525 with Kolmogorov–Smirnov. **(c)** Dorsal skin sections from shCtr and mosaic *shSptan1* 0595-transduced E17.5 embryos immunolabeled for Ankyrin-3. Upper insets show the transduced cells (H2B–GFP+). **(d)** Left graph: Quantification of Ankyrin-3 intensity from data shown in c. Mean ± SD of 30 ROI from *n* = 3 embryos per condition. Bars: mean normalized intensity; dots: microscopy fields. NS: P = 0.235 with Kolmogorov–Smirnov. Right graph: Quantification of Ankyrin-3 cortical enrichment from the data shown in c. Mean ± SD from 90 individual cells from *n* = 3 embryos per condition. Bars: Ankyrin-3 cortex/cytoplasm intensity ration mean; dots: individual cells. ****P < 0.0001 with Kolmogorov–Smirnov. **(e)** αII-Spectrin staining on Ctr and E-cadherin–deficient newborn epidermis sections showing impaired cortical recruitment of αII-Spectrin upon loss of E-cadherin. Representative images of *n* = 3 biological replicates. **(f)** Immunofluorescence analysis for shape using combined staining for desmoglein1,2,3 (Dsg1,2,3) on Ctr and E-cadherin–deficient newborn epidermis sections. **(g)** Quantification of cell sagittal area and shape (perimeter/√area)/layer using stainings as shown in f. **P < 0.005, *P < 0.03; cells: *n* = 420 (basal), *n* = 445 (spinous), 277 (granular) with Kolmogorov–Smirnov per layer. **(h)** Phalloidin staining of single cells isolated from the granular layer of Ctr or E-cadherin–deficient newborn epidermis. **(i)** Quantification of the cell top-view area of isolated SG cells from Ctr and E-cadherin–deficient newborn epidermis as shown in h and treated with latrunculin B (0.1 µM). Dots represent individual cells isolated from three mice. ****P < 0.0001; *n* ≥ 98 cells with Kruskal–Wallis, Dunn's multiple comparison test. All images: Nuclei were stained with DAPI (blue). ROI, region of interest.

increase in the contractile state of F-actin networks (Fig. 4 a), similar to *Drosophila*. Vice versa, inhibition of myosin contractility using the Rho-associated protein kinase inhibitor, Y27632, decreased cortical αII-spectrin levels in E17.5 embryos (Fig. 4, b and c), thus indicating that spectrin and myosin II jointly coordinate organization and tensile states of the F-actin cortex.

We next used primary keratinocytes to assess whether spectrin-dependent recruitment of myosin II altered the tensile state of the cortex. Co-staining of F-actin, αII-spectrin, and myosin-IIa in stratified control keratinocytes also revealed a spot-like integration of myosin IIa into the spectrin-F-actin lattices. Upon depletion of αII-spectrin, the now linear F-actin fibers (Fig. 3 j) were intensely decorated with myosin-IIa in a highly periodic pattern, indicating that these fibers are highly contractile (Fig. 4 d and Fig. S4 a). Lowering myosin II motor activity (blebbistatin, 5 µM) in control and spectrin-depleted keratinocytes was sufficient in both to reorganize honeycomb spectrin-F-actin lattices into ultrafine, almost diffuse F-actin structures (Fig. 4 e and Fig. S4 b), except upon spectrin

depletion, their appearance was more anisotropic and streaklike (Fig. 4, e and f, arrow; Fig. S4 b), thus resembling the in vivo F-actin defects observed in the αII-spectrin–deficient epidermis (Fig. 3 e). Together, these data show that tension is necessary to properly assemble spectrin and actin into a honeycomb network, with spectrin controlling its tension state, albeit to different extents in vivo versus in vitro.

We then assessed the mechanical properties of the cortex by performing linear laser ablation experiments to measure straindependent recoil in stratified keratinocytes. Prior to the formation of cortical spectrin-actomyosin lattices, the ablation of F-actin–positive intercellular junctions elicited a viscoelastic recoil response of junctional vertices as described for simple epithelial cells (Fig. S4, c and d) (Wu et al., 2014). In contrast, virtually no recoil of junctional vertices was observed once spectrin-actomyosin lattices were formed (Fig. S4, c and d). Either subsequent ablation of the lattices themselves adjacent to the already-ablated junctions or ablation of only these lattices induced a strong recoil response with elliptical openings that

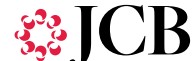

**Figure 3. Cortical F-actin and spectrin organization are mutually dependent. (a)** Dorsal skin sections from E17.5 wild-type embryos treated with DMSO or latrunculin B (2.5 µM) immunolabeled for αII-spectrin. **(b)** Quantification of basal (left graph) and suprabasal (right graph) layer αII-spectrin intensity from data shown in a. Mean ± SD of 30 ROI from $n$ = 3 embryos per condition. Bars: mean normalized intensity; dots: microscopy fields. **P = 0.0072 (basal, latrunculin B), **P = 0.0165 (suprabasal, latrunculin B) with Kolmogorov–Smirnov. **(c)** Dorsal skin sections from shCtr and mosaic shSptan1 0595-transduced E17.5 embryos labeled for F-actin. Upper insets show the transduced cells (H2B–GFP+). **(d)** Quantification of basal (left) and suprabasal (right) layer F-actin intensity from data shown in c. Mean ± SD of 30 ROI from $n$ = 3 embryos per condition. NS: P = 0.1344 (shSptan1 0595, basal), P = 0.354 (shSptan1 9753, basal), **P = 0.001 (shSptan1 0595, suprabasal), **P = 0.001 (shSptan1 9753, suprabasal) by unpaired $t$-test. Nuclei were stained with DAPI; dotted lines indicate the dermal-epidermal border. NS, not significant. **(e)** Newborn epidermal whole-mount immunofluorescence analysis for cortical F-actin organization upon loss of αII-spectrin. Note streak-like (arrow) and spot-like reorganization of F-actin upon loss of αII-spectrin. Representative images of four biological replicates (mice). **(f)** Quantification of the percentage of cells in the granular layer showing either normal (lattice-like) or abnormal (streak-like or spot-like) F-actin organization. *P < 0.05 with Mann–Whitney for the mean of $n$ = 4 biological replicates, including 466 (Ctr) and 385 (Sptan1$^{epi-/-}$) analyzed cells. **(g)** Immunofluorescence analysis for αII-spectrin and F-actin after 6 h or 48 h in high Ca$^{2+}$ at cell–cell interfaces (arrow: actin-spectrin lattice, arrowhead: TJ-supporting apical F-actin ring) and apical surface. Representative images of $n$ ≥ 3 biological replicates each. **(h)** Schematic representation of the image position in the mono or multilayered keratinocytes. **(i)** Immunofluorescence analysis for αII-spectrin and F-actin after 48 h in high Ca$^{2+}$ at cell–cell interfaces with and without latrunculin B treatment (1 h, 0.1 µM). **(j)** Immunofluorescence analysis for αII-spectrin and F-actin (48 h high Ca$^{2+}$) upon αII-spectrin knockdown. Representative images of $n$ = 3 biological replicates each. ROI, region of interest.

increased in size over time (Fig. 4, g–i; and Fig. S4, e and f). These data indicate that the cortical spectrin-actomyosin lattices exhibit viscoelastic properties as shown previously in zebrafish and *C. elegans* (Saha et al., 2016; Thi Kim Vuong-Brender et al., 2017). Interestingly, curved intercellular borders straightened upon ablation, showing that the spectrin-actomyosin lattice also regulated the tension state of AJs to control cell shape (Fig. S4 g). Importantly, depleting αII-spectrin resulted in a faster, two-fold larger recoil with highly irregular openings (Fig. 4, j and k), which was reversed upon blebbistatin treatment (Fig. 4, l and m). Further, E-cadherin loss induced similar recoil behavior with highly irregular, larger openings (Fig. 4, n and o). Taken

together, these data suggest a model in which E-cadherin, through recruitment of spectrin, orchestrates distinct cortical networks in each layer to dissipate the myosin-driven increase in tension and increase the stability of the cortex to prevent damage.

### αII-spectrin enhances epidermal barrier formation

Although suprabasal flattening is a hallmark of differentiation, whether these changes in cell shape promote epidermal differentiation is still unclear (Simpson et al., 2011; Luxenburg and Zaidel-Bar, 2019). We thus asked whether depletion of αII-spectrin also changes epidermal differentiation. In E17.5 control embryos,

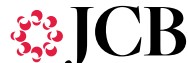

Figure 4. **Spectrin determines junctional actomyosin network structure and stability. (a)** Newborn epidermal whole-mount immunofluorescence analysis for phalloidin (F-actin) and non-muscle myosin heavy chain IIa (myosin-IIa). Minimal max. projections of the denoted layers are shown. **(b)** Dorsal skin sections from E17.5 wild-type embryos treated with DMSO or Y27632 (40 μM) immunolabeled for αII-spectrin. **(c)** Quantification of basal (left graph) and suprabasal (right graph) layer αII-spectrin intensity from data shown in b. Data are the mean ± SD of 30 ROI from $n$ = 3 embryos per condition. Bars: mean normalized intensity; dots: microscopy fields. *P ≤ 0.05; NS: P = 0.3876 with Kolmogorov–Smirnov. **(d)** Immunofluorescence analysis for αII-spectrin, F-actin, and non-muscle myosin heavy chain IIa (myosin-IIa) (48 h high Ca$^{2+}$) in Ctr and αII-spectrin–deficient (KO) cells. **(e)** Immunofluorescence analysis for αII-spectrin and F-actin (48 h high Ca$^{2+}$) upon αII-spectrin (*Sptan1*) knockdown and treatment with either DMSO or low-dose blebbistatin (5 μM). Representative images of $n$ = 3 biological replicates each. **(f)** Immunofluorescence analysis for F-actin (48 h high Ca$^{2+}$) upon αII-spectrin (*Sptan1*) knockdown at low tension (low-dose blebbistatin 5 μM) showing streak-like defects similar to in vivo. Representative images of $n$ = 3 biological replicates each. **(g)** Illustration of laser ablation in multilayered keratinocytes. **(h)** Laser ablation: Still images from live imaging of SiR-actin (F-actin)-labeled Ctr cells (48 h high Ca$^{2+}$) at indicated time points after 17 μm line ablation showing progressive elliptical cortical openings. **(i)** Quantification of the elliptical opening area (red line) i.e., recoil over time. Line: mean ± SD opening area of 13 ablations from $N$ = 4 biological replicates. **(j)** Cortical laser ablation: Still images from live imaging of SiR-actin (F-actin)–labeled Ctr (as shown h) and αII-spectrin knockdown keratinocytes at the time point of analysis (60 s after a linear laser cut [yellow line]) after 48 h high Ca$^{2+}$. The opening in the F-actin cortex is seen as black area around the yellow line. **(k)** Quantification of opening areas in the apical F-actin cortex upon linear laser ablation as shown in j. Lines represent means, dots represent single openings/cells pooled from $n$ = 3 independent experiments/biological replicates. *P = 0.0215 with Kolmogorov–Smirnov. **(l)** Laser ablation (as described for j) of αII-spectrin knockdown keratinocytes treated with DMSO or low dose blebbistatin (5 μM) 1 h prior to ablation. **(m)** Quantification of opening areas in the apical F-actin cortex upon linear laser ablation as shown in l. Lines represent means; dots

represent single openings/cells pooled from *n* = 3 independent experiments/biological replicates. ****P < 0.0001 with Kolmogorov–Smirnov. **(n)** Laser ablation (as described for j) of E-cadherin⁻/⁻ keratinocytes after 48 h high Ca²⁺. **(o)** Quantification of opening areas in the apical F-actin cortex upon linear laser ablation as shown in *N*. Lines represent means; dots represent single openings/cells pooled from *n* = 3 independent experiments/biological replicates. ****P < 0.0001 with Kolmogorov–Smirnov. ROI, region of interest.

staining for keratin (K)14 only marked the basal layer, whereas in *Sptan1ᴷᴰ* epidermis, suprabasal cells were also K14⁺ (Fig. 5 a arrow; Fig. S5, a and b). In agreement, EdU incorporation assays showed that suprabasal proliferation was increased threefold compared with control (Fig. S5, d and e). In contrast, the suprabasal marker K10 was not obviously altered, indicating at least a partial induction of differentiation (Fig. 5 a and Fig. S5 a). Basal spindle orientation that can regulate cell fate (Williams et al., 2011) was also not altered (Fig. S5 c). Thus, in addition to cell shape, spectrin is critical for initiating basal cell differentiation upon translocating suprabasally. We next asked whether the ability to induce differentiation is linked to changes in cell shape. Although both K14⁺- and K14⁻ suprabasal cells showed an increase in cell area compared with control suprabasal cells (Fig. 5 b), only K14⁺ but not K14⁻ suprabasal cells exhibited a defect in flattening, suggesting that flattening but not cell shape changes per se are linked to initiation of differentiation. Further, in contrast to control, where loricrin⁺ granules filled the entire cytoplasm, loricrin was detected predominantly at the cell periphery in the granular layer cells of *Sptan1ᴷᴰ* embryos (Fig. 5 a and Fig. S5 b), suggesting impaired terminal differentiation and barrier formation.

TJ formation is a key feature of the SG2 layer and is necessary for epidermal barrier function (Kubo et al., 2009; Rübsam et al., 2017). To follow the formation of a functional TJ barrier and determine whether spectrin is required, we stratified primary keratinocytes and measured transepithelial electrical resistance (TEER). Although depleting spectrin did not obviously alter the recruitment of the TJ marker occludin to the apical cell–cell contacts (Fig. 5 c), TEER remained low over time (Fig. 5 d), thus indicating the formation of leaky TJs that could not be explained by lower cell numbers (Fig. S5 f). In the mouse epidermis, the tetrakaidecahedron-shaped SG2 cells that transition out of the SG2 layer not only have mature apical TJs but also form a new nascent TJ ring at basal tricellular contacts, which is smaller but aligned with and mirrors the shape of the upper, mature TJ ring (Fig. 5 e, Ctr) due to their geometric stacking arrangement (Yokouchi et al., 2016). This shape correlation and alignment were lost upon loss of αII-spectrin (Fig. 5, e and f), likely due to the alterations in cell shape, which may explain dysfunctional TJs.

To further explore changes in terminal differentiation, we assessed the activity of the cross-linker enzymes TGMs in the granular layer, necessary for terminal differentiation and proper formation of the outermost stratum corneum barrier (Eckert et al., 2005; Simpson et al., 2011). Incubation with a fluorescent substrate peptide to localize TGM1 activity (Sugimura et al., 2008) showed high enrichment of this peptide at the cell cortex of control granular layer cells, which was reduced in *Sptan1ᴷᴰ* E17.5 embryos and *Sptan1ᵉᵖⁱ⁻/⁻* newborns (Fig. 5, g–i; and Fig. S5, g and h). Similarly, *Ecadᵉᵖⁱ⁻/⁻* epidermis also showed decreased TGM activity (Fig. 5, l and m). TGM1 protein levels were not changed upon epidermal loss of either αII-spectrin or E-cadherin

(Fig. S5, i–k). Thus, E-cadherin–dependent cortical organization of spectrin-actomyosin networks directs TGM activation necessary for terminal differentiation (Fig. S5, k and l). Toluidine blue exclusion assays (Hardman et al., 1998) showed no dye penetration in control E17.5 embryos except most ventrally, indicating the formation of an intact barrier (Hardman et al., 1998). In contrast, E17.5 embryos from *Sptan1ᴷᴰ* and *Sptan1ᵉᵖⁱ⁻/⁻* showed dye penetration in the head and larger ventral parts (Fig. 5, j and k; and Fig. S5 l), and E18.5 *Sptan1ᴷᴰ* embryos still showed staining of the head region (Fig. S5 m), indicating a delay in stratum corneum barrier formation. However, transepidermal water loss was not obviously increased in *Sptan1ᵉᵖⁱ⁻/⁻* newborns (Fig. 5 n), suggesting a transient defect in barrier formation during development. Taken together, spectrin regulates cell shape and promotes differentiation to enable functional TJ organization and cortical TGM activation downstream of E-cadherin in the upper suprabasal layers, necessary for proper barrier formation.

## αII-spectrin is required for the activity of the EGFR–TRPV3–TGM pathway

We previously showed that E-cadherin controls TJ formation by regulating the EGFR (Rübsam et al., 2017). Furthermore, EGFR activation promotes the opening of the calcium channel TRPV3 necessary for TGM1 activation and subsequent terminal epidermal differentiation (Cheng et al., 2010). As spectrin functions as a cortical platform that organizes membrane domains and membrane protein activities (Machnicka et al., 2014), we hypothesized that αII-spectrin promotes terminal differentiation through the EGFR–TRPV3 pathway. Immunofluorescence analysis showed EGFR staining in all epidermal layers in both control and *Sptan1ᴷᴰ* E17.5 embryos (Fig. S6 a). Whereas phosphorylated EGFR (pEGFR) was highly enriched at the cortex of Ctr granular layer cells, many *Sptan1ᴷᴰ* granular layer cells showed diffuse or no pEGFR staining, with now many cells showing ectopic pEGFR staining, especially in the spinous layer (Fig. 6, a–c).

To determine whether αII-spectrin controls TRPV3 localization, we infected E9 embryos with lentivirus encoding TRPV3-GFP (Xiao et al., 2008). In control E17.5 epidermis, TRPV3ᴳᶠᴾ was recruited to the cortex in the granular layer (Fig. 6 d), where it co-localized with pEGFR (Fig. S6 b). In contrast, TRPV3ᴳᶠᴾ remained more diffuse in the *Sptan1ᴷᴰ* granular layer (Fig. 6 d and Fig. S6 c). Furthermore, colocalization of endogenous TRPV3 with E-cadherin at the cell membrane was significantly reduced in *Sptan1ᴷᴰ* primary keratinocytes (Fig. 6, f and g). Thus, spectrin spatially controls activation of EGFR and localization of TRPV3 at the cortex.

## αII-spectrin–actomyosin networks regulate the EGFR–TRPV3–TGM pathway

Given that spectrin directs the layer-specific organization and mechanics of the cortical cytoskeleton, we asked whether αII-spectrin

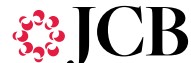

Figure 5. **αII-spectrin regulates epidermal differentiation and barrier function. (a)** Dorsal skin from shCtr and *shSptan1 0595*-transduced E17.5 embryos immunolabeled for the basal layer marker K14, the differentiation marker K10, and the granular layer marker loricrin. Insets show the transduced cells (H2B–GFP+). **(b)** Quantification of cell area and shape of suprabasal *shSptan1* K14+ and K14- cells in E17.5 embryos. Values of basal and suprabasal shCtr and basal *shSptan1* are equal to Fig. 1 Lines: Mean values; dots: single cells pooled from 3 embryos with >100 cells for each condition. ****P < 0.0001 with Kruskal–Wallis, Dunn's multiple comparison test. **(c)** Immunofluorescence analysis for the TJ marker occludin in Ctr and *Sptan1⁻/⁻* primary keratinocytes differentiated for 48 h in high Ca²⁺. Representative example of three biological replicates each. **(d)** Transepithelial resistance (TER) measurements in Ctr and αII-spectrin knockdown keratinocytes after switching to high Ca²⁺. Line represent means over time of three biological replicates each. Representative experiment of *n* > 10 biological replicates. **(e)** Newborn epidermal whole-mount immunofluorescence analysis for the TJ marker ZO-1, revealing impaired alignment of the upper old (red arrowheads) and the lower new TJ rings (blue arrowheads) in the granular layer 2 (SG2). Maximum projection of the granular layer. **(f)** Illustration of cell shapes and TJ organization in the SG2 of Ctr and *Sptan1epi⁻/⁻* epidermis. **(g)** Dorsal skin sections from shCtr and *shSptan1 0595*-transduced E17.5 embryos. Sections were processed for transglutaminase 1 (TGM1) activity assay. Upper Insets show the transduced cells (H2B–RFP+). **(h)** Quantification of TGM1 intensity from data shown in g. Data are the mean ± SD of 30 ROI from *n* = 3 embryos per condition. Bars: mean normalized intensity; dots individual microscopy fields. *P = 0.0471 by unpaired *t*-test. **(i)** Quantification of TGM1 activity cortical enrichment from the data shown in g. Mean ± SD from 60 individual cells from *n* = 3 embryos per condition. Bars: TGM1 activity cortex/cytoplasm intensity ratio mean; dots: individual cells. ***P = 0.0003 with Kolmogorov–Smirnov. **(j)** Dye exclusion assay: shCtr and *shSptan1 0595*-transduced E17.5 embryos were treated with toluidine blue dye to evaluate the skin barrier. **(k)** Dye exclusion assay: Ctr and *Sptan1epi⁻/⁻* E17.5 embryos were treated with toluidine blue dye to evaluate the skin barrier. **(l)** Dorsal skin section from Ctr and *E-cadherinepi⁻/⁻* newborn mice. Sections were processed for TGM1 activity assay or negative Ctr (mutated TGM substrate, pepQNK5). **(m)** Quantification of TGM1 intensity from data shown in l. **(n)** Transepidermal water loss (TEWL) measurements on Ctr and *Sptan1epi⁻/⁻* newborn mice. Dots represent individual mice. Data are the mean of 27 fields of view from *n* = 3 newborn mice per condition. Bars: Mean intensity; dots individual microscopy fields. ****P < 0.0001 with Kolmogorov–Smirnov. ROI, region of interest.

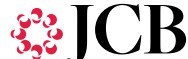

**Figure 6. αII-spectrin–actomyosin networks regulate the EGFR–TRPV3–TGM pathway. (a)** Dorsal skin sections from shCtr, and *shSptan1 0595*-transduced E17.5 embryos immunolabeled for pEGFR. Upper insets show transduced cells (H2B–GFP+). **(b)** Quantification of pEGFR granular layer intensity from data shown in a. Mean ± SD of 30 region of interest (ROI) from *n* = 3 embryos per condition. Horizontal bars: mean normalized intensity, dots: individual microscopy fields. **P = 0.0011 with Kolmogorov–Smirnov. **(c)** Quantification of ectopic pEGFR-positive cells from data shown in a. Horizontal bars: mean ± SD from *n* = 3 embryos per condition, dots: average ectopic pEGFR-positive cells per embryo. P = 0.1 with Kolmogorov–Smirnov. **(d)** Sections of dorsal skin from shCtr; TRPV3-GFP and *shSptan1 0595*; TRPV3-GFP–transduced E17.5 embryos. **(e)** Quantification of TRPV3-GFP cortical enrichment from the data shown in d. Mean ± SD from 30 individual cells from *n* = 3 embryos per condition. Bars: TRPV3-GFP cortex/cytoplasm ratio, dots: individual cells. ****P < 0.0001 with Kolmogorov–Smirnov. **(f)** ShCtr and *shSptan1-0595*–transduced primary mouse keratinocytes (H2B-GFP+) cultured in high-calcium (1.5 mM) medium and immunolabelled for E-cadherin and TRPV3. Insets (right) show magnifications of the boxed areas. **(g)** Quantification of E-cadherin and TRPV3 intensity. Mean ± SD from ∼200 mature junctions from *n* = 3 experiment per condition. Bars: mean normalized intensity, dots: mature junctions. *P < 0.05, ****P < 0.0001 with Kolmogorov–Smirnov. **(h)** Dorsal skin sections from E17.5 wild-type embryos treated with DMSO, latrunculin, or Y27632 immunolabeled for p-EGFR. **(i)** Quantification of pEGFR granular layer intensity from data shown in h. Mean ± SD of 30 ROI from *n* = 3 embryos per condition. Bars: mean normalized intensity; dots: individual microscopy fields with Kolmogorov–Smirnov. **(j)** Quantification of ectopic pEGFR-positive cells from data shown in h. Bars: mean ± SD from *n* = 3 embryos per condition. Dots: average ectopic pEGFR-positive cells from each embryo with Mann–Whitney. **(k)** Dorsal skin section from shCtr; TRPV3-GFP–transduced E17.5 embryos treated with DMSO, latrunculin or Y27632. Insets show the magnification of the boxed area from each epidermal layer. **(l)** Quantification of TRPV3-GFP cortical enrichment from the data shown in k. Data are the mean ± SD from 30 individual cells from *n* = 3 embryos per condition. Horizontal bars represent the TRPV3-GFP cortex/cytoplasm intensity ratio mean, and circles represent individual cells. ****P < 0.0001 with Kolmogorov–Smirnov. **(m)** Primary mouse keratinocytes cultured in high-calcium (1.5 mM) medium treated with DMSO, latrunculin, or Y27632 and immunolabelled for

E-cadherin and TRPV3. **(n)** Quantification of TRPV3 intensity from data shown in c. Mean ± SD from ~150 mature junctions from *n* = 3 experiment per condition. Bars: mean normalized intensity; dots: mature junctions. ****P < 0.0001 with Kolmogorov–Smirnov. **(o)** Dorsal skin sections from E17.5 wild-type embryos treated with DMSO, latrunculin, or Y27632 and processed for TGM activity assay. **(p)** Quantification of crosslinked TGM substrate intensity. Mean ± SD of 30 ROIs from *n* = 3 embryos per condition. Bars: mean normalized intensity; dots: individual microscopy fields. **P = 0.0072 (latrunculin B), *P = 0.0354 (Y27632) with Kolmogorov–Smirnov. **(q)** Quantification of cortically enriched cross-linked TGM substrate. Mean ± SD from 60 individual cells from *n* = 3 embryos per condition. Bars: means of cortex/cytoplasm ratio; dots: individual cells. ****P < 0.0001 with Kolmogorov–Smirnov. Nuclei were stained with DAPI; dotted lines indicate the dermal-epidermal border.

regulates the activation of the EGFR–TRPV3–TGM axis through actomyosin. To this end, we treated shCtr E17.5 embryos with latrunculin and Y27632 to decrease F-actin levels and myosin II motor activity, respectively. Similar to depletion of αII-spectrin, latrunculin or Y27632 treatment decreased pEGFR levels in the granular layer and increased spinous cortical pEGFR levels relative to DMSO-treated embryos (Fig. 6, h–j). Importantly, latrunculin or Y27632 also decreased the enrichment of TRPV3[GFP] in vivo (Fig. 6, k and l) and the recruitment of endogenous TRPV3 to intercellular contacts in cultured keratinocytes (Fig. 6, m and n; and Fig. S6, d and e). Moreover, these treatments further decreased cortical enrichment of the TGM1 substrate peptide in E17.5 embryos (Fig. 6, o–q).

Together, the results demonstrate that the layer-specific spectrin-actomyosin organization and mechanics shaped by E-cadherin and distinctive spectrin and F-actin gradients control the spatial confinement and activation of EGFR and TRPV3 at the cortex to only activate TGM in the granular layer, thereby promoting epidermal barrier formation.

### EGFR activity regulates cortical TRPV3 localization

To further explore the relationship between EGFR activity, known to alter TJs and cortical mechanics (Muhamed et al., 2016; Rübsam et al., 2017), and spectrin-actomyosin cortical organization and TRPV3 localization, we inhibited EGFR using gefitinib in either E17.5 TRPV3[GFP]-expressing embryos or in cultured keratinocytes. Gefitinib treatment significantly reduced cortical enrichment of both αII-spectrin and TRPV3 (Fig. 7, a–g). Interestingly, activation of EGFR by TGFα treatment also decreased αII-spectrin and TRPV3 localization at intercellular junctions (Fig. 7, d–g).

Finally, we asked whether EGFR activity regulates TGM1 activity in the epidermis. As in *Sptan1*[KD] embryos (Fig. 5, g–i), cortical enrichment of the TGM1 substrate peptide was markedly decreased upon gefitinib treatment, indicating reduced cortical TGM1 activity (Fig. 7, h–j). Thus, in the granular layer, it spatially confines EGFR/TRPV3 activity at the cell membrane to promote terminal differentiation and establish a functional epidermis. Taken together, these and previous findings (Rübsam et al., 2017; Cheng et al., 2010) highlight a critical interdependency between the specific organization and tensile state of the spectrin-actomyosin cortex and signaling to coordinate the characteristic cell shape with the proper differentiation state in each epidermal layer.

## Discussion

In 1858, physician Rudolf Virchow established a correlation between cell shape and function, describing how abnormalities in cell shape can lead to disease development (Virchow, 1871). While the regulation of cell shape is a well-studied process dependent on adhesion and actomyosin, the mechanisms by which these processes also regulate differentiation and function require further elucidation. The layered structure of the epidermis provides a unique example of the correlation between cell position, shape, and function, i.e., differentiation and barrier function (Peskoller et al., 2022; Luxenburg and Zaidel-Bar, 2019). This study demonstrates that E-cadherin–dependent integration of αII-spectrin into submembranous actomyosin networks is crucial for the differential organization and tension state of the cortex to coordinate cell shape and differentiation state in each layer, essential to form a functional skin barrier.

Previously, we demonstrated that E-cadherin, a master regulator of cell–cell adhesion, is a key regulator of actomyosin-dependent tension to spatially control the localization and activation of EGFR, essential for TJ barrier function in the epidermis (Rübsam et al., 2017). However, how E-cadherin controls polarized localization of actomyosin tension and EGFR activity across layers remained unclear. Our new data now show that differential gradients of F-actin, myosin II, and spectrin drive a layer-specific organization of honeycomb-shaped spectrin-actomyosin networks necessary to dissipate myosin-dependent tension. Further, spectrin's elastic properties likely allow the cortex to not only accommodate the change in cell shape when moving into a new layer but also to absorb forces experienced during this movement. Finally, this layer-specific organization of the spectrin-actomyosin cortical network regulates the localization and activity of EGFR/TRPV3 Ca[2+] channel complexes necessary to activate TGM only in the granular layer to promote terminal differentiation and epidermal barrier function. Our data thus link cell adhesion with position-dependent cytoskeletal configuration to direct spatial control of cell shape, cell fate, and barrier function.

Previous studies on cell shape-dependent control of differentiation in the epidermis focused on early differentiation of basal progenitor cells and involved regulation of spindle orientation as well as compression-induced delamination (Le et al., 2016; Luxenburg et al., 2011; Miroshnikova et al., 2018). In contrast, other studies in which different actin-binding proteins, such as cofilin, WDR1, thymosin β4, and others, were deleted in the epidermis did not show any obvious changes in global differentiation markers (Luxenburg et al., 2015; Padmanabhan et al., 2020; Mahly et al., 2022). Similarly, K10 was not obviously altered upon loss of αII-spectrin, even if the basal marker K14 was upregulated. Our data now show that changes in cortical spectrin-actomyosin organization not only control basal cell shape and initial differentiation but also drive suprabasal cell

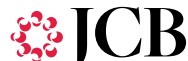

Figure 7. **EGFR activity regulates cortical TRPV3 localization. (a)** Dorsal skin sections from shCtr; TRPV3-GFP–transduced E17.5 embryos treated with DMSO or gefitinib immunolabeled for αII-spectrin and pEGFR. **(b and c)** Quantification of TRPV3-GFP and αII-spectrin cortical enrichment from the data shown

in a. Mean ± SD from 30 individual cells from *n* = 3 embryos per condition. Bars: TRPV3-GFP cortex/cytoplasm ratio; dots: individual cells. **P = 0.0029 for TRPV3-GFP and ***P = 0.0001 for αII-spectrin with Kolmogorov-Smirnov. **(d and f)** Primary mouse keratinocytes cultured in high-calcium (1.5 mM) medium treated with DMSO, gefitinib, or TGF-α and immunolabelled for p-EGFR, TRPV3, and αII-spectrin. Boxes indicate the location of the magnified area. **(e and g)** Quantification of TRPV3 and αII-spectrin intensity from the data shown in d and f. Mean ± SD from ~200 mature junctions from *n* = 3 experiment per condition. Bars: mean normalized intensity; dots: individual junctions. Nuclei were stained with DAPI. **P = 0.001 and ****P > 0.0001 for TRPV3 intensity. ****P > 0.0001 and ***P = 0.0001 for αII-spectrin intensity with Kolmogorov–Smirnov. **(h)** Dorsal skin sections from E17.5 wild-type embryos treated with DMSO and gefitinib. Sections were processed for transglutaminase 1 (TGM1) activity assay. **(i)** Quantification of crosslinked TGM substrate intensity. Mean ± SD of 30 ROIs from *n* = 3 embryos per condition. Bars: mean normalized intensity; dots: individual microscopy fields. NS: P = 0.3876 with Kolmogorov–Smirnov. **(j)** Quantification of cortical enrichment of crosslinked TGM substrate. Mean ± SD from 60 individual cells from *n* = 3 embryos per condition. Bars: means of cortex/cytoplasm ratio; dots: individual cells. ****P < 0.0001 with Kolmogorov–Smirnov. Nuclei were stained with DAPI; dashed lines indicate the dermal-epidermal border. **(k)** Model - High-tension spectrin-actomyosin cortices regulate TGM activity. Illustration of junction and cytoskeleton distribution across epidermal layers, with spectrin most enriched in SG3 and F-actin most enriched in SG1. Lower left: illustration of spectrin and myosin-dependent organization of cortical F-actin. Upper left: Working model of how lattice organization and myosin tension regulate EGFR/TRPV3 signaling complexes, resulting in Ca²⁺ influx and cortical TGM activation. ROI, region of interest.

shape transitions to regulate terminal differentiation, thus co-ordinating cell shape and fate across the tissue.

How spectrin regulates early differentiation when transitioning in the spinous layer is less clear. One potential candidate is YAP, based on previous studies that implicated spectrin and actomyosin tension in controlling YAP signaling in other epithelia (Deng et al., 2015; Fletcher et al., 2015). However, YAP nuclear localization was unchanged in *Sptan1*-depleted epidermis, suggesting that spectrin controls differentiation through a YAP-independent mechanism.

In the epidermis, mechanical regulation of EGFR contributes to barrier function by promoting terminal differentiation through TRPV3 and TGM1 (Cheng et al., 2010) and regulating TJs by reinforcing high-tension states in the SG2 layer (Rübsam et al., 2017), but whether these two functions are facets of the same mechanical regulation or regulated through distinct mechanisms remains unclear. Previous studies have shown that loss of myosin IIa/b in the epidermis impaired TJ barrier function but did not obviously affect stratum corneum barrier function (Sumigray et al., 2012), whereas loss of the Arp2/3 complex that enhances branched actin network assembly affects both barriers (Zhou et al., 2013). These observations further demonstrate the profound complexity of the actomyosin cyto-skeleton and the importance of its fine-tuning for the execution of distinct biological processes.

Our previous and current data indicate that layer-specific organization of E-cadherin–directed spectrin-actomyosin lattices also controls the activity and localization of EGFR, with active EGFR being internalized in lower layers, whereas in the granular layer, active EGFR is retained at the membrane. Either loss of spectrin or reducing actomyosin tension reverses this localization of active EGFR, with membrane localization now being confined to lower layers. The organization of the spectrin-actomyosin cortex can inhibit diffusion of plasma membrane receptors to induce molecular crowding, known as fencing (Fujiwara et al., 2016), as well as organize membrane trafficking events such as endocytosis (Ghisleni et al., 2020). We thus propose a model in which the spectrin-actomyosin organization in the granular layers limits the mobility of EGFR and TRPV3, while the high-tension state of this network also inhibits internalization of activated EGFR, thus bringing these proteins in close proximity to activate TRPV3 and, subsequently, TGM1 (Fig. 7 k). In agreement, both ligand-binding and actin-dependent

membrane organization regulate EGFR distribution at the plasma membrane (Sankaran et al., 2021; Bag et al., 2015). A potential alternative model involves a mechano-sensitive unfolding of spectrin (Leterrier and Pullarkat, 2022; Renn et al., 2019) or other, unknown players within the granular layer cortex that then interact with and cage the EGFR/TRPV3 to the cortex. Thus far, however, there is no evidence for tension-dependent binding partners that may explain EGFR or TRPV3 retention.

In conclusion, our data provide evidence for a model in which E-cadherin shapes the distinct organization and mechanics of the spectrin-actomyosin cortex in each layer to determine cell shape and differentiation status and control localization of pEGFR–TRPV3 complexes at the plasma membrane, resulting in TGM-dependent cross-linking only in the granular layer, thereby fixing cell shapes and promoting stratum corneum barrier function.

## Materials and methods

### Mice

To generate epidermal E-cadherin knockout (*Ecad*^epi−/−) BL/6N mice, female *Cdh1*^flox/flox mice were crossed to male *Ecad*^fl/wt;K14-Cre mice. Transgenic *Ecad*^flox/flox mice and K14-Cre mice and their genotyping were described previously (Hafner et al., 2004; Boussadia et al., 2002). Hsd:ICR (CD1) mice (Envigo) were used for all in vivo knockdown experiments. To generate epidermal αII-Spectrin knockout (*Sptan1*^epi−/−) mice, *Sptan1*^flox mice were ordered from The Jackson Laboratory (Strain ID- B6;129S-*Sptan1*^tm1.1Mnr/J, Strain #033392). Female *Sptan1*^flox/flox mice were crossed to male *Sptan1*^fl/wt;K14-Cre mice. *Sptan1*^flox mice were described previously (Huang et al., 2017). To generate epidermal confetti BL/6N mice, female R26R-Confetti^tg/tg mice were crossed with male K14-Cre mice for stochastic activation and recombination of the brainbow2.1 cassette in the epidermis, resulting in four possible outcomes: Expression of nuclear GFP, membrane-associated CFP, cytoplasmic YFP, or cytoplasmic RFP.

### Quantification of 3D epidermal cell shape parameter using epidermal confetti

Epidermal confetti newborn mice were sacrificed and fixed in 4% PFA overnight at 4°C. Backskin was then taken and mounted in Mowiol. As a basis for rendering the cytoplasmic YFP signal of the brainbow2.1, it was used due to homogenous intracellular

expression across epidermal layers and a good signal/noise ratio. Confocal stacks were acquired using a Leica TCS SP8 (PlanApo 63×, 1.4 NA CS2) in Leica lightning mode, pinhole 0.97 AU, voxel size 0.046 × 0.046 × 0.259, followed by the built-in lightning deconvolution. Imaris was then used for surface rendering using a surface detail of 0.2 µm and smoothing. Cell volume, surface area, and sphericity were extracted from the rendered cells in Imaris. Cell position was determined from XZ projections of the same stack using the measure tool in Fiji. Cell position was determined relative to the stratum corneum by measuring the distance of each cell to the upper side of the stratum corneum. The stratum corneum border was readily detectable due to preservation of the fluorescent signal and high cellularity.

### Isolation and culture of primary keratinocytes

Primary keratinocytes were isolated and cultured as described before (Rübsam et al., 2017): newborn mice were decapitated and incubated in 50% betaisodona/PBS for 30 min at 4°C, 1 min PBS, 1 min 70% EtOH, 1 min PBS, and 1 min antibiotic/antimycotic solution. Skin was incubated in 2 ml dispase (5 mg ml$^{-1}$ in culture medium) solution. After overnight incubation at 4°C, the skin was transferred onto 500 µl FAD medium on a 6-cm dish, and the epidermis was separated from the dermis as a sheet. Epidermis was transferred dermal side down onto 500 µl of TrypLE (Thermo Fisher Scientific) and incubated for 20 min at RT. Keratinocytes were washed out of the epidermal sheet using 3 ml of 10% FCS/PBS. After centrifugation, keratinocytes were resuspended in FAD medium and seeded onto collagen type-1– (0.04 mg ml$^{-1}$) (L7213; Biochrom) coated cell culture plates. Primary murine keratinocytes were kept at 32°C and 5% $CO_2$.

Primary keratinocytes were cultured in DMEM/HAM's F12 (FAD) medium with low $Ca^{2+}$ (50 µM) (Biochrom) supplemented with 10% FCS (chelated), penicillin (100 U ml$^{-1}$), streptomycin (100 µg ml$^{-1}$, A2212; Biochrom), adenine (1.8 × 10$^{-4}$ M, A3159; SIGMA), L-glutamine (2 mM, K0282; Biochrom), hydrocortisone (0.5 µg ml$^{-1}$, H4001; Sigma-Aldrich), EGF (10 ng ml$^{-1}$, E9644; Sigma-Aldrich), cholera enterotoxin (0.1 nM, C-8052; Sigma-Aldrich), insulin (5 µg ml$^{-1}$, I1882; Sigma-Aldrich), and ascorbic acid (0.05 mg ml$^{-1}$, A4034; Sigma-Aldrich).

### Transfection

Overexpression: Keratinocytes were transfected at 100% confluency with ViromerRED. 1.5 µg DNA was diluted in 100 µl Buffer E, added to 1.25 µl ViromerRED, and incubated for 15 min at RT (Rübsam et al., 2017). 33 µl transfection mix was used per well with 0.5 ml FAD medium (24-well plate). Knockdown: Keratinocytes were transfected at 100% confluency with ViromerBLUE (lipocalyx; Halle Germany). The transfection mix was prepared according to the manufacturer's protocol. 100 µl transfection mix was used per well with 1 ml FAD medium (12-well plate, 5 nM siRNA f.c.). siRNA (siPOOLs) against αII-Spectrin were obtained from siPOOLs from siTOOLs.

### Lentiviruses

Lentiviral plasmids were generated by cloning oligonucleotides into pLKO.1-TRC (gift from David Root, Broad Institute, Cambridge, MA, USA; #10878; Addgene plasmid), lentivirus-GFP, or lentivirus-RFP (gift from Elaine Fuchs, Rockefeller University, New York, NY, USA; #25999; Addgene plasmid) by digestion with EcoRI and AgeI, as described in the Genetic Perturbation Platform (GPP) website (http://portals.broadinstitute.org/gpp/public/resources/protocols). See also (Beronja et al., 2010; Soffer et al., 2022) for generation of lentiviruses. shRNA sequences were obtained from GPP (https://portals.broadinstitute.org/gpp/public/): *Sptan1(0595)* construct #TRCN0000090595, target sequence 5′-GCCACCGATGAAGCTTATAAA-3′. *Sptan1(9753)* construct #TRCN0000089753, target sequence 5′-CCAGTCACA ATCACCAATGTT-3′. Ank3(6780) construct #TRCN0000236780, target sequence 5′-TGCCGTGGTTTCCCGGATTAA-3′. The TRPV3-GFP construct was a gift from Michael X. Zhu (UTHealth Houston, Houston, Texas, USA).

### In utero lentivirus injection

Female mice at E8.5 were anesthetized with isoflurane, injected with the painkiller, Rheumocam Veterinary 5 mg/ml according to the manufacturer's instructions (Chanelle Pharma), and each embryo (up to six per litter) was injected with 0.4–1 µl of ~2 × 10$^9$ colony-forming units (CFUs) of the appropriate lentivirus. See also Beronja et al. (2010) for further details. Controls were both uninfected littermates of *shSptan1-0595/9753*;H2B-GFP/*shSptan1-0595*;H2B-RFP/*shAnk3-6780*;H2B-GFP lentivirus-injected embryos and *shScr*;H2B-GFP lentivirus-injected embryos. For *shSptan1-0595/9753; TRPV3-GFP* lentivirus-injected embryos, *shScr; TRPV3-GFP* were controls. In utero lentivirus injection infects ~60–70% of the dorsal skin epidermis (Beronja et al., 2010).

### Immunofluorescence of keratinocytes in vitro

Immunofluorescence stainings of keratinocytes were performed as described before (Sahu et al., 2020): cells were seeded on collagen-coated glass coverslips in a 24-well plate and switched to high $Ca^{2+}$ medium as indicated in the results section. Cells were fixed using 4% PFA for 10 min at RT, washed three times for 5 min using PBS, permeabilized using 0.5% Triton X-100/PBS, and blocked using 5% NGS/1% BSA/PBS for 1 h at RT. Primary antibodies were diluted as indicated in the antibody section in Background Reducing Antibody Diluent Solution (ADS) (DAKO). Coverslips were placed growth surface down onto a 50-µl drop of staining solution on parafilm in a humidified chamber and incubated overnight at 4°C. Coverslips were washed again with PBS three times for 10 min. Secondary antibodies and DAPI (Sigma-Aldrich) were diluted 1:500 in ADS, and coverslips were incubated for 1 h at RT. Secondary antibodies were washed off via three wash steps using PBS for 10 min. Coverslips were mounted using Mowiol (Calbiochem).

### Immunofluorescence on tissue sections

Embryos or newborn mice were embedded in OCT (scigen), frozen, sectioned at 10 µM, and fixed in 4% formaldehyde for 10 min. Sections were then blocked with 0.1% Triton X-100, 1% BSA, 5% normal donkey serum in PBS, or in MOM Basic kit reagent (Vector Laboratories). Sections were incubated with primary antibodies overnight at 4°C and with secondary antibodies for 1 h at RT.

## Antibodies and inhibitors

Ankyrin3 (IF 1:500, WB 1:1,000, #27766-1-AP; Proteintech), E-cadherin (IF 1:200, #610182; BD Transduction Laboratories, clone number 36 or IF 1:500, #3195; Cell Signaling), EGFR (IF 1:500, #ab52894; Abcam), pEGFR (Y1068) (IF 1:500, #ab40815; Abcam), GAPDH (WB 1:10,000, #AM4300; Ambion or WB 1:1,000, #5174; Cell Signaling), GFP (1:3,000, #ab13970; Abcam), Keratin10 (IF 1:1,000, #PRB-159P; BioLegend), Keratin14 (IF 1:2,000, #PRB 155P; Covance), Loricrin (1:1,000, #Poly19051; BioLegend), Myosin heavy chain IIa (IF 1:500, PRB-440; BioLegend), phospho-Myosin Light Chain 2 (Thr18/Ser19) (IF 1:100, #3674; Cell Signaling), occludin (IF 1:400, #33-1,500; Invitrogen), TRPV3 (IF 1:1,000, #b94582; Abcam), αII-Spectrin (IF 1:500, WB 1:1,000, ab11755; Abcam), TGM1 (IF 1:500, #12912-3-AP; Proteintech), α-catenin (IF 1:2,000, #C2081; Sigma-Aldrich), β-catenin (IF 1:1,000, #ab32572; Abcam), and YAP (IF 1:500, #14074; Cell Signaling). Phalloidin was used to stain F-actin (IF 1:500, #P1951; Sigma-Aldrich, TRITC conjugated). Secondary antibodies were species-specific antibodies conjugated with either Alexa Fluor 488, 594, or 647, used at a dilution of 1:500 for immunofluorescence (Molecular Probes, Life Technologies), or with horseradish peroxidase antibodies used at 1:5,000 for immunoblotting (Bio-Rad Laboratories).

Inhibitors used in this study, if not indicated otherwise: Blebbistatin myosin inhibitor, 5 μM (#B0560; Sigma-Aldrich); Latrunculin B, 0.1 μM (L5288; Sigma-Aldrich).

## Preparation of epidermal whole mounts

Epidermal whole mounts were prepared as has been described previously (Rübsam et al., 2017). Backskin was prepared from newborn mice, and subcutaneous fat was removed with curved tweezers. The epidermis was mechanically and carefully peeled off from the dermis with ultrafine curved tweezers, thereby separating the basal from the suprabasal layers. During the whole procedure, the basal side of the sheet was kept floating on PBS2+ (PBS supplemented with 0.5 mM MgCl2 and 0.1 mM CaCl2). Subsequently, the epidermal sheet was fixed, floating on 4% PFA on ice for 10 min, washed in PBS for 5 min, and permeabilized with 0.5% Triton X-100/PBS for 1 h at RT. The permeabilized sheet was washed for 5 min in PBS and blocked with 10% FCS/PBS for 30 min/RT. For staining, epidermal sheets were cut into ~5 × 5-mm pieces. The SC of the epidermis is prone to unspecific binding of ABs. Additionally, it cannot be permeabilized to allow AB permeation. Thus, all following steps were performed by incubating the sheet from the basal side, leaving the SC side dry. Primary ABs were diluted either in AB diluent solution (S3022; Dako) and incubated overnight at 4°C. Secondary Abs, including DAPI and phalloidin, were incubated for 2 h/RT. After each AB incubation, the sheet was rinsed three times with PBS and washed three times for 10 min. Finally, the stained sheet was mounted in 50 μl Mowiol.

## Isolation of single SG and SS cells

Dorsal back skin of newborn mice was excised (~10 mm × 10 mm), and fat tissue was removed with fine curved forceps. The epidermis was mechanically and carefully peeled off from the dermis with ultrafine curved tweezers, thereby separating

the basal from the suprabasal layers. Subsequently, the suprabasal epidermis was floated on a 400-μl droplet of 44 μg/ml rETA (*Staphylococcus aureus* Exfoliative Toxin A)/1 mM CaCl2 and incubated in the humidifying chamber at 37°C for 35 min. ETA diffuses intercellularly up to the tight junctions in the SG2 and cuts desmogleins-1, thereby loosening intercellular connections below the SG2 (Matsui et al., 2021). Layers below the SG2 layer (SS-SG3) were then peeled from the SG-1/2-corneum layers. Both parts were then floated onto 500-μl droplets of 0.05% trypsin/0.48 mM EDTA and incubated in the humidifying chamber at 37°C for 30 min. After the trypsinization, SS-SG3 cells were dissociated, and SG-1/2 cells were washed off from the stratum corneum by manual pipetting and transferred to a 1.5-ml Eppendorf tube. To neutralize, 1 ml of PBS++ was added to the trypsin solution containing SG and SS cells and centrifuged at 3,000 rpm for 5 min at RT. For treatment, cells were resuspended in DMEM (Gibco).

## Cytoskeletal inhibitor treatment on isolated SG and SS cells

After isolation, cells were treated with 1 ml DMEM with or without latrunculin B, 0.1 μM, shaking at 300 rpm for 1 h at 37°C. After the treatment, cells were washed with 1 ml of PBS++, centrifuged at 3,000 rpm for 5 min at RT, and fixed with 4% PFA for 10 min at RT. Cells were permeabilized using 0.5% Triton X-100/PBS for 10 min and washed with 500 μl of PBS++. Consecutively, cells were treated with phalloidin-TRITC and DAPI for 30 min at RT. After the incubation, cells were centrifuged and mounted on coverslips with 30 μl of Mowiol. Cells were then imaged, and the circumference of the cells was delineated using the freehand tool in Fiji. The area and circularity were extracted using the measure function in Fiji.

## Cytoskeletal inhibitor treatment on embryos

For in vivo latrunculin and Y27632 treatments, wild-type embryos or *shScr; TRPV3-GFP*–transduced embryos were collected on E17.5 with 2.5 μM latrunculin (Sigma-Aldrich) or with 40 μM Y27632 (Sigma-Aldrich) or DMSO in serum-free DMEM (Biological Industries) at 37°C for 1 h before embedding in OCT.

## TGM enzyme-substrate assay on cryosections

Cryosections were washed with PBS++ until the cryo tissue embedding compound Tissue-Tek O.C.T Compound (Sakura Finetek 4583) was removed. Then, sections were circled with a hydrophobic barrier marker pen (ReadyProbes - Thermo Fisher Scientific) and blocked with 10% NGS/PBS for 1 h at RT. Subsequently, sections were treated with and without TGM peptide substrate (pepK5: 5/6-FITC-Doa-YEQHKLPSSWPF-NH2/OH and K5pepQN: 5/6-FITC-Doa-YENHKLPSSWPF-NH2/OH, Intavis Peptide Services) for 30 min at RT in a humidified chamber and washed three times with PBS to remove the unspecific binding. Next, sections were fixed with 4% PFA for 5 min at RT and washed with PBS three times for 5 min. Sections were mounted using Mowiol.

## Toluidine blue barrier assay

E17.5 embryos were collected and immersed in an ice-cold methanol gradient in water, taking 2 min per step (1–25%, 2–50%, 3–75%, and 4–100% methanol), and then rehydrated using the

reverse procedure. Embryos were immersed in 0.2% toluidine blue solution. Embryos were washed in PBS before image capture. See also Hardman et al. (1998) for further details.

## Microscopy

Confocal images were obtained with a Leica TCS SP8, equipped with gateable hybrid detectors using LAS X software. Objectives used with this microscope: Leica PlanApo 63×, 1.4 NA CS2. Images to be used for deconvolution were obtained at optimal resolution according to Nyquist and deconvolved using Huygens Deconvolution. Alternatively, images were acquired with a Nikon C2+ laser-scanning confocal microscope using NIS Elements C software. Objectives: 60×, 1.4 NA oil or a 20×, 0.75 NA air objective (Nikon). Imaging was performed at RT, and samples were mounted in Mowiol.

## Laser ablation

Keratinocytes were seeded on glass-bottom dishes (# 81158; Ibidi) in FAD low $Ca^{2+}$ medium. At confluency, the medium was switched to a high $Ca^{2+}$ medium for 48 h. 2 h prior to imaging, the medium was changed to a high $Ca^{2+}$ medium containing 1 µM SiR-actin (#CY-SC001; Spirochrome). For laser ablation, a spinning Disk microscope (UltaView VoX, Perkin Elmer) equipped with a 355 nM pulsed YAG laser (Rapp Opto Electronic) and a Hamamatsu EMCCD C9100-50 camera was used with Volocity software. Experiments were performed at 37°C and 5% $CO_2$ using a water immersion objective 60×, 1.2 NA (Nikon). The apical area of apical stratified cells was ablated by drawing a line of consistent length with 30–50% laser power transmission and 10 iterations using UGA-40 Software (Biomedical Instruments).

## Transepithelial resistance measurement

A total of $5 \times 10^5$ keratinocytes were seeded on transwell filters (Corning [#3460], 0.4-µm pore size). Cells were allowed to settle and then switched to 1.8 mM high $Ca^{2+}$ medium. Formation of TER was measured over time using an automated cell monitoring system (cellZscope, nanoAnalytics).

## Semiquantitative RT-PCR

RNA was extracted from samples using a Direct-zol RNA extraction kit (R2060; Zymo Research), and equal amounts of RNA were reverse transcribed using the ProtoScript First Strand cDNA Synthesis Kit (New England Biolabs). Semiquantitative PCR was conducted using a StepOnePlus System (Thermo Fisher Scientific). Data are presented as mRNA levels of the gene of interest normalized to peptidylprolyl isomerase B (Ppib) mRNA levels. Primers used in this study: *Sptan1* forward 5′-TCGACAAGGACAAGTCTGGC-3′; *Sptan1* reverse 5′-AACAGGGCAAGCAGTGTAGG-3′; *Ank3* forward 5′-CTGACGTTCACGAGGGAGTT-3′; *Ank3* reverse 5′-TATCTA ACGTGTCCGCTGCC-3′; *PPIB* forward 5′-GTGAGCGCTTCCCAG ATGAGA-3′; *PPIB* reverse 5′-TGCCGGAGTCGACAATGATG-3′.

## Image quantification

### Quantification of F-actin, αII-spectrin, pEGFR, Ankyrin3, and TGM1 substrate levels in embryos

Intensities from layers as indicated in the figures and legends were measured using the freehand selection tool (Image J) with a width of five pixels. All intensity measurements were determined by corrected total cell F=florescence = integrated density-(area X mean of fluorescence of background readings). Intensity levels in each layer were normalized according to controls.

### Quantification of cell shapes in embryos and newborns

Embryos that were injected with lentiviruses encoding *shScr; H2B-GFP, shsptan1;H2B-GFP*, and *shAnk3-6780;H2B-GFP* on E8.5 and littermates were harvested at E17.5 frozen in OCT, sectioned (10 µm), fixed, and stained for E-cadherin (3195; Cell Signaling Technology, 1:500). For the analysis of Ctr, *Cdh1epi−/−*, and *Sptan1epi−/−* newborn epidermis, cryosections were PFA fixed and stained with a combination of Dsg1/2 antibody (61002; Progen, 1: 200) and Dsg3 antibody (D218-3; MBL, 1:2,000). Samples were imaged using confocal microscopy. The cell borders were delineated by the Freehand tool (Fiji) and cell area and perimeter using the "measure" function of Fiji. The cell shape index was calculated by dividing the perimeter by the square root of the area (Sahu et al., 2020).

### YAP+ cell quantification

The number of GFP+ and YAP+ cells was counted manually. The percentage of YAP+ cells was calculated as (number of YAP+GFP+ double-positive cells/total number of GFP+ cells) × 100 for the basal and suprabasal layers.

### Quantification of cortical enrichment

The mean gray value of the granular layer cells' cortical and cytoplasmic intensity levels were measured with the "wand tool" (ImageJ) with a width of five pixels. The ratio between the two means was calculated and referred to as the fold change.

### Statistics and repeatability of experiments

The number of independent experiments and biological replicates performed for all experiments, P values, and the statistical tests that were used are indicated in the figure legends.

### Online supplemental material

Fig. S1 provides extended information on 3D cell rendering using epidermal confetti mice, αII-spectrin expression in epidermal development, αII-spectrin knockdown and knockout efficiency, and epidermal cell shapes upon epidermis specific knockout of αII-spectrin, related to Fig. 1. Fig. S2 shows knockdown efficiency of Ankyrin-3, Ankyrin-3–dependent organization of F-actin and E-cadherin in the developing epidermis, the interdependency of αII-spectrin and E-cadherin in junctional recruitment in vitro, and extended information on latrunculin B treatment of isolated SG cells, related to Fig. 2. Fig. S3 shows actomyosin-dependent levels of E-cadherin in the developing epidermis, extended data on the spatial organization and expression levels of αII-spectrin in stratified keratinocytes in vitro, and the interdependency of cortical F-actin and αII-spectrin in vitro, related to Fig. 3. Fig. S4 provides extended information on the cortical interdependency of αII-spectrin and myosin IIa in vitro and laser ablation experiments showing force generation and dissipation of cortical F-actin networks in keratinocytes in vitro, related to Fig. 4. Fig. S5 shows extended

information on αII-spectrin–dependent regulation of epidermal differentiation, suprabasal proliferation and normal spindle orientation upon epidermal knockdown of αII-spectrin, reduced TGM1 activity in epidermal αII-spectrin knockouts, and impaired outside-in barrier function of αII-spectrin knockdown embryos, related to Fig. 5. Fig. S6 shows normal distribution of total EGFR in the epidermis upon αII-spectrin knockdown, extended information on αII-spectrin–dependent cortical localization of TRPV3 in the epidermis, and normal junctional E-cadherin localization upon low level interference with actomyosin in vitro, related to Fig. 6.

## Ethics declaration

All animal protocols in this study have been approved by either the animal experiment committee of LANUV, North Rhine-Westphalia, Germany, or University Animal Care and Use Committee, confirmation number TAU-MD-IL-2206-162-4. All methods were carried out in accordance with relevant guidelines and regulations. The animal protocols and the reporting in this manuscript follow the recommendations in the ARRIVE guidelines.

## Data availability

The authors declare that the data supporting the findings of this study are available within the paper and its supplementary information files. Additional data are available from the corresponding author upon reasonable request.

## Acknowledgments

We would like to thank the CECAD imaging facility under the supervision of Astrid Schauss.

This work is supported by the Deutsche Forschungsgemeinschaft, German Research Foundation: Germany's Excellence Strategy – EXC 2030/CECAD – 390661388 (to C.M. Niessen); CECAD project: CECAD Career-Promoting Grant for Senior Postdoctoral Scientists (to M. Rübsam); DFG-Schwerpunktprogramm (SPP) 1782 NI 1234/6-2, Project-ID 388932620 (to C.M. Niessen); DFG-Forschungsgruppe (FOR) 2743 NI 1234/7-1, Project-ID 678823 (to C.M. Niessen); ANR/DFG NI 1234/9, Project-ID 505673300 (to C.M. Niessen); European Union NETSKINMODELS, COST Action no. CA21108 (to C.M. Niessen); Israel Science Foundation (ISF) grant number 145/23 (to C. Luxenburg). This work was carried out in partial fulfilment of the requirements for a Ph.D. degree for AS. from the Faculty of Medical & Health Sciences, Tel-Aviv University Open Access funding provided by University of Cologne Medical Department; Department of Biology.

Author contributions: Arad Soffer: conceptualization, data curation, formal analysis, investigation, methodology, visualization, and writing—original draft, review, and editing. Aishwarya Bhosale: conceptualization, data curation, formal analysis, investigation, methodology, and project administration. Roohallah Ghodrat: formal analysis and investigation. Marc Peskoller: data curation, investigation, and project administration. Takeshi Matsui: methodology. Carien M. Niessen: conceptualization, funding acquisition, project administration, resources, supervision, validation, visualization, and writing—original draft, review, and editing. Chen Luxenburg: conceptualization, funding acquisition, project administration, supervision, and writing—original draft, review, and editing. Matthias Rübsam: conceptualization, data curation, formal analysis, funding acquisition, investigation, methodology, project administration, resources, supervision, validation, visualization, and writing—original draft, review, and editing.

Disclosures: The authors declare no competing interests exist.

Submitted: 11 February 2025

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

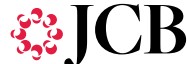

# Supplemental material

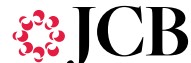

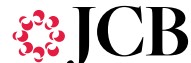

**Figure S1. αII-spectrin determines epidermal cell shape. (a)** From left to right: 3D whole mount of confetti epidermis from newborn mice showing expression of all four transgenic confetti colors, imaging of the cytoplasmic YFP signal, overlay of the rendering based on the YFP signal, and rendered cell volumes. **(b)** Quantification of cell sphericity from rendered cells per layer corresponding to Fig. 1 c. Dots: mean values per mouse. **(c)** Newborn epidermal whole-mount immunofluorescence analysis for phalloidin (F-actin), αII-spectrin, and tight junction marker occludin marking the SG2 layer. Overview of protein distribution across the layer corresponding to Fig. 1 c. Max. projections of the epidermal layers and a full projection (3D) are shown. **(d)** Immunofluorescence analysis for phalloidin (F-actin) and αII-spectrin on newborn epidermis cryosections of embryonic time points as indicated. Dashed line marks the epidermal-dermal boarder. **(c and d)** Representative images of N ≥ 3 biological replicates. **(e)** Quantitative qPCR analysis of *Sptan1* mRNA in primary mouse keratinocytes transduced with Ctr shRNA or one of two *Sptan1*-specific shRNAs (0595 and 9753). Mean ± SD of six preparations. ****P < 0.0001 by unpaired *t* test. **(f)** Western blot analysis of primary mouse keratinocytes transduced with *Scr*, *Sptan1 0595*, or *Sptan1 9753* shRNAs and quantification of αII-spectrin protein levels. Data are the mean ± SD of three preparations. ****P > 0.0001, ***P = 0.0006 by unpaired *t* test. **(g)** Dorsal skin mosaic tissue sections from *shSptan1 0595*-transduced E17.5 embryos immunolabeled for αII-spectrin. Line indicates areas of infected cells; dashed line indicates the dermal-epidermal border. Nuclei were stained with DAPI. Quantification of αII-spectrin intensity. Data are the mean ± SD of 60 individual cells from *n* = 3 embryos. Bars: mean normalized intensity; dots: individual cells. ****P > 0.0001 by unpaired *t* test. **(h)** Newborn epidermal whole-mount immunofluorescence analysis of αII-spectrin in Ctr and αII-spectrin–deficient epidermis (*Sptan1epi−/−*). Max. projection of the SG3 layer. **(i)** Western blot analysis of primary mouse keratinocytes isolated from Ctr and αII-spectrin–deficient epidermis (*Sptan1epi−/−*). **(j)** Immunofluorescence analysis for shape using combined staining for desmoglein1,2,3 (Dsg1,2,3) on Ctr and αII-spectrin–deficient newborn epidermis sections. **(k)** Quantification of cell sagittal area and shape (perimeter/√area)/layer using stainings as shown in j. ****P < 0.0001, **P < 0.005, and *P < 0.05; cells: *n* = 701 (basal), *n* = 927 (spinous), 688 (granular) with Kolmogorov–Smirnov per layer. Source data are available for this figure: SourceData FS1.

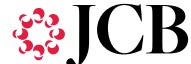

Figure S2. **E-cadherin controls cell shape upstream of spectrin. (a)** Western blot analysis of primary mouse keratinocytes transduced with shCtr and *Ank3* 6780 shRNAs. Blots were probed for Ankyrin-3 and GAPDH. **(b)** qPCR analysis of *Ank3* mRNA in primary mouse keratinocytes transduced with shCtr shRNA or Ank3-specific shRNAs (6780). Data are the mean ± SD of six preparations. **P = 0.002 with Kolmogorov–Smirnov. **(c)** Sagittal views of dorsal skin mosaic tissue sections from *shAnk3* 6780-transduced E17.5 embryos immunolabeled for Ankyrin-3. Graph: Quantification of Ankyrin-3 intensity. Mean ± SD of 60 individual cells from *n* = 3 embryos. Bars: mean normalized intensity; dots: individual cells. ****P > 0.0001 with Kolmogorov–Smirnov. **(d)** Sagittal views of dorsal skin sections from shCtr and *shAnk3* 6780-transduced E17.5 embryos immunolabeled for F-actin. Insets show transduced cells (H2B–GFP+). **(e)** Quantification of basal (left) and suprabasal (right) layer F-actin intensity from data shown in d. Data are the mean ± SD of 30 region of interest (ROI) from *n* = 3 embryos per condition. Horizontal bars represent the mean normalized intensity, and circles/squares represent microscopy fields. NS: P = 0.1344 (basal), NS: P = 0.9525 (suprabasal) with Kolmogorov–Smirnov. **(f)** Sagittal views of dorsal skin sections from shCtr and *shAnk3* 6780-transduced E17.5 embryos immunolabeled for E-cadherin. Insets show transduced cells (H2B–GFP+). **(g)** Quantification of basal (left) and suprabasal (right) layer cell cross-section area from data shown in f. Data are the mean ± SD from ∼200 individual cells from *n* = 3 embryos per condition. Horizontal bars represent the cell cross-section area mean, and circles/squares represent individual cells. NS: P = 0.2220 (basal), NS: P = 0.796 (suprabasal) by Kolmogorov–Smirnov. **(h)** Immunofluorescence analysis for the recruitment of E-cadherin and αII-spectrin to intercellular contacts in shCtr and *shEcad* 2287-transduced primary keratinocytes (GFP-positive nuclei, asterisks). Representative images of *n* = 3 biological replicates each. **(i)** Immunofluorescence analysis for E-cadherin (red) based on early intercellular contacts in shCtr and *shSptan1 0595*-transduced primary keratinocytes at the indicated Ca²⁺ time points. Representative images of *n* = 3 biological replicates each. **(j)** Phalloidin staining of single cells isolated from the granular layer treated with or without (nt) latrunculin B (0.1 μM). **(k)** Quantification of F-actin intensity (phalloidin) of isolated SG cells, as shown in j, treated with or without latrunculin B. Dots represent individual cells pooled from three mice. Source data are available for this figure: SourceData FS2.

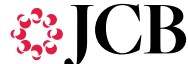

Figure S3. **Cortical F-actin and spectrin organization are mutually dependent. (a)** Sagittal views of dorsal skin sections from E17.5 wild-type embryos treated with DMSO, latrunculin, and Y27632 immunolabeled for E-cadherin. Nuclei were stained with DAPI; dotted lines indicate the dermal-epidermal border. NS, not significant. **(b)** Immunofluorescence analysis for αII-spectrin and F-actin after 6 or 48 h in high $Ca^{2+}$ at cell–cell interfaces and apical surface. Single channels corresponding to Fig. 3 g. **(c)** Western blot analysis and quantification for αII-spectrin protein levels after $Ca^{2+}$ switch for the time point indicated. Normalized to GAPDH and to 0 h $Ca^{2+}$ time point. Dots represent biological replicates from $n = 4$ independent primary keratinocyte isolates. **P = 0.003 with Kruskal–Wallis, Dunn's multiple comparison test. **(d)** Immunofluorescence analysis for E-cadherin, αII-spectrin, and F-actin after 48 h in high $Ca^{2+}$ at cell–cell interfaces. **(e)** Immunofluorescence analysis for αII-spectrin and F-actin after 48 h in high $Ca^{2+}$ at cell–cell interfaces with and without latrunculin B treatment (1 h, 0.1 µM). Single channels corresponding to Fig. 3 i. **(f)** Western blot analysis for αII-spectrin protein levels upon siRNA (siPOOLs)-mediated knockdown 96 h after transfection (72 h $Ca^{2+}$). **(g)** Western blot quantification for αII-spectrin as shown in f, normalized to GAPDH. Dots represent biological replicates, $n = 3$ with Mann–Whitney. Representative example of $n = 6$ independent primary keratinocyte isolates. **(h)** Immunofluorescence analysis of F-actin organization at apical junction rings after 48 h in high $Ca^{2+}$ and siRNA-mediated knockdown of αII-spectrin. **(i)** Quantification of F-actin intensity at apical junction rings (mean gray value, junctions/cytoplasm) as shown in h. Dots represent pooled values of single cells from $n = 3$ biological replicates. ****P < 0.0001 with Kolmogorov–Smirnov. Right graph: Mean values from $n = 3$ biological replicates tested with Mann–Whitney. Source data are available for this figure: SourceData FS3.

Figure S4. **Spectrin determines junctional actomyosin network structure and stability. (a)** Immunofluorescence analysis for αII-spectrin, F-actin, and non-muscle myosin heavy chain IIa (myosin-IIa, Myosin-9) (48 h high Ca²⁺) in Ctr and αII-spectrin–deficient (KO) cells. Single channels corresponding to Fig. 4 d. **(b)** Immunofluorescence analysis for αII-spectrin and F-actin (48 h high Ca²⁺) upon αII-spectrin knockdown and treatment with either DMSO or low-dose blebbistatin (5 μM). Representative images of *n* = 3 biological replicates each. Single channel gray scale and merged channels corresponding to Fig. 4 e. **(c)** Junctional laser ablation in apical keratinocytes with (48 h Ca²⁺) or without (40 h Ca²⁺) a cortical F-actin lattice. Time points after ablation (yellow line) are shown as indicated. **(d)** Quantification of the increase in vertex (dashed line) distance upon ablation. Lines: mean ± SD recoil of 11 ablations (no lattice) and 20 ablations (with lattice) from *n* = 3 biological replicates each. **(e)** Sequential ablation: short junctional ablation followed by a longer 17 μm ablation across the same junction. The latter one ablating the cortex connected to the ablated junction. **(f)** Quantification of vertex distance increase of sequential ablations. Line: mean ± SD recoil of seven sequential ablation from *n* = 2 biological replicates. **(g)** Cortical laser ablation showing straightening of curved cell–cell borders (yellow line) after linear ablation of the lattice.

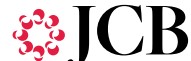

Figure S5. **αII-spectrin regulates epidermal differentiation and barrier function. (a)** Dorsal skin sections from *shSptan1 0595*-transduced E17.5 embryos immunolabeled for the basal layer marker K14 and the suprabasal marker K10. **(b)** Dorsal skin sections from *shSptan1 9753*-transduced E17.5 embryos immunolabeled for the basal layer marker K14 and the granular layer marker loricrin. Insets show the transduced cells (H2B–GFP+). **(c)** Dorsal skin sections from *shSptan1 0595*-transduced E17.5 embryos immunolabeled for the cleavage furrow marker survivin. Yellow lines show representative axes of division. Graph: Quantification of spindle orientation plotted as a cumulative frequency distribution. NS: P = 0.3485 with Mann–Whitney. **(d)** Dorsal skin sections from *shSptan1 0595*-transduced E17.5 embryos immunolabeled for EdU. **(e)** Quantification of EdU+ basal and suprabasal layer cells from the data shown in d. Bars: mean ± SD from *n* = 3 embryos per condition. Dots: average EdU+ basal and suprabasal layer cells from each embryo with Mann–Whitney. **(f)** Quantification of cell (nuclei) numbers from primary Ctr and *Sptan1*−/− or *Sptan1* siRNA-treated keratinocytes differentiated for 48 h in high Ca²⁺. Dots: Mean values from biological replicates. >360 cells counted for Ctr/*Sptan1*−/− each and >20,000 cells for siCtr/*siSptan1* each with Mann–Whitney. **(g)** Dorsal skin section from Ctr and *Sptan1*epi−/− newborn mice. Sections were processed for transglutaminase 1 (TGM1) activity assay or negative Ctr (mutated TGM substrate, pepQNK5). **(h)** Quantification of TGM1 intensity from data shown in g. Data are the mean of 30 fields of view from *n* = 3 newborn mice per condition. Bars: Mean intensity; dots individual microscopy fields. ****P < 0.0001 with Kolmogorov–Smirnov. **(i)** Dorsal skin sections from Ctr, *Ecad*epi−/− and *Sptan1*epi−/− newborn mice immunolabeled for total TGM1 protein. **(j and k)** Quantification of TGM1 intensity in Ctr, *Ecad*epi−/− and *Sptan1*epi−/−. Lines: Mean values/biological replicate. Nonsignificant with Mann–Whitney. **(l)** Dye exclusion assay: shCtr and *shSptan1 9753*-transduced E17.5 embryos were treated with toluidine blue dye to evaluate the skin barrier. **(m)** Dye exclusion assay: shCtr and *shSptan1 0595*-transduced E18.5 embryos were treated with toluidine blue.

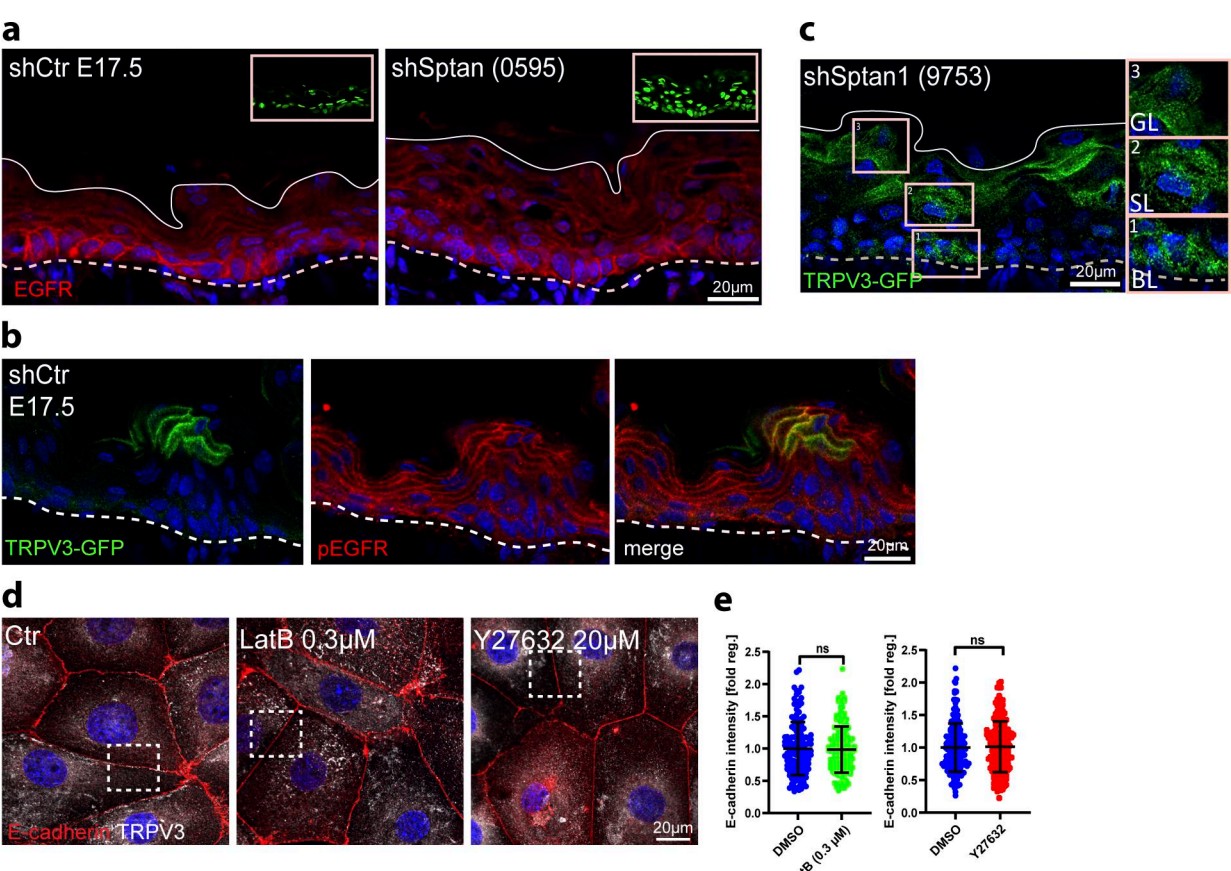

Figure S6. **αII-spectrin–actomyosin networks regulate the EGFR–TRPV3–TGM pathway. (a)** Dorsal skin sections from *shCtr* and *shSptan1 0595*-transduced E17.5 embryos immunolabeled for EGFR. Upper Insets show the transduced cells (H2B–GFP+). **(b)** Dorsal skin sections from shCtr; TRPV3-GFP–transduced E17.5 embryos immunolabeled for pEGFR. **(c)** Dorsal skin sections from *shSptan1 9753*; TRPV3-GFP–transduced E17.5 embryos. Insets show the magnification of the boxed area from each epidermal layer. **(d)** Primary mouse keratinocytes cultured in high-calcium (1.5 mM) medium treated with DMSO, latrunculin, or Y27632 and immunolabelled for E-cadherin and TRPV3. Overviews corresponding to Fig. 6 m. Boxes indicate the location of the magnified area. **(e)** Quantification of E-cadherin intensity. Mean ± SD from ~150 mature junctions from *n* = 3 experiment per condition with Kolmogorov–Smirnov. Bars: mean normalized intensity; dots: mature junctions. Nuclei were stained with DAPI.

