## [Peer Review File · The Journal of Cell Biology]

Spectrin coordinates cell shape and signaling essential for epidermal differentiation

Arad Soffer, Aishwarya Bhosale, Roohallah Ghodrat, Marc Peskoller, Takeshi Matsui, Carien Niessen, Chen Luxenburg, and Matthias Rübsam

Corresponding Author(s): Matthias Rübsam, University Hospital Cologne; Carien Niessen, University Hospital Cologne; and Chen Luxenburg, Tel Aviv University

Review Timeline:

Submission Date:	2025-02-11
Editorial Decision:	2025-03-04
Revision Received:	2025-10-10
Editorial Decision:	2025-11-14
Revision Received:	2025-12-16

Monitoring Editor: Ian Macara

Scientific Editor: Tim Spencer

Transaction Report:

DOI: <https://doi.org/10.1083/jcb.202502071>

Revision 0

Review #1

1. Evidence, reproducibility and clarity:

Evidence, reproducibility and clarity (Required)

This manuscript describes the in vivo impact of spectrin in the complex epithelium of the skin. Spectrin is a membrane-skeleton organizer that has been implicated in many aspects of cell biology, including cell-cell interactions via adherens junctions. Much of this work has been done in vitro and there is less precedent for testing its impact in tissues, in part because the diverse cell-biological functions of spectrin have made it difficult to directly attribute physiological function to cellular mechanism. This paper demonstrates that spectrin has a key role in skin homeostasis that can be linked to an impact on adherens junctions. The authors show that alphaII-spectrin depletion disturbs skin barrier function and differentiation. Spectrin concentrated at AJ and was recruited in response to both E-cadherin adhesion and cellular contractility, suggesting that it responds to mechanically-active AJ. In turn, spectrin was necessary to recruit EGFR and TRPV3, elements known to participate in epidermal differentiation.

****Specific comments:****

1. Relationship between spectrin and EGFR, TRPV signaling. These signals have been implicated in skin differentiation in other studies and the authors implicate them in the spectrin phenotype based on loss of signal and localization. I wonder if it is possible to obtain more direct functional data in their system. For example, would inhibiting EGFR signaling accentuate the spectrin-depletion phenotype. I appreciate that genetic tests may be beyond the reasonable scope of a revision.
2. Characterization of cell mechanics. I think that the kinetics of recoil are more informative as proxies of tension than single time points. Can they extract time-series from their movies? It would also be helpful to analyse the recoil against a model of viscoelasticity.

****Tiny points****

p16, top paragraph - some words seems to have been lost from the last sentence.

2. Significance:

Significance (Required)

Overall, this suggests a high-level pathway where spectrin mediates between cell-cell junctions and epidermal homeostasis. The report doesn't contain novel cellular mechanism, but it is a valuable contribution to the field with data of high quality. It provides in vivo data that will be a foundation for further developments in the field.

3. How much time do you estimate the authors will need to complete the suggested revisions:

Estimated time to Complete Revisions (Required)

(Decision Recommendation)

Between 3 and 6 months

4. Review Commons values the work of reviewers and encourages them to get credit for their work. Select 'Yes' below to register your reviewing activity at Web of Science Reviewer Recognition Service (formerly Publons); note that the content of your review will not be visible on Web of Science.

No

Review #2

1. Evidence, reproducibility and clarity:

Evidence, reproducibility and clarity (Required)

In this manuscript, Soffer, Bosale et al. examine the role of spectrin in differentiating mouse keratinocytes in vivo and in vitro with a focus on cell shape, actomyosin contractility, and barrier formation. They find that all-spectrin is present at higher levels in differentiating keratinocytes (spinous and granular layers) and is required for the cell flattening that accompanies terminal differentiation. While cortical recruitment of spectrin is independent of ankyrin G, it is disrupted by the loss of E-cadherin or inhibition of the actomyosin cytoskeleton. In turn, spectrin regulates the lattice-like organization of cortical actomyosin, and its deletion in in vitro induces the formation of contractile stress fibers. Finally, spectrin depletion alters expression patterns of canonical differentiation markers, prevents proper barrier formation, and decreases both EGFR phosphorylation and cortical enrichment of TRPV2. Overall, the authors conclude that all-spectrin acts downstream of E-cadherin to regulate actomyosin organization, cell shape, and epidermal differentiation.

****Major comments:****

It is difficult to compare the cell shape changes in spectrin knockdown embryos vs Ecad mutants (Figure 1g vs 1l. The graph axes/units are different and 1l also includes a "shape index" (not described in the Methods) and subdivides cells by layer position- please standardize. Currently it appears that the spectrin KD embryos show a more severe effect- can the authors comment on why this might be?

A key observation is that spectrin and F-actin co-localize in a lattice-like distribution in differentiating cells, and that loss of spectrin in vitro leads to the formation of contractile stress fibers. It is difficult to reconcile the in vitro data with the in vivo phenotype in Figure 2h. The authors should provide top-down images from spectrin KD embryos (from whole mounts or isolated cells) to show actin organization changes in vivo. Ideally, similar data should be presented for the Ecad mutant embryos to demonstrate that the spectrin-actin lattice is Ecad dependent.

There does not appear to be much Ank3 knockdown by immunofluorescence, although it is hard to tell (Figure S2c). Can the authors provide quantification for protein level (IF signal)? This is crucial to make the point that Ank3 is not important for spectrin localization.

The main data supporting the claim that spectrin KD embryos have a differentiation defect is the ectopic expression of K14 in suprabasal layers. From Figure 4a it looks like there are localized regions where K14+ cells accumulate, and adjacent regions where the layer organization is relatively normal. Please quantify how much of the epidermis (by length) is occupied by this type of ectopic K14 accumulation, and demonstrate with co-staining whether the K14+ suprabasal cells are also K10 (or Loricrin) positive.

The conclusion that spectrin promotes TGM by regulating the EGFR-TRPV3 pathway is not currently well supported. First, it is not clear to what extent mislocalization of TRPV3-GFP serves as a readout for endogenous TRPV3 activity. Second, there is no direct evidence supporting a cause-and-effect relationship. One suggestion would be to treat cultured KTCs with V3 agonists, which should stimulate TGM activity in wt cells but not in spectrin KD cells if TRPV3 activity is indeed affected. Alternately, does the overexpression of TRPV2-GFP lead to any rescue of TGM in the embryos?

Figure 1 starts out with some nice observations about the differences between contact dependence and spectrin/F-actin levels between upper (SG1/SG2) and lower (SG3/SS)

suprabasal layers. With the exception of Fig 1l, less care is taken throughout the rest of the manuscript to characterize layer specificity of spectrin LOF phenotypes or the spectrin-actin relationship. Can the authors clarify in the discussion how they interpret their results in the context of the different layers and their dependence/independence on cell contact?

****Minor comments:****

Some staining for TJ markers (ZO-1, occludin) in the spectrin KD embryos should ideally be provided along with the TEER measurements in Fig 4B to conclude that the liquid-liquid barrier is defective.

In some of the representative images (eg. Fig 2, S1e), it looks like the shSptan cells are much more abundant in the suprabasal layers than in the basal layer. Can the authors comment on whether these cells are more likely to get outcompeted/preferentially differentiate?

The model in Figure 6 could use more annotation- why is there more F-actin (or more AJs?) in the SG1 cells on the left vs the right?

Page 7- supplementary figure 1h is mislabelled, should be 1j; supplementary figure 1i should be 1k.

Page 16, final sentence of the first paragraph- typo?

The title for the final results section is "all-spectrin-actomyosin networks regulate the actomyosin skeleton"- is this correct?

2. Significance:

Significance (Required)

This manuscript focuses on an intriguing and understudied phenomenon: the coincidence of cell shape changes and progressive differentiation as keratinocytes move upward through the epidermal layers. Several cytoskeletal components have been identified that regulate shape but not differentiation, and vice versa, indicating that the two can at least to some extent be uncoupled- but in general the relationship is not well understood.

The main advance of this work is the identification of all-spectrin as a novel component that concurrently regulates cell shape, cytoskeletal organization, and barrier formation in

the mammalian epidermis. With the exception of Arp2/3, this is to my knowledge the only cytoskeletal regulator whose loss affects barrier function. The authors also show relationships, albeit direct or indirect, with other key molecular players (eg Ecad, EGFR, TGM), improving mechanistic understanding of the network controlling terminal cytoskeletal changes and epidermal differentiation. The integration between in vivo phenotypes and in vitro experiments (eg the laser ablations) is also a strength.

The main drawback of this study is that the insights are largely correlative in nature. Loss of spectrin clearly affects both cell shape and barrier function, but the cause and effect are not clear here and all-spectrin could simply be influencing each aspect via parallel pathways. Overall, a unified model for how spectrin regulates and is regulated by adhesion, the cytoskeleton, EGF signaling, and/or calcium channels is missing.

The audience for this work would be largely those with relatively specialized interest in tissue morphogenesis, cytoskeletal biology, and epidermal development.

Reviewer expertise: epidermal biology, differentiation, signaling, mouse models, stem cells

3. How much time do you estimate the authors will need to complete the suggested revisions:

Estimated time to Complete Revisions (Required)

(Decision Recommendation)

Between 1 and 3 months

No

Review #3

1. Evidence, reproducibility and clarity:

Evidence, reproducibility and clarity (Required)

****Summary:**** In this manuscript, Soffer et al. aim to elucidate the mechanisms driving cell shape during epidermal development and how those shape changes alter cell fates. The authors propose that spectrin integrates with E-cadherin and actomyosin networks to regulate cortical organization, mechanical properties, and differentiation. While the data presented are experimentally rigorous and propose several interesting models, several conclusions are overstated. Furthermore, while reading it, the organization seemed to jump between several ideas, and eventually felt like pieces of three separate stories. In addition, there are concerns with statistical analyses, and several aspects require clarification or more data. Below are specific comments and concerns:

****Major concerns:****

1. The suggestion that cell shape and cell fate are linked through spectrin is very exciting. But, as presented, the link is weak, preliminary, and speculative. The data presented regarding cell differentiation is really focused on cell function, which may or may not be caused by defects in differentiation.

- To test changes in cell fate, the authors only examine keratin expression and demonstrate that K14 expression is maintained in suprabasal cells of Sptan1-KD epidermis. To test whether cell identity is perturbed, a more comprehensive analysis should be performed. In particular, it would be beneficial to see K14 and K10 together, as both appear to have interesting expression patterns in the knockdown epidermis. P63 staining would also be useful to determine whether K14+ cells in suprabasal layers are still expressing multiple basal cell markers. The shapes in the K14+ suprabasal knockdown cells make it appear as though it is possible that basal cells are piling up on each other rather than suprabasal cells not flattening.

- For barrier function assays, the authors demonstrate decreased TEER in primary keratinocytes after calcium switch. Do tight junction proteins look perturbed in their localization and/or expression patterns in vitro and in vivo?

- For dye exclusion assays, the back seems to have formed an efficient barrier in the knockdown animal. Is this a true barrier defect or a developmental delay. If you look at E18.5, has the barrier on the ventral side of the animal formed more robustly? Do these animals survive perinatally? TEWL?

- TGM1 activity is down in shSptan1 and Ecadherin mutant epidermis. But what about TGM1 protein? Of course activity would be decreased if the cells weren't differentiating properly, as proposed by the authors.

2. The paper begins by demonstrating that SG2/1 cells maintain shapes when they are isolated. This is very clever experiment! However, there are no quantifications accompanying the images. In addition, it is difficult to say for sure, but the SS cells look like

their nuclei get smaller between in vivo and isolated, which may say more about the cellular viability than cell shapes.

3. In many of the quantifications throughout the manuscript, statistical analyses appear to be run on number of cells quantified rather than number of experimental animals. While I appreciate that there is a dynamic range within animals, using number of cells as the statistical n feels inappropriate, and is assigning very high statistical significance to what appear to be very subtle changes.

4. The data put a lot of emphasis on the micro-honeycomb lattice, but their significance is unclear. It does appear that they are spot junctions forming between the two layers of cells, but whether they contribute to cell shape, tension, etc, is very unclear.

5. The conclusion that ablation results demonstrate viscoelastic behavior (Supplementary Fig. 3i) is not adequately supported. The authors should provide quantitative analysis, such as recoil velocity or relaxation time, to distinguish elastic and viscous properties.

Additionally, the imaging duration (60 seconds) is insufficiently justified, and there is no correlation between the expansion direction and cortical behavior. Did the authors try to laser cut a cell junction and look at cortical tension differences?

****Minor concerns:****

6. The scale bars are mislabeled with " μM " instead of " μm " for micrometers, reflecting a lack of rigor in figure annotations. More broadly, figure labels are insufficiently detailed, detracting from clarity.

7. There are several typos and sentence fragments throughout the text.

8. Can the authors do the isolated cell shape experiments on spectrin KD epidermal cells? If it is important in cell shapes, while the starting shape is clearly different already, it may also have defects in maintaining that shape. Similar for Ecad KO epidermis.

9. Quantifications of Western blot and all-spectrin immunofluorescence in knockdown skin is needed.

10. The authors suggest that all-spectrin recruitment to the cell cortex requires myosin II activity. Do they have proposed mechanisms that could be added to the discussion? For example, is increasing contractility sufficient to increase all-spectrin localization? Is this through mechanosensitive molecules? Is spectrin thought to be mechanosensitive?

11. Are there any defects in proliferation/cell cycle in spastin KD cells in culture? Or cell spreading? It would be important to rule these out as possible contributors to decreased TEER - have they efficiently formed a confluent monolayer at a similar timing to controls?

2. Significance:

Significance (Required)

This paper combines in vivo and in vitro approaches to understand the role the spectrin cytoskeleton plays during epidermal development. In particular, it contributes to our understanding of how cells begin to flatten out as they differentiate into suprabasal cells. While the authors hint at the link between cell shape and cell fate/differentiation, it is more appropriate to say that they have linked cell shape to cell/tissue function, particularly barrier function.

This manuscript would be interesting and relevant to several audiences - while definitely more basic research, it does have implications for dermatology. More broadly, this study contributes to our understanding of how cell junctions and actin are regulated and organized.

3. How much time do you estimate the authors will need to complete the suggested revisions:

Estimated time to Complete Revisions (Required)

(Decision Recommendation)

Between 3 and 6 months

Yes

Revision Plan

Manuscript number: RC-2024-02746

Corresponding author(s): Carien M. Niessen, Chen Luxenburg, Matthias Rübsam

[The “revision plan” should delineate the revisions that authors intend to carry out in response to the points raised by the referees. It also provides the authors with the opportunity to explain their view of the paper and of the referee reports.]

The document is important for the editors of affiliate journals when they make a first decision on the transferred manuscript. It will also be useful to readers of the reprint and help them to obtain a balanced view of the paper.

*If you wish to submit a full revision, please use our "Full Revision" template. **It is important to use the appropriate template to clearly inform the editors of your intentions.**]*

1. General Statements [optional]

This section is optional. Insert here any general statements you wish to make about the goal of the study or about the reviews.

We would like to thank the reviewers for their thoughtful comments and the editor's guidance on strengthening our manuscript "Spectrin coordinates cortical actomyosin organization and differentiation essential for a functional barrier." We are happy to see that the reviewers find our data "of high quality" that advances "mechanistic understanding" and is of "interest to several audiences."

It is important to note that the main finding of this manuscript is that spectrin spatially coordinates actomyosin-dependent cell shape changes and differentiation in epithelium in vivo. Deletion of the cytoskeletal protein spectrin in the epidermis results in profound changes in actomyosin organization, cell shape, and mechanics, resulting in disturbed epithelial differentiation and barrier function.

Nearly 170 years ago, Rudolf Virchow highlighted the critical role of cell shape in cellular function and disease. However, the mechanisms connecting cell shape, differentiation, and function remain poorly understood. Our study thus provides a new mechanism of actomyosin-dependent fate regulation in an environment determined by cell neighbors instead of the already well-described changes induced by bimodal interfaces where cells face either matrix or lumen, with important implications for, e.g., tumor environments.

Notably, this study provides an important missing link in our knowledge of how adhesive junctions spatially coordinate cellular actomyosin tension states and receptor tyrosine kinase signaling to control differentiation and epithelial barrier function. Our understanding of the role of spectrin in epithelia is only beginning to emerge, and this work provides a thorough and mechanistic understanding of the in vivo role of spectrin in epithelium with implications far beyond the skin.

Revision Plan

In the revision, we will comprehensively address the reviewer's comments, and we are pleased to report that we have already initiated the necessary experimental work. Specifically, we plan to:

1. Extend the in vivo characterization of α -spectrin KD using IF-based fate markers and quantifications of cell shape and position to strengthen the evidence that spectrin-coordinated changes in cell shape regulate early epidermal differentiation and proliferation.
2. To improve the mechanistic link of how spectrin-actomyosin organization and cell shape regulates EGFR/TRPV3-mediated Ca^{2+} influx and keratinocyte differentiation, we will use a combination of EGFR, TRPV3, and actomyosin agonists and antagonists together with IF-based readouts for intracellular calcium, protein localization, and cell differentiation.
3. Perform in vivo 3D analysis of the spectrin-actomyosin lattice and tight junctional network in embryos and newborns to demonstrate the relevance of our in vitro findings on how spectrin-dependent cortical F-actin lattice configuration regulates barrier function.
4. Strengthen our laser ablation studies by quantifying recoil over time to characterize more how the loss of spectrin alters the visco-elastic behavior of the actomyosin cortex in suprabasal cells.

We are confident that these additions fully address the reviewer's comments and substantially improve our study's clarity, depth, mechanistic insights, and overall impact.

2. Description of the planned revisions

Insert here a point-by-point reply that explains what revisions, additional experimentations and analyses are planned to address the points raised by the referees.

Reviewer #1 (Evidence, reproducibility and clarity (Required)):

This manuscript describes the in vivo impact of spectrin in the complex epithelium of the skin. Spectrin is a membrane-skeleton organizer that has been implicated in many aspects of cell biology, including cell-cell interactions via adherens junctions. Much of this work has been done in vitro and there is less precedent for testing its impact in tissues, in part because the diverse cell-biological functions of spectrin have made it difficult to directly attribute physiological function to cellular mechanism. This paper demonstrates that spectrin has a key role in skin homeostasis that can be linked to an impact on adherens junctions. The authors show that α -spectrin depletion disturbs skin barrier function and differentiation. Spectrin concentrated at AJ and was recruited in response to both E-cadherin adhesion and cellular contractility, suggesting that it responds to mechanically-active AJ. In turn, spectrin was necessary to recruit EGFR and TRPV3, elements known to participate in epidermal differentiation.

We thank the reviewer for recognizing the importance and impact of addressing the in vivo function of spectrin in epithelia and more specifically its role in regulation epidermal differentiation and barrier formation.

Revision Plan

Specific comments:

1. Relationship between spectrin and EGFR, TRPV signaling. These signals have been implicated in skin differentiation in other studies and the authors implicate them in the spectrin phenotype based on loss of signal and localization. I wonder if it is possible to obtain more direct functional data in their system. For example, would inhibiting EGFR signaling accentuate the spectrin-depletion phenotype. I appreciate that genetic tests may be beyond the reasonable scope of a revision.

We thank the reviewer for this important comment. For clarity, although EGFR and TRPV signaling have been linked to skin differentiation and barrier function in other studies (e.g., Cheng et al. Cell 2010), these studies did not explore the role of the cytoskeleton and adhesion in regulating EGFR-dependent activation of TRPV3 nor has spectrin itself been implicated in epithelial barrier function.

To explore further and address how spectrin regulates EGFR and TRPV3 and examine interactions between these three proteins, we will employ a combination of in vivo and in vitro approaches: Our data thus far show that either loss of spectrin, or inhibition of actin organization or actomyosin contractility disturbs EGFR and TRPV3 localization and alters TGM activity. In the revised manuscript, we will also manipulate EGFR activity in vivo in the presence and absence of spectrin and examine how this impacts spectrin organization, TRPV3 localization, and TGM activity where possible.

As the reviewer noted, genetic manipulation in vivo has limitations. To overcome this, we will use our in vitro primary mouse keratinocyte cell culture system to characterize the spectrin-EGFR-TRPV interactions further. Our preliminary results indicate that key aspects of the in vivo system can be recapitulated in this system, including AJ-dependent cortical spectrin-actomyosin organization and TRPV3 recruitment to the plasma membrane (see Figure below). Therefore, in addition to manipulating spectrin, actin, or myosin, we will also inhibit and activate EGFR and TRPV and examine outcomes on Ca^{2+} signaling using a Ca^{2+} sensor (GCaMP) in combination with IF and/or RT-PCR analysis for differentiation markers. Together, these experiments will provide a comprehensive understanding of the spectrin-actomyosin cytoskeleton-EGFR-TRPV pathway and how it promotes differentiation.

Endogenous TRPV3 localization requires spectrin and actomyosin activity (A) *ShScr* (Ctrl) and *shSptan1-0595* transduced primary mouse keratinocytes (H2B-GFP+) were cultured in high-calcium (1.5 mM) media and then immunolabelled for E-cadherin (Red) and TRPV3 (Gray). Insets (Lower right) show magnification of the areas pointed by arrows. **(B)** Quantification of E-cadherin and TRPV3 intensity from data shown in A. Data are the mean \pm SD from ~200 mature junctions from n=3 experiment per condition. Horizontal bars represent the mean normalized intensity and circles represent mature junctions. Asterisks represent statistical significance. P=0.0112 for *ctrl* versus *shSptan1 0595* and not significant (P=0.4838) for *ctrl* versus *shSptan1 9753* by unpaired t-test for E-cadherin intensity. P<0.0001 for *ctrl* versus *shSptan1 0595* and P<0.0001 for *ctrl* versus *shSptan1 9753* by unpaired t-test for TRPV3 intensity. **(C)** Wild-type primary mouse keratinocytes cultured in high-calcium (1.5 mM) media treated with DMSO and latrunculin and then immunolabelled for E-cadherin (Red) and TRPV3 (Gray). Insets (Lower right) show magnification of the areas pointed by arrows. **(D)** Quantification of E-cadherin and TRPV3 intensity from data shown in C. Data are the mean \pm SD from ~150 mature junctions from n=3 experiment per condition. Horizontal bars

Revision Plan

represent the mean normalized intensity and circles represent mature junctions. Not significant ($P=0.7232$) for DMSO versus latrunculin by unpaired t-test for E-cadherin intensity. $P<0.0001$ for DMSO versus latrunculin by unpaired t-test for TRPV3 intensity. **(E)** Wild-type primary mouse keratinocytes cultured in high-calcium (1.5 mM) media treated with DMSO and Y27632 and then immunolabelled for E-cadherin (Red) and TRPV3 (Gray). Insets (Lower right) show magnification of the areas pointed by arrows. **(F)** Quantification of E-cadherin and TRPV3 intensity from data shown in E. Data are the mean \pm SD from ~ 200 mature junctions from $n=3$ experiment per condition. Horizontal bars represent the mean normalized intensity and circles represent mature junctions. Not significant ($P=0.74$) for DMSO versus Y27632 by unpaired t-test for E-cadherin intensity. $P<0.0001$ for DMSO versus Y27632 by unpaired t-test for TRPV3 intensity. Nuclei were stained with DAPI. Scale bars = 20 μm .

2. Characterization of cell mechanics. I think that the kinetics of recoil are more informative as proxies of tension than single time points. Can they extract time-series from their movies? It would also be helpful to analyse the recoil against a model of viscoelasticity.

We agree with the reviewer that the kinetics of recoil may provide more insight. Therefore, we have already started to re-analyze our laser ablation sequences to describe recoil behavior and kinetics more thoroughly and assess if and ensure this behavior aligns with a visco-elasticity model (see data below). See also answer to a related point of reviewer 3. Our preliminary analysis of the recoil dynamics appears to be in line with a viscoelastic cortical behavior.

Cortical laser ablation. Live imaging of SiR-actin (F-actin) labeled stratified Ctr keratinocytes after 48h high Ca^{2+} . **a** Junctional ablation in apical keratinocytes with or without a cortical F-actin lattice. Timepoints after ablation (yellow line) are shown as indicated. **b** Quantification of the increase in vertex (dashed line) distance upon ablation. Lines: mean \pm SD recoil of 11 ablations (no lattice) and 20 ablations (with lattice) from $n=3$ biological replicates each. **c** sequential ablation: short junctional ablation followed by a longer 17 μm ablation across the same junction. The latter one ablating the cortex connected to the ablated

Revision Plan

junction. **d** Quantification of vertex distance increase of sequential ablations as shown in **c**. Line: mean +/- SD recoil of 7 sequential ablation from N=2 biological replicates. **e** Cortical laser ablation of a 17 μ m line and **f** quantification of the elliptical opening area (red line) over time. Line: mean +/-SD opening area of 13 ablations from N=4 biological replicates.

Tiny points

p16, top paragraph - some words seems to have been lost from the last sentence.

Thank you for pointing this out. In fact the “initial E-cadherin engagement” was duplicated by accident from the previous sentence and has been removed.

Reviewer #1 (Significance (Required)):

Overall, this suggests a high-level pathway where spectrin mediates between cell-cell junctions and epidermal homeostasis. The report doesn't contain novel cellular mechanism, but it is a valuable contribution to the field with data of high quality. It provides in vivo data that will be a foundation for further developments in the field.

We appreciate the reviewer's positive feedback, even though we respectfully disagree with the assertion that our manuscript lacks a 'novel cellular mechanism.' While EGFR and TRPV3 have been previously implicated in regulating epidermal terminal differentiation (Cheng et al. Cell 2010)— and our own work has shown that adherens junctions coordinate tension and EGFR signaling to position the tight junctional barrier in the upper suprabasal layers (Rübsam et al., Nat Commun 2017)— the role of spectrin in these pathways was entirely unknown. Our study offers a new regulatory perspective by, for the first time, manipulating spectrin and ankyrin in the developing epidermis.

Moreover, while the role of actomyosin-dependent cell mechanics is well established in regulating basal-to-spinous fate transitions, much less is known about its role in suprabasal fate changes and terminal differentiation. In this manuscript, we identify spectrin as a key downstream component of adherens junctions that governs actomyosin cortex organization and mechanics, linking this to TGM1 activation in the stratum granulosum. Thus, our study provides novel insight into how fundamental cortical actomyosin mechanics drive cell shape changes and how this process is integrated into terminal cell fate regulation via the EGFR-TRPV3-calcium pathway.

Our data further demonstrate that, in addition to differentiation regulated by changes in gene expression, layer-specific changes in structure and mechanics of the cell cortex serves as an additional regulatory layer necessary to spatially activate enzymes whose expression is induced by changes in gene expression (“genetic differentiation”). A recent study from the Lechler lab (Prado-Mantilla et al., *eLife* preprint, 2024) also showed that inducing contractility is sufficient to

Revision Plan

drive granular cell fate, even within the spinous layer. Our study expands on this study and now provides mechanistic insight into how changes in actomyosin contractility control granular layer cell fate by revealing that spectrin controls layer-dependent actomyosin organization to enable EGFR-TRPV3-mediated activation of TGM in the granular layer necessary for cortex differentiation, which is independent of gene regulation.

Reviewer #2 (Evidence, reproducibility and clarity (Required)):

In this manuscript, Soffer, Bosale et al. examine the role of spectrin in differentiating mouse keratinocytes in vivo and in vitro with a focus on cell shape, actomyosin contractility, and barrier formation. They find that all-spectrin is present at higher levels in differentiating keratinocytes (spinous and granular layers) and is required for the cell flattening that accompanies terminal differentiation. While cortical recruitment of spectrin is independent of ankyrin G, it is disrupted by the loss of E-cadherin or inhibition of the actomyosin cytoskeleton. In turn, spectrin regulates the lattice-like organization of cortical actomyosin, and its deletion in in vitro induces the formation of contractile stress fibers. Finally, spectrin depletion alters expression patterns of canonical differentiation markers, prevents proper barrier formation, and decreases both EGFR phosphorylation and cortical enrichment of TRPV2. Overall, the authors conclude that all-spectrin acts downstream of E-cadherin to regulate actomyosin organization, cell shape, and epidermal differentiation.

Major comments:

It is difficult to compare the cell shape changes in spectrin knockdown embryos vs Ecad mutants (Figure 1g vs 1l. The graph axes/units are different and 1l also includes a "shape index" (not described in the Methods) and subdivides cells by layer position- please standardize. Currently it appears that the spectrin KD embryos show a more severe effect- can the authors comment on why this might be?

We appreciate the reviewer's suggestion and will ensure in the future revised version that the same cell shape measurements are used consistently throughout the manuscript and are properly described in the material and methods. We will also carefully examine and compare the phenotypes to assess how the phenotype of spectrin loss compares to that of E-cadherin. It is important to point out that whereas E-cadherin is genetically inactivated in all epidermal cells, spectrin is knocked down but not in all cells, making it potentially difficult to directly compare severity of phenotypes. As a small reminder, genetic loss of E-cadherin in the epidermis results in transepidermal water loss and perinatal death (Tunggal et al., 2005, Rübsam et al. 2017).

While we are still collecting the data, we can already highlight one key difference: whereas spectrin promotes cortical F-actin recruitment in all cortical layers, E-cadherin inhibits cortical actin recruitment in the spinous layer but not in the granular layer (Rübsam et al., Nat. Commun., 2017). Several factors may contribute to this difference. First, E-cadherin loss does not eliminate all classical cadherins, allowing spectrin recruitment through alternative cadherins such as P-

Revision Plan

cadherin in the basal layer. Additionally, as demonstrated by our work and others E-cadherin directly regulates actomyosin recruitment independent of spectrin, and, as mentioned, E-cadherin differentially regulates cortical F-actin organization across the different suprabasal layers. Moreover, E-cadherin also influences actin-modifying proteins and signaling pathways, including Rho kinases, as shown in multiple studies. Together, this may explain why the effects of E-cadherin and spectrin loss only partially overlap, and loss of E-cadherin beyond its role in spectrin recruitment, may lead to additional independent consequences.

A key observation is that spectrin and F-actin co-localize in a lattice-like distribution in differentiating cells, and that loss of spectrin in vitro leads to the formation of contractile stress fibers. It is difficult to reconcile the in vitro data with the in vivo phenotype in Figure 2h. The authors should provide top-down images from spectrin KD embryos (from whole mounts or isolated cells) to show actin organization changes in vivo. Ideally, similar data should be presented for the Ecad mutant embryos to demonstrate that the spectrin-actin lattice is Ecad dependent.

The reviewer brings up an important point. To address this, we will perform 3D epidermal whole-mount analyses of the spectrin-F-actin lattice in spectrin KD and E-cadherin mutant embryos.

There does not appear to be much Ank3 knockdown by immunofluorescence, although it is hard to tell (Figure S2c). Can the authors provide quantification for protein level (IF signal)? This is crucial to make the point that Ank3 is not important for spectrin localization.

As the reviewer suggests, in the revised manuscript, we will include quantification of immunofluorescence for ANK3, along with Western blot and qPCR analyses.

The main data supporting the claim that spectrin KD embryos have a differentiation defect is the ectopic expression of K14 in suprabasal layers. From Figure 4a it looks like there are localized regions where K14+ cells accumulate, and adjacent regions where the layer organization is relatively normal. Please quantify how much of the epidermis (by length) is occupied by this type of ectopic K14 accumulation, and demonstrate with co-staining whether the K14+ suprabasal cells are also K10 (or Loricrin) positive

As the reviewer suggests, we will quantify K14/K10 double-positive cells. Additionally, to further support our claim, we have already analyzed EdU incorporation. Consistent with the ectopic suprabasal expression of K14, our new data show increased suprabasal proliferation, as indicated by EdU incorporation in spectrin KD epidermis (see Figure below). These findings demonstrate that spectrin loss disrupts early epidermal differentiation, specifically the basal-to-suprabasal transition. In the revised manuscript we will also better highlight this important point. In addition, we will analyse how the inability to properly induce differentiation is linked to cell shape by examining whether suprabasal K14-positive cells also are smaller in size.

all-spectrin depletion enhances suprabasal proliferation. (A) Sagittal views of 10- μ m sections of dorsal skin from *shScr* (Ctrl) and *shSptan1 0595* transduced E17.5 embryos. Sections were labeled for EdU. arrows show suprabasal EdU+ cells. Upper Insets show the transduced cells (H2B-GFP+). **(B)** Quantification of EdU+ basal and suprabasal layers cells from the data shown in A. Horizontal bars represent the mean \pm SD from n=3 embryos per condition. Circles represent the average of EdU+ basal and suprabasal layers cells from each embryo. Not significant (P=0.1104) for *ctrl* versus *shSptan1 0595* and not significant (P=0.01097) for *ctrl* versus *shSptan1 9753* by unpaired t-test for EdU+ basal layer cells. P=0.0028 for *ctrl* versus *shSptan1 0595* and P=0.0489 for *ctrl* versus *shSptan1 9753* by unpaired t-test for EdU+ suprabasal layer cells. Nuclei were stained with DAPI, and dotted lines indicate the dermal-epidermal border. Scale bars = 20 μ m. ns, not significant.

The conclusion that spectrin promotes TGM by regulating the EGFR-TRPV3 pathway is not currently well supported. First, it is not clear to what extent mislocalization of TRPV3-GFP serves as a readout for endogenous TRPV3 activity.

While the reviewer is correct that altered TRPV3 localization is an indirect measure of TRPV3 activity, this approach has been widely used in the field as a proxy for activity (Li et al., 2016; Lei et al., 2023). We tested multiple antibodies to assess whether spectrin loss affects endogenous TRPV3 localization, but unfortunately, none worked in vivo. Therefore, we utilized TRPV3-GFP as a reporter to examine its localization, as done in previous studies (e.g., Xiao et al., 2008; Liu et al., 2021).

We have now identified a TRPV3 antibody that functions in cultured keratinocytes, though not in vivo. Using this TRPV3 antibody, we observed that endogenous TRPV3 localizes to cell junctions (cell periphery). Notably, spectrin knockdown or pharmacological disruption of actomyosin severely impaired TRPV3 localization at cell-cell contacts, mirroring the TRPV3-GFP pattern observed in spectrin knockdown embryos. These findings validate TRPV3-GFP as a reliable reporter for TRPV3 localization and further support our in vivo observations (see data below).

Endogenous TRPV3 localization requires spectrin and actomyosin activity (A) *ShScr* (Ctrl) and *shSptan1-0595* transduced primary mouse keratinocytes (H2B-GFP+) were cultured in high-calcium (1.5 mM) media and then immunolabelled for E-cadherin (Red) and TRPV3 (Gray). Insets (Lower right) show magnification of the areas pointed by arrows. **(B)** Quantification of E-cadherin and TRPV3 intensity from data shown in A. Data are the mean \pm SD from ~200 mature junctions from n=3 experiment per condition. Horizontal bars represent the mean normalized intensity and circles represent mature junctions. Asterisks represent statistical significance. P=0.0112 for *ctrl* versus *shSptan1 0595* and not significant (P=0.4838) for *ctrl* versus *shSptan1 9753* by unpaired t-test for E-cadherin intensity. P<0.0001 for *ctrl* versus *shSptan1 0595* and P<0.0001 for *ctrl* versus *shSptan1 9753* by unpaired t-test for TRPV3 intensity. **(C)** Wild-type primary mouse keratinocytes cultured in high-calcium (1.5 mM) media treated with DMSO and latrunculin and then immunolabelled for E-cadherin (Red) and TRPV3 (Gray). Insets (Lower right) show magnification of the areas pointed by arrows. **(D)** Quantification of E-cadherin and TRPV3 intensity from data shown in C. Data are the mean \pm SD from ~150 mature junctions from n=3 experiment per condition. Horizontal bars

Revision Plan

represent the mean normalized intensity and circles represent mature junctions. Not significant ($P=0.7232$) for DMSO versus latrunculin by unpaired t-test for E-cadherin intensity. $P<0.0001$ for DMSO versus latrunculin by unpaired t-test for TRPV3 intensity. **(E)** Wild-type primary mouse keratinocytes cultured in high-calcium (1.5 mM) media treated with DMSO and Y27632 and then immunolabelled for E-cadherin (Red) and TRPV3 (Gray). Insets (Lower right) show magnification of the areas pointed by arrows. **(F)** Quantification of E-cadherin and TRPV3 intensity from data shown in E. Data are the mean \pm SD from ~200 mature junctions from $n=3$ experiment per condition. Horizontal bars represent the mean normalized intensity and circles represent mature junctions. Not significant ($P=0.74$) for DMSO versus Y27632 by unpaired t-test for E-cadherin intensity. $P<0.0001$ for DMSO versus Y27632 by unpaired t-test for TRPV3 intensity. Nuclei were stained with DAPI. Scale bars = 20 μm .

To directly assess changes in TRPV3 activity, we will utilize keratinocytes expressing the Ca^{2+} sensor GCaMP to monitor intracellular Ca^{2+} influx following spectrin loss, allowing us to correlate TRPV3 localization with function in our in vitro system. In more detail, we will also explore the relation between spectrin loss, actomyosin activity, EGFR, TRPV3, and terminal differentiation, as commented in detail to reviewer 1.

Second, there is no direct evidence supporting a cause-and-effect relationship. One suggestion would be to treat cultured KTCs with V3 agonists, which should stimulate TGM activity in wt cells but not in spectrin KD cells if TRPV3 activity is indeed affected.

We agree with the reviewer and are already using the aforementioned in vitro system to further investigate the spectrin-EGFR-TRPV pathway. Our preliminary findings in spectrin KD cells, as well as experiments using actomyosin inhibitors, show promising results. We are now expanding these studies by incorporating EGFR and TRPV inhibitors/activators.

However, in our hands—and as confirmed through personal communication with other laboratories—TGM expression is not reliably induced upon differentiation in cultured mouse keratinocytes. To address this, we are exploring alternative differentiation readouts to gain functional insights into how spectrin-mediated regulation of TRPV3 activity influences differentiation.

Alternately, does the overexpression of TRPV2-GFP lead to any rescue of TGM in the embryos?

Unfortunately, we cannot reliably quantify TGM activity since we can only generate small clones of *shScr; TRPV-GFP / shSptan1; TRPV-GFP* transduced epidermis due to the large size of the insert (*TRPV3-GFP*) and the low titer that it allows. Nevertheless, our results show that upon overexpression of TRPV3-GFP (*shScr; TRPV-GFP*) the epidermis exhibits the typical flat morphology in the granular layer while epidermis that combines spectrin KD and TRPV3-GFP overexpression (*shSptan1; TRPV-GFP*) show a profound defect in cell compaction, similar to spectrin KD cells, suggesting that overexpression of TRPV3-GFP is not sufficient to rescue either cell shape changes, and indirectly terminal differentiation and barrier function. In the revised version we will add K14/K10 and filaggrin staining to further support this idea.

Revision Plan

This observation aligns with our observations and model in which spectrin is crucial for organizing the cortical cytoskeleton and maintaining proper cell shape, which in turn spatially regulates EGFR activity at the membrane to determine TRPV3 localization and activation.

Figure 1 starts out with some nice observations about the differences between contact dependence and spectrin/F-actin levels between upper (SG1/SG2) and lower (SG3/SS) suprabasal layers. With the exception of Fig 1I, less care is taken throughout the rest of the manuscript to characterize layer specificity of spectrin LOF phenotypes or the spectrin-actin relationship. Can the authors clarify in the discussion how they interpret their results in the context of the different layers and their dependence/independence on cell contact?

We thank the reviewer for pointing out this lack of clarity. Our data indicate that in our spectrin mutants both spinous and granular layer cells show changes in shape, but we did not carefully distinguish between these different suprabasal layers throughout all experiments manuscript, and will do so in the revised manuscript. We will also integrate this new data to then carefully address/discuss the role of junction-dependent and independent roles spectrin in organizing actomyosin and cell shape in different layers.

Minor comments:

Some staining for TJ markers (ZO-1, occludin) in the spectrin KD embryos should ideally be provided along with the TEER measurements in Fig 4B to conclude that the liquid-liquid barrier is defective.

To provide a more comprehensive understanding of TJ function and organization following spectrin loss, we will analyze ZO-1 and occludin protein localization and intensity in stratified in vitro cultures as well as in E17.5 spectrin KD embryos.

In some of the representative images (eg. Fig 2, S1e), it looks like the shSptan cells are much more abundant in the suprabasal layers than in the basal layer. Can the authors comment on whether these cells are more likely to get outcompeted/preferentially differentiate?

The reviewer raises an interesting point. Our new data indeed reveal a defect in the transition from the basal to the spinous layer, as evidenced by an increase in K14+/K10+ suprabasal cells and the suprabasal incorporation of EdU (as shown in the Figure “all-spectrin depletion enhances suprabasal proliferation” before). To further elucidate the nature of these defects, we will conduct experiments to assess whether spectrin KD epidermis exhibits abnormalities in cell delamination (K14+/K10+ basal layer cells; Soffer et al., 2022) and/or spindle orientation (survivin staining; Soffer et al., 2022). These analyses will help clarify the underlying mechanisms contributing to the observed defects.

The model in Figure 6 could use more annotation- why is there more F-actin (or more AJs?) in the SG1 cells on the left vs the right?

Revision Plan

We apologize for any confusion and will provide clearer annotations.

As previously published by us (Rübsam et al., Nat Commun. 2017) and again quantified in Figure 1e, the average F-actin intensity is highest in the SG1 layer. However, F-actin levels within SG1 cells can vary significantly (Rübsam et al., Nat Commun. 2017)—some cells exhibit extremely high levels, while others have low or even undetectable levels. This heterogeneity is also reflected in our model, and we have noted this observation in the figure legend.

Page 7- supplementary figure 1h is mislabelled, should be 1j; supplementary figure 1i should be 1k.

We corrected the mislabeling.

Page 16, final sentence of the first paragraph- typo?

We corrected the text.

The title for the final results section is "all-spectrin-actomyosin networks regulate the actomyosin skeleton"- is this correct?

The reviewer is correct, there is a mistake in the title. The correct title is "all-spectrin-actomyosin networks regulate the EGFR-TRPV3-TGM pathway" and has been corrected.

Reviewer #2 (Significance (Required)):

This manuscript focuses on an intriguing and understudied phenomenon: the coincidence of cell shape changes and progressive differentiation as keratinocytes move upward through the epidermal layers. Several cytoskeletal components have been identified that regulate shape but not differentiation, and vice versa, indicating that the two can at least to some extent be uncoupled- but in general the relationship is not well understood.

The main advance of this work is the identification of all-spectrin as a novel component that concurrently regulates cell shape, cytoskeletal organization, and barrier formation in the mammalian epidermis. With the exception of Arp2/3, this is to my knowledge the only cytoskeletal regulator whose loss affects barrier function. The authors also show relationships, albeit direct or indirect, with other key molecular players (eg Ecad, EGFR, TGM), improving mechanistic understanding of the network controlling terminal cytoskeletal changes and epidermal differentiation. The integration between in vivo phenotypes and in vitro experiments (eg the laser ablations) is also a strength.

The main drawback of this study is that the insights are largely correlative in nature. Loss of spectrin clearly affects both cell shape and barrier function, but the cause and effect are not

Revision Plan

clear here and all-spectrin could simply be influencing each aspect via parallel pathways. Overall, a unified model for how spectrin regulates and is regulated by adhesion, the cytoskeleton, EGF signaling, and/or calcium channels is missing.

The audience for this work would be largely those with relatively specialized interest in tissue morphogenesis, cytoskeletal biology, and epidermal development.

Reviewer expertise: epidermal biology, differentiation, signaling, mouse models, stem cells

We appreciate the reviewer's recognition of the novelty and significance of our study.

Regarding the "cause and effect" relationship, our findings demonstrate for the first time that the activity of the EGFR-TRPV-TGM pathway can be regulated by the actomyosin cytoskeleton. Additionally, we show that spectrin is a key regulator of actomyosin cytoskeletal organization and activity both in vitro and in vivo. Based on these results, our working model proposes that E-cadherin recruits spectrin, which organizes the contractile machinery essential for the activation of the EGFR-TRPV-TGM pathway. However, as the reviewer suggests, we cannot rule out the possibility that spectrin may also influence this pathway through additional mechanisms.

Revision Plan

Reviewer #3 (Evidence, reproducibility and clarity (Required)):

Summary: In this manuscript, Soffer et al. aim to elucidate the mechanisms driving cell shape during epidermal development and how those shape changes alter cell fates. The authors propose that spectrin integrates with E-cadherin and actomyosin networks to regulate cortical organization, mechanical properties, and differentiation. While the data presented are experimentally rigorous and propose several interesting models, several conclusions are overstated. Furthermore, while reading it, the organization seemed to jump between several ideas, and eventually felt like pieces of three separate stories. In addition, there are concerns with statistical analyses, and several aspects require clarification or more data. Below are specific comments and concerns:

We thank the reviewer for the insights and useful comments that have greatly helped to improve the clarity, to better and more clearly connect the main findings, and better communicate the main message of the story

Major concerns:

1. The suggestion that cell shape and cell fate are linked through spectrin is very exciting. But, as presented, the link is weak, preliminary, and speculative. The data presented regarding cell differentiation is really focused on cell function, which may or may not be caused by defects in differentiation.

- To test changes in cell fate, the authors only examine keratin expression and demonstrate that K14 expression is maintained in suprabasal cells of Sptan1-KD epidermis. To test whether cell identity is perturbed, a more comprehensive analysis should be performed. In particular, it would be beneficial to see K14 and K10 together, as both appear to have interesting expression patterns in the knockdown epidermis. P63 staining would also be useful to determine whether K14+ cells in suprabasal layers are still expressing multiple basal cell markers. The shapes in the K14+ suprabasal knockdown cells make it appear as though it is possible that basal cells are piling up on each other rather than suprabasal cells not flattening.

We appreciate the reviewer's feedback on this important point. To better understand defects in cell differentiation, we have already quantified K14/K10 double-positive cells and performed an EdU incorporation assay. Our results show a significant increase in K14/K10 double-positive cells in spectrin KO epidermis. EdU incorporation analysis revealed a comparable number of EdU+ basal cells between control and spectrin KD epidermis but a significant increase in EdU+ suprabasal cells in spectrin KD epidermis (see Figure below).

all-spectrin depletion enhances suprabasal proliferation. (A) Sagittal views of 10- μ m sections of dorsal skin from *shScr* (*Ctrl*) and *shSptan1 0595* transduced E17.5 embryos. Sections were labeled for EdU. arrows show suprabasal EdU+ cells. Upper Insets show the transduced cells (H2B-GFP+). **(B)** Quantification of EdU+ basal and suprabasal layers cells from the data shown in A. Horizontal bars represent the mean \pm SD from n=3 embryos per condition. Circles represent the average of EdU+ basal and suprabasal layers cells from each embryo. Not significant ($P=0.1104$) for *ctrl* versus *shSptan1 0595* and not significant ($P=0.1097$) for *ctrl* versus *shSptan1 9753* by unpaired t-test for EdU+ basal layer cells. $P=0.0028$ for *ctrl* versus *shSptan1 0595* and $P=0.0489$ for *ctrl* versus *shSptan1 9753* by unpaired t-test for EdU+ suprabasal layer cells. Nuclei were stained with DAPI, and dotted lines indicate the dermal-epidermal border. Scale bars = 20 μ m. ns, not significant.

As suggested, we will also include P63 staining, as well as examine how the inability to properly induce differentiation is linked to cell shape by examining whether suprabasal K14-positive cells also are smaller in size. Together, these findings further support the role of spectrin in cell fate regulation and epidermal differentiation and link these with changes in shape.

- For barrier function assays, the authors demonstrate decreased TEER in primary keratinocytes after calcium switch. Do tight junction proteins look perturbed in their localization and/or expression patterns in vitro and in vivo?

In order to provide a more comprehensive overview of TJ function and organization upon loss of spectrin, we will analyze ZO-1 and occludin protein localization to examine changes in junctional membrane organization and apical TJ structure in the most apical cells in stratified keratinocyte cultures and, also examine TJ organization in embryonic epidermal whole mounts at E17.5.

- For dye exclusion assays, the back seems to have formed an efficient barrier in the knockdown animal. Is this a true barrier defect or a developmental delay. If you look at E18.5, has the barrier on the ventral side of the animal formed more robustly? Do these animals survive perinatally? TEWL?

Revision Plan

As the reviewer suggested, we performed dye exclusion assays on E18.5 embryos. Unlike control embryos, spectrin KD embryos exhibited patches of dye on their heads (see Figure below). However, the reduced dye penetration in E18.5 spectrin KD embryos compared to E17.5 may indicate a developmental delay. Therefore, as the reviewer recommended, we will examine whether spectrin KD animals survive postnatally. Notably, the barrier defects observed in spectrin KD embryos are comparable to those observed in TRPV3 KO embryos (Cheng et al., Cell, 2011).

α II-spectrin activity is essential for barrier in E18.5 embryos.

(A) Dye exclusion assay. Wild type and *shSptan1 0595* transduced E18.5 embryos were treated with toluidine blue dye to evaluate the skin barrier. Lower insets show the H2B-GFP+ transduced embryo.

We did not perform TEWL analysis in embryos due to the mosaic nature of the KD technology, makes it hard to get consistent TEWL data, as we also experienced in earlier experiments with other knockdowns that affected barrier function.

- TGM1 activity is down in *shSptan1* and Ecadherin mutant epidermis. But what about TGM1 protein? Of course, activity would be decreased if the cells weren't differentiating properly, as proposed by the authors.

We agree that total TGM1 levels should be analyzed and will perform stainings accordingly.

2. The paper begins by demonstrating that SG2/1 cells maintain shapes when they are isolated. This is very clever experiment! However, there are no quantifications accompanying the images. In addition, it is difficult to say for sure, but the SS cells look like their nuclei get smaller between in vivo and isolated, which may say more about the cellular viability than cell shapes.

We have now quantified SG and SS 2D cell size and shape before and after isolation and included the quantification in Figure 1 of the preliminary revision (see also Figure below). Our quantifications show that SG cells maintain their cell shape but show a 19% reduction in size after isolation. In contrast, spinous cells lose their shape and round up becoming fully circular, also resulting in strong decrease in cell size by 86%.

Revision Plan

Cell-cell contact independent cell shapes in the granular layer. **a** Newborn epidermal whole-mount immunofluorescence analysis for Phalloidin (F-actin) revealing cell shapes in the spinous (SS cells) and granular layer (SG cells) within tissue (in vivo). Right column: Phalloidin staining of single cells isolated from the spinous or granular layer showing deformation upon isolation only in spinous cells. **b** Quantification of cell area/shape of granular and spinous cells in the tissue and after isolation using stainings as shown in **a**. * $P < 0.05$, **** $P < 0.0001$; >100 cells from 3 mice (isolated) or 6 mice (in tissue) with Kolmogorov-Smirnov per layer. **c** Quantification of cell area/shape of granular and spinous cells in the tissue and after isolation shown as mean values per mouse corresponding to **b**. * $P < 0.05$ with Mann-Whitney test.

3. In many of the quantifications throughout the manuscript, statistical analyses appear to be run on number of cells quantified rather than number of experimental animals. While I appreciate that there is a dynamic range within animals, using number of cells as the statistical n feels inappropriate, and is assigning very high statistical significance to what appear to be very subtle changes.

Although we agree that many individual datapoints, in this case cells, within one mouse as a basis for statistics will generally overestimate the significance, we very carefully performed the statistical analysis using not only number of cells but also compare the variation in mean/animal to avoid data bias and improper conclusions. Whereas the first can result in underestimating statistical errors, the latter overestimates statistical errors. To account for this potential problem, we have used non-parametric ANOVAs (Kruskal-Wallis followed by Dunn's multiple comparison test) when comparing more than two groups of accumulated cell numbers. Where possible, we used Kolmogorov-Smirnov that is more suitable for high N -numbers but cannot compare multiple

groups. In addition, we also included the mean values per animal and showed these graphs in the supplementary figures for a number of experiments and will add those were missing so far. Some quantifications of individual datapoints that were based on T-test will be redone using Kolmogorov-Smirnov. In case where we found that the difference between accumulated cell numbers is significant and the means per animal clearly show distinct populations even if these differences were statistically not significant with N=3 animals due to the stringent, non-parametric tests that we used, we still consider the differences relevant.

4. The data put a lot of emphasis on the micro-honeycomb lattice, but their significance is unclear. It does appear that they are spot junctions forming between the two layers of cells, but whether they contribute to cell shape, tension, etc, is very unclear.

Our data provide evidence that the organization of spectrin-actomyosin into a micro-honeycomb lattice cortical network plays a key role in regulating tension and cell shape. We also demonstrate that spectrin is essential for the formation of the network, and that the loss of the network in spectrin KD cells changes the properties of the actin cortex.

Notably, pharmacological inhibition of myosin II in spectrin KD cells does not restore the honeycomb-like organization, but it partially reverses the increased recoil. Together, these data indicate that a key role of the spectrin-actomyosin micro-honeycomb network is to ensure cortical stability by distributing actomyosin-generated tension.

Although understanding the detailed molecular mechanisms by which the micro-honeycomb network organization regulates cellular mechanics and cell shape would require extensive additional analysis beyond the scope of this manuscript, we will use whole mounts also assess how in vivo changes in honeycomb network organization upon spectrin KD relate to cell shape changes.

Our data further suggest that an intact spectrin-actomyosin honeycomb network supports the cortical localization of pEGFR. However, the precise role of this micro-honeycomb organization in processes such as receptor clustering or retention at the plasma membrane remains an open question for future investigation. To acknowledge this, we will add the following sentence to the discussion: "Whether the micro-honeycomb configuration of spectrin-actomyosin is necessary for organizing pEGFR and TRPV3 at the plasma membrane requires further investigation.

5. The conclusion that ablation results demonstrate viscoelastic behavior (Supplementary Fig. 3i) is not adequately supported. The authors should provide quantitative analysis, such as recoil velocity or relaxation time, to distinguish elastic and viscous properties. Additionally, the imaging duration (60 seconds) is insufficiently justified, and there is no correlation between the expansion direction and cortical behavior. Did the authors try to laser cut a cell junction and look at cortical tension differences?

Revision Plan

We agree with the reviewer that further analysis of this data is necessary, see also reviewer 1. While our ablation data closely resemble findings from previous studies, that demonstrated visco-elastic behavior (e.g., Saha et al., 2016; Vuong-Brender et al., 2017), we have conducted additional quantitative analyses over time to further assess visco-elastic properties (see preliminary results in Figure below).

We initially did laser cut individual cell junctions in differentiated cells; however, a small junctional cut did not produce a recoil on the vertex. We have now quantified this (see Figure a below). Upon additional ablation of the same junction, but now also including ablating the cortical micro-honeycomb lattice adjacent to this junctions resulted in vertex recoil (Figure c,d below), showing that this junction-coupled network is under tensile stress. We now also provide evidence that the presence of the cortical spectrin-F-actin micro honeycomb lattice vertices maintains tensile function even if a single junction is cut; (1) High-resolution imaging revealed that in keratinocytes perijunctional F-actin cables integrate with the cortical spectrin-actomyosin honeycomb network to form a transcellular spectrin-actomyosin cortical network. (2) Importantly, junctional ablation in keratinocytes with less mature cortices that have not formed a cortical micro honeycomb lattice yet, did result in vertex recoil (Figure a,b below). Together, these new data indicate the importance of distributing tensile stress along the micro-honeycomb network instead of parallel to junctions as seen in simple epithelia, thus stabilizing cell shape and cell contacts.

In our ablation experiments the tension generated by perijunctional F-actin cables did not appear to exceed the overall tension of the network, as evidenced by elliptical cortical openings following line ablation—even across junctions (see two examples in the second Figure below; yellow arrows indicate perijunctional actin cables). We thus focused on cortical ablation and have quantified the cortical behavior and recoil dynamics upon ablation (Figure e,f below). Using these data we will now assess whether this behavior aligns with a viscoelastic model. The data described here will be included in the revised manuscript. Together, these results will provide a better understanding of the mechanical relevance and properties of the lattices.

Revision Plan

Cortical laser ablation. Live imaging of SiR-actin (F-actin) labeled stratified Ctr keratinocytes after 48h high Ca^{2+} . **a** Junctional ablation in apical keratinocytes with or without a cortical F-actin lattice. Timepoints after ablation (yellow line) are shown as indicated. **b** Quantification of the increase in vertex (dashed line) distance upon ablation. Lines: mean \pm SD recoil of 11 ablations (no lattice) and 20 ablations (with lattice) from $n=3$ biological replicates each. **c** sequential ablation: short junctional ablation followed by a longer 17µm ablation across the same junction. The latter one ablating the cortex connected to the ablated junction. **d** Quantification of vertex distance increase of sequential ablations as shown in **c**. Line: mean \pm SD recoil of 7 sequential ablation from $N=2$ biological replicates. **e** Cortical laser ablation of a 17µm line and **f** quantification of the elliptical opening area (red line) over time. Line: mean \pm SD opening area of 13 ablations from $N=4$ biological replicates.

Cortical laser ablation (yellow line) across intercellular junctions (yellow arrows) 30 seconds after ablation.

Minor concerns:

6. The scale bars are mislabeled with "µM" instead of "µm" for micrometers, reflecting a lack of rigor in figure annotations. More broadly, figure labels are insufficiently detailed, detracting from clarity.

We will meticulously examine the figures and legends, correcting any errors.

7. There are several typos and sentence fragments throughout the text.

Revision Plan

We thank the reviewer for pointing this out and will carefully go over the text, and identify and correct any errors.

8. Can the authors do the isolated cell shape experiments on spectrin KD epidermal cells? If it is important in cell shapes, while the starting shape is clearly different already, it may also have defects in maintaining that shape. Similar for Ecad KO epidermis.

We appreciate the reviewer's suggestion. However, isolating granular layer cells is technically challenging, and obtaining sufficient spectrin KD cells for analysis is difficult, as many cells will still be wild-type. In the future, we plan to generate an epidermal-specific spectrin knockout, but this may not be feasible within the timeframe of the revisions.

9. Quantifications of Western blot and all-spectrin immunofluorescence in knockdown skin is needed.

We have quantified spectrin levels in both immunofluorescence and western blots. The revised manuscript will include the new data.

10. The authors suggest that all-spectrin recruitment to the cell cortex requires myosin II activity. Do they have proposed mechanisms that could be added to the discussion? For example, is increasing contractility sufficient to increase all-spectrin localization? Is this through mechanosensitive molecules? Is spectrin thought to be mechanosensitive?

We thank the reviewer for pointing this out and we will discuss myosin-dependent recruitment of spectrin and its mechanosensitivity in the revised manuscript. Non-erythrocytic spectrin was shown to be mechanosensitive in endothelial cells where it transmits shear forces to membrane tension and the tension-sensitive Ca channel PIEZO1 and thus shear-induced Ca-influx (Mylvaganam et al. NCB 2022). We will discuss these findings in light of our own data.

11. Are there any defects in proliferation/cell cycle in spastin KD cells in culture? Or cell spreading? It would be important to rule these out as possible contributors to decreased TEER - have they efficiently formed a confluent monolayer at a similar timing to controls?

We thank the reviewer for bringing up these points. While our TEER experiments did not reveal any obvious defects in cell density or layer formation, we will include quantifications of cell shape and cell proliferation in the revised manuscript, as suggested.

Reviewer #3 (Significance (Required)):

This paper combines in vivo and in vitro approaches to understand the role the spectrin cytoskeleton plays during epidermal development. In particular, it contributes to our

Revision Plan

understanding of how cells begin to flatten out as they differentiate into suprabasal cells. While the authors hint at the link between cell shape and cell fate/differentiation, it is more appropriate to say that they have linked cell shape to cell/tissue function, particularly barrier function. This manuscript would be interesting and relevant to several audiences - while definitely more basic research, it does have implications for dermatology. More broadly, this study contributes to our understanding of how cell junctions and actin are regulated and re.

We thank the reviewer for the kind words that acknowledge the general significance of our findings. We would like to emphasize that our data do not only link cell shape to tissue function/barrier function in our case, but also identify a key role for spectrin in regulating early cell fate and differentiation, as shown by the changes in K14 and K10 as well as late cell fate as shown by changes in TGM1 activation important for terminal differentiation and that we link to changes in EGFR/TRPV3 localization and activation.

Importantly, our proposed experiments will considerably strengthen and better characterize how E-cadherin-spectrin-actomyosin-dependent cell shape changes relate to changes in cell fate and early and late differentiation and are linked to EGFR/TRPV3 localization and activation.

Description of the revisions that have already been incorporated in the transferred manuscript
Please insert a point-by-point reply describing the revisions that were already carried out and included in the transferred manuscript. If no revisions have been carried out yet, please leave this section empty.

Reviewer2

Page 7- supplementary figure 1h is mislabelled, should be 1j; supplementary figure 1i should be 1k.

We corrected the mislabeling.

Page 16, final sentence of the first paragraph- typo?

We corrected the text.

The title for the final results section is "all-spectrin-actomyosin networks regulate the actomyosin skeleton"- is this correct?

The reviewer is correct, there is a mistake in the title. The correct title is "all-spectrin-actomyosin networks regulate the EGFR-TRPV3-TGM pathway" and has been corrected.

Reviewer3

Revision Plan

2. The paper begins by demonstrating that SG2/1 cells maintain shapes when they are isolated. This is very clever experiment! However, there are no quantifications accompanying the images. In addition, it is difficult to say for sure, but the SS cells look like their nuclei get smaller between in vivo and isolated, which may say more about the cellular viability than cell shapes.

We have now quantified SG and SS 2D cell size and shape before and after isolation and included the quantification in Figure 1b and Supplementary Fig. 1a of the preliminary revision. Our quantifications show that SG cells maintain their cell shape but show a 19% reduction in size after isolation. In contrast, spinous cells lose their shape and round up becoming fully circular, also resulting in strong decrease in cell size by 86%.

Minor concerns:

6. The scale bars are mislabeled with " μM " instead of " μm " for micrometers, reflecting a lack of rigor in figure annotations. More broadly, figure labels are insufficiently detailed, detracting from clarity.

The mislabeled μM has been corrected in all figures.

7. There are several typos and sentence fragments throughout the text.

We thank the reviewer for pointing this out have corrected mistakes in the text.

In addition we have partially reorganized Figure1 and removed old Figure1h,i being redundant with old Figure1m. New representative images for isolated Ctr and EcadKO SG cells were added. The supplementary Figure1 was adapted accordingly. Also the text to Figure1 in the results has been revised.

3. Description of analyses that authors prefer not to carry out

Please include a point-by-point response explaining why some of the requested data or additional analyses might not be necessary or cannot be provided within the scope of a revision. This can be due to time or resource limitations or in case of disagreement about the necessity of such additional data given the scope of the study. Please leave empty if not applicable.

Not applicable.

We thank the reviewers for their insightful feedback. As outlined above, we are confident that we can address the reviewers' comments in a timely manner.

March 4, 2025

RE: JCB Manuscript #202502071TR

Matthias Rübsam

University Hospital Cologne

Dear Dr. Matthias Rübsam,

Thank you for submitting your manuscript entitled "Spectrin coordinates cell shape and signaling essential for epidermal differentiation" to Journal of Cell Biology. I have consulted with a relevant member of the Editorial Board, and I am pleased to inform you that we are interested in considering this manuscript further at JCB. I am contacting you with a question concerning your revision plan.

We appreciated the novel observations that establish a role for spectrin in promoting homeostatic keratinocyte differentiation and tissue function. However we feel the current revision plan would leave important details unresolved. Rather than return this manuscript to you for submission elsewhere, we are contacting you to seek clarification and request additional data to resolve these issues.

In particular, the current work and revision plan state that actomyosin organization, dependent on spectrin, is required for cell shape changes and proper differentiation. However this work does not distinguish, to our reading, between the presence of filamentous actin vs its spatial arrangement with myosin at the cortex. Similarly, the means by which a pool of spectrin-dependent cortical actin enforces cell shape is not entirely clear (currently hinted at by data in Fig 1M and Fig S1I). The manuscript text (including the abstract) state that actomyosin organization in these cells is dependent on spectrins, however the architecture/spatial arrangement of cortical actin with myosin is not quantitatively evaluated. Blocking all actin polymerization by Latrunculin B tests a requirement for actin filaments throughout the entire cell but does not inform on a role for its spatial organization at the cortex. We feel these details are central to the main findings, and would remain obscure with the current revision plan. On this note Harmon et al also described a role for linear actin filaments in keratinocyte differentiation, which may be helpful (<https://doi.org/10.1083/jcb.202101008>).

Therefore we would be happy to invite submission of a revised manuscript, provided it offers greater clarity and new details on actin architecture at the cell cortex and its relation to cell shape. This evidence could be provided by cell culture assays without new genetic perturbations in vivo.

Please contact me with any questions about our requests. If you are able to furnish these additional details, we will be happy to invite a revised manuscript with these changes in addition to those stipulated already in the revision plan.

With best wishes,

Tim Fessenden
Senior Scientific Editor
Journal of Cell Biology

Full Revision

Manuscript number: RC-2024-02746

Corresponding author(s): Carlen M. Niessen, Chen Luxenburg, Matthias Rübsam

1. General Statements

We would like to thank the reviewers for their thoughtful comments as well as for the editor's guidance on strengthening our manuscript "Spectrin coordinates cortical actomyosin organization and differentiation essential for a functional barrier." We are happy to see that the reviewers find our data "of high quality" that advances "mechanistic understanding" and is of "interest to several audiences."

It is important to note that the main finding of this manuscript is that it identifies spectrin as a key spatially coordinating actomyosin-dependent cell shape changes and differentiation in an in vivo epithelium. Deletion of the cytoskeletal protein spectrin in the epidermis profoundly changes actomyosin organization to alter cell shape, and cell and tissue mechanics, resulting in disturbed epithelial differentiation and barrier function.

Nearly 170 years ago, Rudolf Virchow highlighted the critical role of cell shape in cellular function and disease. However, the mechanisms connecting cell shape, differentiation, and function remain poorly understood. Our study thus provides a new mechanism of actomyosin-dependent fate regulation in an environment determined by cell neighbors instead of the already well-described changes induced by bimodal interfaces where cells face either matrix or lumen, with important implications for, e.g., tumor environments.

Notably, this study provides an important missing link in our knowledge of how adhesive junctions spatially coordinate cellular actomyosin tension states and receptor tyrosine kinase signaling to control differentiation and epithelial barrier function. Our understanding of the role of spectrin in epithelia is only beginning to emerge, and this work provides a thorough and mechanistic understanding of the in vivo role of spectrin in epithelium with implications for other epithelia and beyond.

In the revision, we comprehensively addressed the reviewer's comments and included substantial new data to address the following key points brought up by the reviewers:

1. We now provide additional in vivo evidence that spectrin-coordinated changes in cell shape regulate early epidermal differentiation and proliferation using IF-based fate markers and quantifications of cell shape and position in the spectrin kd mice (new Fig. 5a,b; Supplementary Fig. 5a-e). We also have generated and now include a new epidermal knockout of all-spectrin that confirmed the changes in cell shape and barrier function and additionally revealed in vivo defects in cortical actomyosin lattice organization upon loss of all-spectrin (new Supplementary Fig. 1j,k; Fig. 3e,f; Fig. 4a,d; Fig. 5e,k; Supplementary Fig. 5g,h).

2. We have considerably strengthen the mechanistic link of how spectrin-actomyosin organization regulate EGFR/TRPV3 activity and keratinocyte differentiation (new Fig. 6f,g,m,n). Specifically using a combination of EGFR, and actomyosin agonists and antagonists together with IF-based

Full Revision

readouts for protein localization and cell differentiation, we find that spectrin-actomyosin tensions states in the granular layer are essential to localize active EGFR and TRPV3 at the membrane necessary for TGM1 activity while EGFR activity itself also regulates the organization of the spectrin-actin myosin network, that, based on our previous publication, also controls the tension state (new Fig. 7).

3. We now also included a careful *in vivo* and *in vitro* characterization of how depletion of all-spectrin changes the association of MyosinIIa with the cortical actin network (new Fig. 4a,c) and how this organization links to changes in tension (reorganized Figure 4). In brief, it shows how spectrin is essential to differentially control the organization and tensile states of cortical spectrin-actomyosin networks in each of the epidermal layers.

4. We also provide new evidence on how the configuration of spectrin-dependent cortical F-actin networks regulates barrier function. To this end we performed *in vivo* 3D analysis of the spectrin-actomyosin lattice (new Fig. 3e,f) and tight junctional network (new 5e,f) in embryos and newborns. We find that loss of all-spectrin through changes in cell shape alters the coordinated positioning of TJs in the granular layer, thus showing the *in vivo* relevance of our *in vitro* findings that spectrin is essential for proper TJ barrier function.

5. We extended our laser ablation studies and provide *in depth* quantification to characterize recoil behavior over time, and to better assess how the loss of spectrin alters the visco-elastic behavior of the actomyosin cortex in suprabasal cells (new Fig. 4h,i; Supplementary Fig. 4c-f).

6. We have further added a thorough 3D analysis of the actual cell shape changes during keratinocyte differentiation *in vivo* (new Fig. 1a-c; Supplementary Fig. 1a,b).

We are confident that these additions have addressed all major comments of the reviewer and substantially improved our study's clarity, depth, mechanistic insights, and overall impact.

Senior Scientific Editor - Journal of Cell Biology - Tim Fessenden

We appreciated the novel observations that establish a role for spectrin in promoting homeostatic keratinocyte differentiation and tissue function. However we feel the current revision plan would leave important details unresolved. Rather than return this manuscript to you for submission elsewhere, we are contacting you to seek clarification and request additional data to resolve these issues.

In particular, the current work and revision plan state that actomyosin organization, dependent on spectrin, is required for cell shape changes and proper differentiation. However this work does not distinguish, to our reading, between the presence of filamentous actin vs its spatial arrangement with myosin at the cortex. Similarly, the means by which a pool of spectrin-dependent cortical actin enforces cell shape is not entirely clear (currently hinted at by data in Fig

Full Revision

1M and Fig S1I). The manuscript text (including the abstract) state that actomyosin organization in these cells is dependent on spectrins, however the architecture/spatial arrangement of cortical actin with myosin is not quantitatively evaluated. Blocking all actin polymerization by Latrunculin B tests a requirement for actin filaments throughout the entire cell but does not inform on a role for its spatial organization at the cortex. We feel these details are central to the main findings, and would remain obscure with the current revision plan. On this note Harmon et al also described a role for linear actin filaments in keratinocyte differentiation, which may be helpful (<https://doi.org/10.1083/jcb.202101008>).

We thank the editor for pointing out this important and central point on how myosin is arranged within the cortical spectrin-actin network. We included novel data that better describe how spectrin organizes the cortical actomyosin cytoskeleton both in vivo and in vitro. To this end, we stained for F-actin, α -spectrin and (non-muscle) myosin-IIa in vivo in 3D epidermal whole mounts, and in vitro in differentiated, multilayered keratinocyte cultures (New Fig. 4a,d). In vivo we find that concomitant with major flattening of keratinocytes in the granular vs. the spinous layer, myosin-IIa relocates from lateral intercellular contacts in the spinous layer to a spot-like localization across the cortical spectrin-actin networks in the granular layer, thus indicating an increase in tensile state of this network when cells move from the spinous to the granular layer, in line with our previous work (Rübsam et al *Nat. Comm.* 2017). Importantly, loss of spectrin increased myosin intensity and size of spots in all suprabasal layers, indicating that loss of spectrin increased the contractile state of F-actin networks. In vitro Myosin-IIa similarly decorates the cortical spectrin-F-actin honeycomb-like lattice of suprabasal cells in a spot-like manner across the network. Upon loss of α -spectrin, F-actin is organized in stress fiber-like filaments that now show an intense periodic decoration of myosin-IIa, indicating highly contractile F-actin fibers (new Fig. 4d; new Supplementary Fig. 4a), in line with increased recoil upon laser ablation (Fig. 4j). Taken together, our in vivo and in vitro data show that spectrin is essential for the formation of continuous sub-membranous lateral and apical spectrin F-actin lattices in which myosin-IIa is regularly distributed in a spot-like manner to dissipate tension across the network. This organization that also incorporates the elastic properties of spectrin is likely necessary to accommodate not only shape changes itself, but also allow to absorb forces experienced when cells move into a new layer. Importantly, the cell shape of each layer is characterized by a defined network organization and tensile state that differs from other layers. In agreement, in vivo depletion of α -spectrin alters the conformation and tensile state of these networks likely explaining observed changes in cell shape. From that point on, we can only hypothesize how spectrin-actomyosin networks mechanically control cell shapes: We think the local contraction of a continuous network that incorporates the elastic properties of spectrin and is coupled through AJ to neighboring cells enables a very defined deformation of the cortex as local pulling forces can be transmitted in all directions. In contrast, linear cortical F-actin organization as observed in simple epithelia or F-actin stress fibers only allow force transmission in one direction. Moreover, the elasticity provided by spectrin prevents rupture of the F-actin network in a contractile state during shape changes.

Full Revision

Reviewer #1 (Evidence, reproducibility and clarity (Required)):

This manuscript describes the in vivo impact of spectrin in the complex epithelium of the skin. Spectrin is a membrane-skeleton organizer that has been implicated in many aspects of cell biology, including cell-cell interactions via adherens junctions. Much of this work has been done in vitro and there is less precedent for testing its impact in tissues, in part because the diverse cell-biological functions of spectrin have made it difficult to directly attribute physiological function to cellular mechanism. This paper demonstrates that spectrin has a key role in skin homeostasis that can be linked to an impact on adherens junctions. The authors show that alphaII-spectrin depletion disturbs skin barrier function and differentiation. Spectrin concentrated at AJ and was recruited in response to both E-cadherin adhesion and cellular contractility, suggesting that it responds to mechanically-active AJ. In turn, spectrin was necessary to recruit EGFR and TRPV3, elements known to participate in epidermal differentiation.

We thank the reviewer for recognizing the importance and impact of addressing the in vivo function of spectrin in epithelia and more specifically its role in regulation epidermal differentiation and barrier formation.

Specific comments:

1. Relationship between spectrin and EGFR, TRPV signaling. These signals have been implicated in skin differentiation in other studies and the authors implicate them in the spectrin phenotype based on loss of signal and localization. I wonder if it is possible to obtain more direct functional data in their system. For example, would inhibiting EGFR signaling accentuate the spectrin-depletion phenotype. I appreciate that genetic tests may be beyond the reasonable scope of a revision.

We thank the reviewer for this important comment. For clarity, although EGFR and TRPV signaling have been linked to skin differentiation and barrier function in other studies (e.g., Cheng et al. Cell 2010), these studies did not explore the role of the cytoskeleton and adhesion in regulating EGFR-dependent activation of TRPV3 nor has spectrin itself been implicated in epithelial barrier function.

To explore further and address how spectrin regulates EGFR and TRPV3 and examine interactions between these three proteins, we have employed a combination of in vivo and in vitro approaches: Our data show that either loss of spectrin, or inhibition of actin organization or actomyosin contractility disturbs EGFR and TRPV3 membrane localization especially in the granular layer as well as alters TGM activity (new Fig. 6). In the revised manuscript, we now include data in which we used EGFR inhibitors in vivo and find that its inhibition not only inhibits membrane localization of TRPV3 and TGM1 activity, in agreement with the findings of Chen et al., 2010), but also impairs cortical recruitment of spectrin (Fig. 7a-c, h-j). Previously, we have shown that EGFR itself controls cortical tension downstream of E-cadherin (Rübsam et al., Nat. Comm. 2017). Moreover, we show in vitro that either inhibition or overactivation of the EGFR

Full Revision

impairs both junctional spectrin and TRPV3 recruitment (Fig. 7d-g), again very much in line with previous observations that the timing and level of EGFR activation is critical for barrier function in keratinocytes (Rübsam et al. Nat. Comm. 2017).

Integrating this old data with our new data thus provide direct evidence that E-cadherin-dependent organization and tensile state of the cortical spectrin-actomyosin cortex is essential to spatially control EGFR and TRPV3 activity at the membrane in the granular layer. This in turn not only activates TGM1 but through EGFR also provides a positive feedback loop to maintain the spectrin-actomyosin cortical organization in this layer.

2. Characterization of cell mechanics. I think that the kinetics of recoil are more informative as proxies of tension than single time points. Can they extract time-series from their movies? It would also be helpful to analyse the recoil against a model of viscoelasticity.

We agree with the reviewer that the kinetics of recoil provides more insight, and now included a thorough analysis of recoil behavior and kinetics. These data show that the recoil behavior aligns with a visco-elasticity model (see Fig. 4g-l; Supplementary Fig. 4 c-f). Please also see our answer to a related point of reviewer 3.

Tiny

points

p16, top paragraph - some words seems to have been lost from the last sentence.

Thank you for pointing this out and we have corrected this by removing the accidental duplication of "initial E-cadherin engagement" that caused the feeling of words lost.

Reviewer #1 (Significance (Required)):

Overall, this suggests a high-level pathway where spectrin mediates between cell-cell junctions and epidermal homeostasis. The report doesn't contain novel cellular mechanism, but it is a valuable contribution to the field with data of high quality. It provides in vivo data that will be a foundation for further developments in the field.

We appreciate the reviewer's positive feedback, even though we respectfully disagree with the assertion that our manuscript lacks a 'novel cellular mechanism.' While EGFR and TRPV3 have been previously implicated in regulating epidermal terminal differentiation (Cheng et al. Cell 2010)— and our own work has shown that adherens junctions coordinate tension and EGFR signaling to position the tight junctional barrier in the upper suprabasal layers (Rübsam et al., Nat. Comm. 2017)— the role of spectrin in these pathways was entirely unknown. Our study offers a new regulatory perspective by, for the first time, manipulating spectrin and ankyrin in the developing epidermis.

Full Revision

Moreover, while the role of actomyosin-dependent cell mechanics is well established in regulating basal-to-spinous fate transitions, much less is known about its role in suprabasal fate changes and terminal differentiation. In this manuscript, we identify spectrin as a key downstream component of adherens junctions that governs actomyosin cortex organization and mechanics, linking this to TGM1 activation in the stratum granulosum. Thus, our study provides novel insight into how differential gradients of all-spectrin, actin and myosin II activity spatially control cortical actomyosin mechanics to drive cell shape changes and how this process is integrated into terminal cell fate regulation via the EGFR-TRPV3-calcium pathway.

Our data further demonstrate that, in addition to differentiation regulated by changes in gene expression, layer-specific changes in structure and mechanics of the cell cortex serve as an additional regulatory layer necessary to spatially activate enzymes whose expression is induced by changes in gene expression ("genetic differentiation"). A recent study from the Lechler lab (Prado-Mantilla et al., *eLife* preprint, 2024) also provided evidence that inducing contractility is sufficient to drive granular cell fate, even within the spinous layer. Our study expands on this study and now provides novel mechanistic insight into how changes in actomyosin contractility control granular layer cell fate by revealing that spectrin controls layer-dependent actomyosin organization to enable EGFR-TRPV3-mediated activation of TGM in the granular layer necessary for cortex differentiation, which is independent of gene regulation.

Reviewer #2 (Evidence, reproducibility and clarity (Required)):

In this manuscript, Soffer, Bosale et al. examine the role of spectrin in differentiating mouse keratinocytes in vivo and in vitro with a focus on cell shape, actomyosin contractility, and barrier formation. They find that all-spectrin is present at higher levels in differentiating keratinocytes (spinous and granular layers) and is required for the cell flattening that accompanies terminal differentiation. While cortical recruitment of spectrin is independent of ankyrin G, it is disrupted by the loss of E-cadherin or inhibition of the actomyosin cytoskeleton. In turn, spectrin regulates the lattice-like organization of cortical actomyosin, and its deletion in in vitro induces the formation of contractile stress fibers. Finally, spectrin depletion alters expression patterns of canonical differentiation markers, prevents proper barrier formation, and decreases both EGFR phosphorylation and cortical enrichment of TRPV2. Overall, the authors conclude that all-spectrin acts downstream of E-cadherin to regulate actomyosin organization, cell shape, and epidermal differentiation.

Major comments:

It is difficult to compare the cell shape changes in spectrin knockdown embryos vs Ecad mutants (Figure 1g vs 1l. The graph axes/units are different and 1l also includes a "shape index" (not described in the Methods) and subdivides cells by layer position- please standardize. Currently it appears that the spectrin KD embryos show a more severe effect- can the authors comment on why this might be?

Full Revision

We have now used the cell shape (perimeter/varea) throughout next to cell area and provided a clear explanation of these parameters in the methods. For the SpectrinKD skin we only discriminate between basal and suprabasal layers as not all cells are expressing the short hairpin and, moreover, in the embryo it is more challenging as the individual; suprabasal layers are less well defined. To directly compare to the changes in newborns upon loss of E-cadherin (Fig. 2 f,g), we therefore have generated and now included a K14-Cre driven all-spectrin knockout model and analyzed cell size (sagittal area) and shape index in the different suprabasal layers also at newborn stage (new Supplementary Fig. 1j,k). Importantly, spectrinKO images and its analysis confirm that either knockdown or loss of spectrin affects cell size and shape, even if the changes in cell shape are no longer significant at the newborn stage unlike at embryo stages. Loss of E-cadherin also affects cell shape in newborns, but here changes in shape are most obvious whereas for spectrin changes in cell size are most obvious. This is perhaps not so surprising taken that next to spectrin, E-cadherin-based AJ also recruit actin itself as well as many other regulators of actomyosin organization and this may affect cell shape beyond loss of spectrin from the cortex. This is perhaps best illustrated in the spinous layer where loss of E-cadherin, unlike loss of spectrin, increases flattening likely as a result of increased cortical actin recruitment (Fig. 2 f,g and Rüksam et al. Nat. Comm. 2017). In agreement, epidermal loss of E-cadherin results in perinatal death with large breaches in epidermal TJ barrier (Tunggal et al EMBO J 2005; Rüksam et al, Nat. Comm. 2017), whereas epidermal loss of spectrin alters TJ organization but are viable.

A key observation is that spectrin and F-actin co-localize in a lattice-like distribution in differentiating cells, and that loss of spectrin in vitro leads to the formation of contractile stress fibers. It is difficult to reconcile the in vitro data with the in vivo phenotype in Figure 2h. The authors should provide top-down images from spectrin KD embryos (from whole mounts or isolated cells) to show actin organization changes in vivo. Ideally, similar data should be presented for the Ecad mutant embryos to demonstrate that the spectrin-actin lattice is Ecad dependent.

The reviewer brings up an important point and we initially addressed this by performing 3D epidermal whole-mount analyses of the spectrin-F-actin lattice in spectrin KD. However, for technical reasons such as e.g., more cellular undulations in the embryonic epidermis, we did not achieve sufficient resolution to properly analyze the delicate cortical F-actin structures. Instead, we therefore analyzed all-spectrin KO newborn whole mounts and observed, more subtle defects in the F-actin network now showing a streak- and dot-like appearance (Fig. 3 e,f). We then extend our in vitro analyses and found that when we reduced tension through low dosis of blebbistatin (5 μ M) in spectrin-deficient stratified keratinocytes resulted in similar streak- and dot-like F-actin structures instead of stress fibers (Fig. 4f compare to Fig.3j), indicating that the cortical tensile state of in vitro stratified keratinocytes is much higher than those of in vivo, thus explaining why the in vivo defect in network organization is different from that in vitro.

There does not appear to be much Ank3 knockdown by immunofluorescence, although it is hard to tell (Figure S2c). Can the authors provide quantification for protein level (IF signal)? This is crucial to make the point that Ank3 is not important for spectrin localization.

Full Revision

As the reviewer suggested, we included quantification of immunofluorescence for ANK3, along with Western blot and qPCR analyses showing a clear reduction of ANK3 transcript and protein (Supplementary Fig. 2a-c).

The main data supporting the claim that spectrin KD embryos have a differentiation defect is the ectopic expression of K14 in suprabasal layers. From Figure 4a it looks like there are localized regions where K14+ cells accumulate, and adjacent regions where the layer organization is relatively normal. Please quantify how much of the epidermis (by length) is occupied by this type of ectopic K14 accumulation, and demonstrate with co-staining whether the K14+ suprabasal cells are also K10 (or Loricrin) positive

As this reviewer and reviewer #3 also suggested, we co-stained for K14/K10 and found that upon SpectrinKD all suprabasal cells were K10 positive independent of whether cells were K14 positive or negative. Moreover, we stained for the basal marker Lef1 which was negative in the suprabasal layers, indicating that spectrin only partially interferes with early differentiation (reviewer panel 1). Nevertheless, consistent with ectopic suprabasal expression of K14, our new data also show increased suprabasal proliferation, as indicated by EdU incorporation in spectrin KD epidermis (Supplementary Fig. 5d,e). Together these findings demonstrate that spectrin loss disrupts proper early epidermal differentiation when basal cells move suprabasally. We then asked whether the inability to induce a full early differentiation program is linked to cell shape and found a more round phenotype only in the K14-positive suprabasal cells whereas the increase in size upon SpectrinKD was less dependent on the differentiation status (Fig. 5a,b). Together, these data suggest that spectrin is essential for initial cell shape changes in cell shape and proper early differentiation, whereas differentiation itself then may be necessary to drive the full shape changes during the basal-to-suprabasal transition, perhaps by inducing expression of key cytoskeletal regulators that promote initial flattening.

Reviewer Panel 1: Dorsal skin sections from *shSptan1 0595* transduced E17.5 embryos immunolabeled for LEF1.

The conclusion that spectrin promotes TGM by regulating the EGFR-TRPV3 pathway is not currently well supported. First, it is not clear to what extent mislocalization of TRPV3-GFP serves as a readout for endogenous TRPV3 activity.

Full Revision

While the reviewer is correct that altered TRPV3 localization is an indirect measure of TRPV3 activity, this approach has been widely used in the field as a proxy for activity (e.g. Li et al., 2016; Lei et al., 2023). We tested multiple antibodies to assess whether spectrin loss affects endogenous TRPV3 localization, but unfortunately, none of these worked in vivo. Therefore, we utilized TRPV3-GFP as a reporter to examine its localization, as done in previous studies (e.g., Xiao et al., 2008; Liu et al., 2021).

We have now identified a TRPV3 antibody that functions in cell culture but not in tissues. Using this TRPV3 antibody, we observed that endogenous TRPV3 localizes to cell junctions (cell periphery). Notably, spectrin knockdown or pharmacological disruption of actomyosin severely impaired TRPV3 localization at cell-cell contacts, mirroring the TRPV3-GFP pattern observed in spectrin knockdown embryos. These findings validate TRPV3-GFP as a reliable reporter for TRPV3 localization and further support our in vivo observations (see data below).

We also tried to establish Ca²⁺ imaging but so far without clear success for several reasons. We had obtained immortalized keratinocytes expressing the Ca²⁺ reporter GCaMP6, but unfortunately, these cells did not differentiate properly and thus could not be used to analyze effect of TRPV3/EGFR or spectrin perturbation in properly stratified cells. These cells did respond to a TRPV3 agonist 2-ABP but the signal was very short lived (3-5min) and hence this treatment seemed inappropriate to study effects on differentiation either. We then utilized Cal520, a cell membrane permeable Ca²⁺ reporter dye. The signal-to-noise ratio of this dye was poor and further the dye was washed out after 2h during live imaging. For future experiments we plan to generate keratinocytes stably expressing a Ca²⁺ reporter as well as cross the GCaMP mice to our all-spectrinKO mice but we feel strongly with other additional new data that these experiments are outside the scope of this manuscript.

Second, there is no direct evidence supporting a cause-and-effect relationship. One suggestion would be to treat cultured KTCs with V3 agonists, which should stimulate TGM activity in wt cells but not in spectrin KD cells if TRPV3 activity is indeed affected.

We agree with the reviewer and we really would have liked to close this gap. However, as described above we found that the TRPV3 agonist only resulted in brief activation the TRPV3. A further complicated factor is that in our hands and as confirmed through personal communication with other laboratories TGM expression and other key hallmarks of differentiation are not reliably induced upon differentiation in cultured mouse keratinocytes, thus making it very difficult to assess whether TRPV3 stimulation is sufficient to activate TGM1. We are currently exploring alternative differentiation readouts either in mouse or human keratinocytes to gain functional insights into how spectrin-mediated regulation of TRPV3 activity influences differentiation in future studies.

Alternately, does the overexpression of TRPV2-GFP lead to any rescue of TGM in the embryos?

Full Revision

Unfortunately, we cannot reliably quantify TGM activity since we can only generate small clones of *shScr; TRPV-GFP / shSptan1; TRPV-GFP* transduced epidermis due to the large size of the insert (*TRPV3-GFP*) and the low titer that this size allows. Nevertheless, our results show that upon overexpression of TRPV3-GFP (*shScr; TRPV-GFP*) the epidermis exhibits the typical flat morphology in the granular layer, while combining spectrin KD with TRPV3-GFP overexpression (*shSptan1; TRPV-GFP*) in the epidermis still shows a profound defect in cell compaction, similar to spectrin KD cells (Fig. 6 d). These results suggest that overexpression of TRPV3-GFP is not sufficient to rescue either cell shape changes, or terminal differentiation and barrier function. Instead, EGFR and TRPV3 might perhaps have to work in concert for these changes.

Figure 1 starts out with some nice observations about the differences between contact dependence and spectrin/F-actin levels between upper (SG1/SG2) and lower (SG3/SS) suprabasal layers. With the exception of Fig 1l, less care is taken throughout the rest of the manuscript to characterize layer specificity of spectrin LOF phenotypes or the spectrin-actin relationship. Can the authors clarify in the discussion how they interpret their results in the context of the different layers and their dependence/independence on cell contact?

For the SpectrinKD skin we only discriminate between basal and suprabasal layers as not all cells are expressing the short hairpin and, moreover, in the embryo it is more challenging as the individual; suprabasal layers are less well defined. To characterize layer specificity of the spectrin LOF and to directly compare to the changes in newborns upon loss of E-cadherin (Fig. 2 f,g), we therefore have generated and also included a K14-Cre driven all-spectrin knockout model and analyzed cell size (sagittal area) and shape index in the different suprabasal layers also at newborn stage (new Supplementary Fig. 1j,k). Importantly, spectrinKO images and its analysis confirm that either knockdown or loss of spectrin affects cell shape, even if the changes in cell shape index are no longer significant at the newborn stage unlike at embryo stages. Loss of E-cadherin also alters cell shape in all layers in newborns, but here changes in the shape index are most obvious whereas for spectrin changes in cell size are most obvious. This is perhaps not so surprising taken that next to spectrin, E-cadherin-based AJ also recruit actin itself as well as many other regulators of actomyosin organization and this may affect cell shape beyond loss of spectrin from the cortex. This is perhaps best illustrated in the spinous layer where loss of E-cadherin, unlike loss of spectrin, increases flattening perhaps as a result of increased cortical actin recruitment (Fig. 2 f,g and Rübsam et al. Nat. Comm. 2017).

Minor

comments:

Some staining for TJ markers (ZO-1, occludin) in the spectrin KD embryos should ideally be provided along with the TEER measurements in Fig 4B to conclude that the liquid-liquid barrier is defective.

To provide a more comprehensive understanding of TJ function and organization following spectrin loss, we have analyzed occludin in stratified in vitro cultures and despite the clear

Full Revision

difference in trans-epithelial resistance we did not observe an obvious change in occludin localization or intensity. We then performed 3D whole mount analysis of all-spectrinKO newborn epidermis staining for ZO-1 and occludin (not shown) both showing the same pattern but the ZO-1 showing a much better signal to noise. We found that while intensity remained unchanged, the TJ network organization was altered. More specifically, the newly forming TJ rings did not align with the existing ones anymore due to the altered cell shapes and decreased alignment of the tetradecahedron shaped SG1-3 cells (Fig. 5e,f).

In some of the representative images (eg. Fig 2, S1e), it looks like the shSptan cells are much more abundant in the suprabasal layers than in the basal layer. Can the authors comment on whether these cells are more likely to get outcompeted/preferentially differentiate?

The reviewer raises an interesting point. Our new data indeed reveal a defect in the transition from the basal to the spinous layer, as evidenced by an increase in K14+/K10+ suprabasal cells and the suprabasal incorporation of EdU. We have therefore also analyzed the spindle orientation (survivin staining; Soffer et al., 2022) of proliferating basal cells in Ctr and SpectrinKD E17.5 embryos but did not observe any obvious difference. At this point we cannot say if suprabasal SpectrinKD cells originate from delaminated basal cells or suprabasal proliferation. In the future we plan to analyze delamination of Spectrin deficient cells more specifically but this may require live imaging of delaminating cells which is outside the scope of this manuscript.

The model in Figure 6 could use more annotation- why is there more F-actin (or more AJs?) in the SG1 cells on the left vs the right?

We apologize for any confusion. As previously published by us (Rübsam et al., Nat. Commun. 2017) and again quantified in Figure 1e, the average F-actin intensity is highest in the SG1 layer. However, F-actin levels within SG1 cells can vary significantly (Rübsam et al., Nat. Commun. 2017) some cells exhibit extremely high levels, while others have low or even undetectable levels. This heterogeneity is also reflected in our model, and we have noted this observation in the figure legend.

Page 7- supplementary figure 1h is mislabelled, should be 1j; supplementary figure 1i should be 1k.

We corrected the mislabeling.

Page 16, final sentence of the first paragraph- typo?

We corrected the text.

The title for the final results section is "all-spectrin-actomyosin networks regulate the actomyosin skeleton"- is this correct?

Full Revision

The reviewer is correct, there is a mistake in the title. The correct title is “all-spectrin-actomyosin networks regulate the EGFR-TRPV3-TGM pathway” and has been corrected.

Reviewer #2 (Significance (Required)):

This manuscript focuses on an intriguing and understudied phenomenon: the coincidence of cell shape changes and progressive differentiation as keratinocytes move upward through the epidermal layers. Several cytoskeletal components have been identified that regulate shape but not differentiation, and vice versa, indicating that the two can at least to some extent be uncoupled- but in general the relationship is not well understood.

The main advance of this work is the identification of all-spectrin as a novel component that concurrently regulates cell shape, cytoskeletal organization, and barrier formation in the mammalian epidermis. With the exception of Arp2/3, this is to my knowledge the only cytoskeletal regulator whose loss affects barrier function. The authors also show relationships, albeit direct or indirect, with other key molecular players (eg Ecad, EGFR, TGM), improving mechanistic understanding of the network controlling terminal cytoskeletal changes and epidermal differentiation. The integration between in vivo phenotypes and in vitro experiments (eg the laser ablations) is also a strength.

The main drawback of this study is that the insights are largely correlative in nature. Loss of spectrin clearly affects both cell shape and barrier function, but the cause and effect are not clear here and all-spectrin could simply be influencing each aspect via parallel pathways. Overall, a unified model for how spectrin regulates and is regulated by adhesion, the cytoskeleton, EGF signaling, and/or calcium channels is missing.

The audience for this work would be largely those with relatively specialized interest in tissue morphogenesis, cytoskeletal biology, and epidermal development.

Reviewer expertise: epidermal biology, differentiation, signaling, mouse models, stem cells

We appreciate the reviewer’s recognition of the novelty and significance of our study.

Regarding the “cause and effect” relationship, our findings demonstrate for the first time that the activity of the EGFR-TRPV-TGM pathway can be regulated by the actomyosin cytoskeleton. Additionally, we show that spectrin is a key regulator of actomyosin cytoskeletal organization and activity both in vitro and in vivo. Based on the initial and new added results, our working model proposes that E-cadherin recruits spectrin, which spatially organizes the contractile actomyosin machinery essential for the activation of the EGFR-TRPV-TGM pathway in the granular layer. However, as the reviewer suggests, we cannot rule out the possibility that spectrin may also influence this pathway through additional mechanisms.

Full Revision

Reviewer #3 (Evidence, reproducibility and clarity (Required)):

Summary: In this manuscript, Soffer et al. aim to elucidate the mechanisms driving cell shape during epidermal development and how those shape changes alter cell fates. The authors propose that spectrin integrates with E-cadherin and actomyosin networks to regulate cortical organization, mechanical properties, and differentiation. While the data presented are experimentally rigorous and propose several interesting models, several conclusions are overstated. Furthermore, while reading it, the organization seemed to jump between several ideas, and eventually felt like pieces of three separate stories. In addition, there are concerns with statistical analyses, and several aspects require clarification or more data. Below are specific comments and concerns:

We thank the reviewer for the insights and useful comments that have greatly helped to improve the clarity, to better and more clearly connect the main findings, and better communicate the main message of the story

Major concerns:

1. The suggestion that cell shape and cell fate are linked through spectrin is very exciting. But, as presented, the link is weak, preliminary, and speculative. The data presented regarding cell differentiation is really focused on cell function, which may or may not be caused by defects in differentiation.

- To test changes in cell fate, the authors only examine keratin expression and demonstrate that K14 expression is maintained in suprabasal cells of Sptan1-KD epidermis. To test whether cell identity is perturbed, a more comprehensive analysis should be performed. In particular, it would be beneficial to see K14 and K10 together, as both appear to have interesting expression patterns in the knockdown epidermis. P63 staining would also be useful to determine whether K14+ cells in suprabasal layers are still expressing multiple basal cell markers. The shapes in the K14+ suprabasal knockdown cells make it appear as though it is possible that basal cells are piling up on each other rather than suprabasal cells not flattening.

As this reviewer and reviewer #1 also suggested, we co-stained for K14/K10 and found that upon SpectrinKD all suprabasal cells were K10 positive independent of whether cells were K14 positive or negative. We stained for the basal marker Lef1 which was negative in the suprabasal layers (reviewer panel 1 below), indicating that spectrin only partially interferes with early differentiation (reviewer panel 1). Nevertheless, consistent with ectopic suprabasal expression of K14, our new data did show increased suprabasal proliferation, as indicated by EdU incorporation in spectrin KD epidermis (Supplementary Fig. 5d,e). Together these findings demonstrate that spectrin loss disrupts proper early epidermal differentiation when basal cells move suprabasally even if they initiate expression of K10 and downregulate Lef1. We then asked whether the inability to fully induce an early differentiation program is linked to cell shape and found a more round phenotype only in the K14-positive suprabasal cells (Fig. 5a,b). Together, data suggest that spectrin is

Full Revision

essential for initial cell shape changes and proper early differentiation, whereas differentiation itself then seems necessary to drive the full shape changes during the basal-to-suprabasal transition, perhaps by inducing expression of key cytoskeletal regulators that promote further flattening.

Reviewer panel 1: Dorsal skin sections from *shSptan1 0595* transduced E17.5 embryos immunolabeled for Lef1.

- For barrier function assays, the authors demonstrate decreased TEER in primary keratinocytes after calcium switch. Do tight junction proteins look perturbed in their localization and/or expression patterns in vitro and in vivo?

To provide a more comprehensive understanding of TJ function and organization following spectrin loss, we have analyzed occludin in stratified in vitro cultures and despite the clear difference in trans-epithelial resistance we did not observe an obvious change in occludin localization or intensity. We then performed 3D whole mount analysis of all-spectrinKO newborn epidermis staining for ZO-1 and occludin (not shown) both showing the same pattern but the ZO-1 showing a much better signal to noise. We found that while intensity remained unchanged, the TJ network organization was altered. More specifically, the newly forming TJ rings did not align with the existing ones anymore due to the altered cell shapes and decreased alignment of the tetrakaidecahedron shaped SG1-3 cells (Fig. 5e,f).

- For dye exclusion assays, the back seems to have formed an efficient barrier in the knockdown animal. Is this a true barrier defect or a developmental delay. If you look at E18.5, has the barrier on the ventral side of the animal formed more robustly? Do these animals survive perinatally? TEWL?

As the reviewer suggested, we performed dye exclusion assays on E18.5 embryos. Unlike control embryos, spectrin KD embryos exhibited patches of dye on their heads (Supplementary Fig. 5o). However, the reduced dye penetration in E18.5 spectrin KD embryos compared to E17.5 may indeed indicate a developmental delay. We did not perform TEWL analysis in spectrinKD embryos due to the mosaic nature of the KD technology, which makes it hard to get consistent TEWL data, as we also experienced in earlier experiments with other knockdowns that affected barrier

Full Revision

function. Instead we have added a *αII-spectrin* (*Sptan1*)KO model to delete *αII-spectrin* in all epidermal cells. These KO newborns survive and show no obvious increased transepidermal water loss upon birth (Fig. 5n) despite altered TJ organization (Fig. 5e,f) and reduced TGM activity (Supplementary Fig. 5h,i). We hypothesize that spectrin is more relevant in mechanically challenged conditions which may explain why the spectrin phenotype is pronounced during the rapid expansion of the newly forming epidermis during development. We plan to do stretching experiments in the future to test if spectrin is required to buffer and protect from cortical mechanical stress.

- TGM1 activity is down in *shSptan1* and *Ecadherin* mutant epidermis. But what about TGM1 protein? Of course, activity would be decreased if the cells weren't differentiating properly, as proposed by the authors.

To examine whether TGM1 levels in relation to its localization are changed we did immunofluorescence analysis for TGM1 on skin sections of *Sptan1*KO and *E-cadherin* KO epidermis. This analysis showed that overall intensity and localization of TGM1 in the granular layer was not changed. These data confirm that *E-cadherin*, spectrin/actomyosin dependent cortex organization regulates cortical TGM activity specifically rather than TGM protein expression or localization.

2. The paper begins by demonstrating that SG2/1 cells maintain shapes when they are isolated. This is very clever experiment! However, there are no quantifications accompanying the images. In addition, it is difficult to say for sure, but the SS cells look like their nuclei get smaller between *in vivo* and isolated, which may say more about the cellular viability than cell shapes.

We have now quantified SG and SS 2D cell size and shape before and after isolation (Fig. 1d,e). Our quantifications show that SG cells maintain their cell shape but show a 19% reduction in size after isolation. In contrast, spinous cells lose their shape and round up becoming fully circular, also resulting in strong decrease in cell size by 86%. The latter is expected as the nuclear shape is coupled to the overall cell shape through the cytoskeleton. Moreover, we cultured these cells for prolonged periods, and neither their cell shapes nor their nuclear shapes were affected, which one would expect if over time viability would go down.

3. In many of the quantifications throughout the manuscript, statistical analyses appear to be run on number of cells quantified rather than number of experimental animals. While I appreciate that there is a dynamic range within animals, using number of cells as the statistical *n* feels inappropriate, and is assigning very high statistical significance to what appear to be very subtle changes.

Although we in principle agree that using many individual datapoints, here most often representing individual cells, as a basis for statistics will generally overestimate the significance, we very

Full Revision

carefully performed the statistical analysis using not only number of cells but also compare the variation in mean/animal to avoid data bias and improper conclusions. Whereas the first can result in underestimating statistical errors, the latter overestimates statistical errors. To account for this potential problem, we have used non-parametric ANOVAs (Kruskal-Wallis followed by Dunn's multiple comparison test) when comparing more than two groups of accumulated cell numbers. Where possible, we used Kolmogorov-Smirnov that is more suitable for high N-numbers but cannot compare multiple groups. We initially planned to include graphs for mean values as well but due to the high number of quantifications this will overload the figure. Some quantifications of individual datapoints that were based on T-test have been redone using Kolmogorov-Smirnov. In case where we found that the difference between accumulated cell numbers is significant and the means per animal clearly show distinct populations even if these differences were statistically not significant for N=3 animals, due to the stringent, non-parametric tests that we used, we still consider the differences relevant.

4. The data put a lot of emphasis on the micro-honeycomb lattice, but their significance is unclear. It does appear that they are spot junctions forming between the two layers of cells, but whether they contribute to cell shape, tension, etc, is very unclear.

We included novel data that better describe how spectrin organizes the cortical actomyosin cytoskeleton both in vivo and in vitro. To this end, we stained for F-actin, α II-spectrin and (non-muscle) myosin-IIa in vivo in 3D epidermal whole mounts, and in vitro in differentiated, multilayered keratinocyte cultures (New Fig. 4a,d). In vivo we find that concomitant with a strong increase in flattening of keratinocytes in the granular vs. the spinous layer, myosin-IIa relocates from lateral intercellular contacts in the spinous layer to a spot-like localization across the cortical spectrin-actin networks in the granular layer, thus indicating an increase in tensile state of this network when cells move from the spinous to the granular layer, in line with our previous work (Rübsam et al Nat. Comm. 2017). Importantly, loss of spectrin increased myosin intensity and size of spots in all suprabasal layers, indicating that loss of spectrin increased the contractile state of F-actin networks. In vitro Myosin-IIa similarly decorates the cortical spectrin-F-actin honeycomb-like lattice of suprabasal cells in a spot-like manner across the network. Upon loss of α II-spectrin, F-actin is organized in stress fiber-like filaments that now show an intense periodic decoration of myosin-IIa, indicating highly contractile F-actin fibers (new Fig. 4d; new Supplementary Fig. 4a), in line with increased recoil upon laser ablation (Fig. 4j).

Taken together, our in vivo and in vitro data show that spectrin is essential for the formation of continuous sub-membranous lateral and apical spectrin F-actin lattices in which myosin-IIa is regularly distributed in a spot-like manner to dissipate tension across the network. This organization that also incorporates the elastic properties of spectrin is likely necessary to accommodate not only shape changes itself, but also absorb forces experienced when cells move into a new layer. Importantly, the cell shape of each layer is characterized by a defined network organization and tensile state that differs from other layers. In agreement, in vivo depletion of α II-spectrin alters the conformation and tensile state of these networks likely explaining observed changes in cell shape.

Full Revision

Understanding the detailed molecular mechanisms by which the micro-honeycomb network organization regulates cellular mechanics and cell shape would require extensive additional analysis beyond the scope of this manuscript. Nevertheless, we now did detailed analysis on our SpectrinKO newborn whole mounts and show streak- and dot-like defects in the F-actin lattice (Fig. 3e). Our extended in vitro analyses show that similar defects occur in Spectrin deficient keratinocytes at low tension (low dose of blebbistatin, 5 μ M) (Fig. 4f) indicating that the more subtle in vivo defects are explained by less overall cortical tension.

Taken together, our data provide evidence that the organization of spectrin-actomyosin into a micro-honeycomb lattice cortical network plays a key role in regulating tension states and cell shape across all layers.

5. The conclusion that ablation results demonstrate viscoelastic behavior (Supplementary Fig. 3i) is not adequately supported. The authors should provide quantitative analysis, such as recoil velocity or relaxation time, to distinguish elastic and viscous properties. Additionally, the imaging duration (60 seconds) is insufficiently justified, and there is no correlation between the expansion direction and cortical behavior. Did the authors try to laser cut a cell junction and look at cortical tension differences?

We agree with the reviewer that further analysis of this data was necessary, see also reviewer 1. While our ablation data closely resemble findings from previous studies, that demonstrated viscoelastic behavior (e.g., Saha et al., 2016; Vuong-Brender et al., 2017), we have conducted additional quantitative analyses over time to further assess visco-elastic properties Fig. 4g-i; Supplementary Fig. 4c-f)

We initially did laser cut individual cell junctions in differentiated cells; however, a small junctional cut did not produce a recoil on the vertex. We have now quantified this (Supplementary Fig. 4c,d). Upon additional ablation of the same junction, but now also ablating the cortical micro-honeycomb lattice adjacent to this junctions resulted in vertex recoil (Supplementary Fig. 4c,d), showing that this junction-coupled network is under tensile stress. We now also provide evidence that the presence of the cortical spectrin-F-actin micro honeycomb lattice vertices maintains tensile function even if a single junction is cut; (1) High-resolution imaging revealed that in keratinocytes perijunctional F-actin cables integrate with the cortical spectrin-actomyosin honeycomb network to form a transcellular spectrin-actomyosin cortical network. (2) Importantly, junctional ablation in keratinocytes with less mature cortices that have not formed a cortical micro honeycomb lattice yet, did result in vertex recoil (Supplementary Fig. 4c,d). Together, these new data indicate the importance of distributing tensile stress along the micro-honeycomb network instead of parallel to junctions as seen in simple epithelia, thus stabilizing cell shape and cell contacts.

In our ablation experiments the tension generated by perijunctional F-actin cables did not appear to exceed the overall tension of the network, as evidenced by elliptical cortical openings following line ablation—even across junctions (see two examples in the second Figure below; yellow arrows indicate perijunctional actin cables). We thus focused on cortical ablation and have therefore quantified the cortical behavior and recoil dynamics upon ablation.

Cortical laser ablation (yellow line) across intercellular junctions (yellow arrows) 30 seconds after ablation.

Minor concerns:

6. The scale bars are mislabeled with "μM" instead of "μm" for micrometers, reflecting a lack of rigor in figure annotations. More broadly, figure labels are insufficiently detailed, detracting from clarity.

We have examined the figures and legends, correcting any errors.

7. There are several typos and sentence fragments throughout the text.

We thank the reviewer for pointing this out and have carefully gone over the text, and identified and corrected errors and did our best to not introduce new ones.

8. Can the authors do the isolated cell shape experiments on spectrin KD epidermal cells? If it is important in cell shapes, while the starting shape is clearly different already, it may also have defects in maintaining that shape. Similar for Ecad KO epidermis.

We appreciate the reviewer's suggestion. However, isolating granular layer cells is technically challenging and obtaining sufficient SpectrinKO mice was not feasible within the timeframe of the revisions. For E-cadherin we have included the isolated cell shape analysis and show that although smaller, the granular SG1 and SG2 flattened tetrakaidecahedron shape is maintained and also is not affected by inhibition actin polymerization (Fig. 2h,i; Supplementary Fig. 2j,k).

9. Quantifications of Western blot and all-spectrin immunofluorescence in knockdown skin is needed.

We have quantified spectrin levels in qPCR, immunofluorescence and western blots (Supplementary Fig. 1e-g).

10. The authors suggest that all-spectrin recruitment to the cell cortex requires myosin II activity. Do they have proposed mechanisms that could be added to the discussion? For example, is increasing contractility sufficient to increase all-spectrin localization? Is this through mechanosensitive molecules? Is spectrin thought to be mechanosensitive?

Full Revision

We have now discussed spectrin mechanosensitivity and developed a better model how actomyosin tension and spectrin promote differentiation. We added the following paragraph to the discussion:

Our previous and current data indicate that layer-specific organization of E-cadherin-directed spectrin-actomyosin lattices also controls the activity and localization of EGFR, with active EGFR being internalized in lower layers whereas in the granular layer active EGFR is retained at the membrane. Either loss of spectrin or reducing actomyosin tension reverses this localization of active EGFR, with membrane localization now being confined to lower layers. The organization of the spectrin-actomyosin cortex can inhibit diffusion of plasma membrane receptors to induce molecular crowding, known as fencing (Fujiwara et al., 2016), as well as organize membrane trafficking events such as endocytosis (Ghisleni et al., 2020). We thus propose a model in which the spectrin-actomyosin organization in the granular layers limits mobility of EGFR and TRPV3, while the high tension state of this network also inhibits internalization of activated EGFR, thus bringing these proteins in close proximity to activate TRPV3 and, subsequently, TGM1 (Fig.7K). In agreement, both ligand binding and actin-dependent membrane organization regulates EGFR distribution at the plasma membrane (Sankaran et al., 2021; Bag et al., 2015). A potential alternative model involves a mechano-sensitive unfolding of spectrin (Leterrier and Pullarkat, 2022; Renn et al., 2019) or other, unknown players within the granular layer cortex that then interact with and cage the EGFR/TRPV3 to the cortex. Thus far, however, there is no evidence for tension-dependent binding partners that may explain EGFR or TRPV3 retention.

We also aimed at answering your question whether increasing contractility is sufficient to increase all-spectrin recruitment using the Rho activator CN03. Unfortunately, despite previous successful application of this drug in keratinocytes in our lab, several different keratinocyte cell lines did not respond to treatment using F-actin, myosin2a and all-spectrin as a readout. We are working on a solution but as we already include extensive revisions on how spectrin regulates cortical actomyosin organization and contractile properties both in vivo and in vitro, we decided to not include these experiments in this manuscript. However, cortical myosin activity is known to locally concentrate spectrin at sites of contraction in fibroblasts (Ghisleni et al. Nat. Comm. 2024).

11. Are there any defects in proliferation/cell cycle in spastin KD cells in culture? Or cell spreading? It would be important to rule these out as possible contributors to decreased TEER - have they efficiently formed a confluent monolayer at a similar timing to controls?

We thank the reviewer for bringing up these points. In vivo, we see increased suprabasal proliferation as now included in Supplementary Fig. 5d,e. We also examined whether this increase was related to a change in cell division orientation in spectrin depleted embryos but could not detect any changes here (Supplementary Fig. 5c). In vitro, our multilayered cultures for the TEER measurement did not show any obvious defect in stratification of the all-spectrin KO or KD cells. We have counted cells at the timepoint of TEER measurement (48h Ca²⁺) (Supplementary Fig. 5f) and did not find a difference in cell numbers, 72h after seeding the same number of Ctr and Spectrin deficient cells. Moreover, during long term culture of SpectrinKO keratinocytes we did

Full Revision

not detect a difference in cell growth when regularly counting these cells for passaging (not shown). We thus conclude that *in vitro* all-spectrin does not obviously affect proliferation and can therefore not explain the changes in TEER function.

Reviewer #3 (Significance (Required)):

This paper combines *in vivo* and *in vitro* approaches to understand the role the spectrin cytoskeleton plays during epidermal development. In particular, it contributes to our understanding of how cells begin to flatten out as they differentiate into suprabasal cells. While the authors hint at the link between cell shape and cell fate/differentiation, it is more appropriate to say that they have linked cell shape to cell/tissue function, particularly barrier function. This manuscript would be interesting and relevant to several audiences - while definitely more basic research, it does have implications for dermatology. More broadly, this study contributes to our understanding of how cell junctions and actin are regulated and organized.

We thank the reviewer for the kind words that acknowledge the general significance of our findings. We would like to emphasize that our data do not only link cell shape to tissue function/barrier function in our case, but also identify a key role for spectrin in regulating early cell fate and differentiation, as shown by the changes in K14 and K10 as well as late cell fate as shown by changes in TGM1 activation important for terminal differentiation and that we link to changes in EGFR/TRPV3 localization and activation.

November 14, 2025

RE: JCB Manuscript #202502071TR

Matthias Rübsam
University Hospital Cologne

Dear Dr. Rübsam:

Thank you for submitting your revised manuscript entitled "Spectrin coordinates cell shape and signaling essential for epidermal differentiation". Your paper has now been assessed again by the original reviewers, all of whom now recommend acceptance. Therefore, we would be happy to publish your paper in JCB pending final revisions necessary to meet our formatting guidelines (see details below).

****As you will see, reviewer #2 has one final concern that we feel should be addressed in the final revision. Please be sure to provide a brief description of how this point was addressed in the final revised manuscript.****

A. MANUSCRIPT ORGANIZATION AND FORMATTING:

1) Text limits: Character count for Articles and Tools is < 40,000, not including spaces. Count includes the abstract, introduction, results, discussion, and acknowledgments. Count does not include title page, materials and methods, figure legends, references, tables, or supplemental legends. Your paper is currently below this limit but please bear it in mind when revising.

2) Figure formatting: Scale bars must be present on all microscopy images, including inset magnifications. Molecular weight or nucleic acid size markers must be included on all gel electrophoresis.

3) Statistical analysis: Error bars on graphic representations of numerical data must be clearly described in the figure legend. The number of independent data points (n) represented in a graph must be indicated in the legend. Statistical methods should be explained in full in the materials and methods. For figures presenting pooled data the statistical measure should be defined in the figure legends. Please also be sure to indicate the statistical tests used in each of your experiments (both in the figure legend itself and in a separate methods section) as well as the parameters of the test (for example, if you ran a t-test, please indicate if it was one- or two-sided, etc.).

****Also, since you used parametric tests in your study (e.g. t-tests, ANOVA, etc.), you should have first determined whether the data was normally distributed before selecting that test. In the stats section of the methods, please indicate how you tested for normality. If you did not test for normality, you must state something to the effect that "Data distribution was assumed to be normal but this was not formally tested."****

4) Materials and methods: Should be comprehensive and not simply reference a previous publication for details on how an experiment was performed. Please provide full descriptions (at least in brief) in the text for readers who may not have access to referenced manuscripts. The text should not refer to methods "...as previously described."

5) Please be sure to provide the sequences for all of your primers/oligos and RNAi constructs in the materials and methods. You must also indicate in the methods the source, species, and catalog numbers (where appropriate) for all of your antibodies.

6) Microscope image acquisition: The following information must be provided about the acquisition and processing of images:

- a. Make and model of microscope
- b. Type, magnification, and numerical aperture of the objective lenses
- c. Temperature
- d. imaging medium
- e. Fluorochromes
- f. Camera make and model
- g. Acquisition software
- h. Any software used for image processing subsequent to data acquisition. Please include details and types of operations involved (e.g., type of deconvolution, 3D reconstitutions, surface or volume rendering, gamma adjustments, etc.).

7) References: There is no limit to the number of references cited in a manuscript. References should be cited parenthetically in the text by author and year of publication. Abbreviate the names of journals according to PubMed.

- 8) Supplemental materials: There are typically strict limits on the allowable amount of supplemental data. Articles may usually have up to 5 supplemental figures. However, in this case, we agree that all 6 supplementary figures are necessary and so we will allow you to exceed this limit.
Please also note that tables, like figures, should be provided as individual, editable files. A summary of all supplemental material (that is, in addition to the supplementary figure legends) should appear at the end of the Materials and methods section. Please see any recent JCB paper for an example of this.
- 9) Conflict of interest statement: JCB requires inclusion of a statement in the acknowledgements regarding competing financial interests. If no competing financial interests exist, please include the following statement: "The authors declare no competing financial interests." If competing interests are declared, please follow your statement of these competing interests with the following statement: "The authors declare no further competing financial interests."
- 10) A separate author contribution section is required following the Acknowledgments in all research manuscripts. All authors should be mentioned and designated by their first and middle initials and full surnames. We encourage use of the CRediT nomenclature (<https://casrai.org/credit/>).
- 11) ORCID IDs: ORCID IDs are unique identifiers allowing researchers to create a record of their various scholarly contributions in a single place. Please note that ORCID IDs are now ***required*** for all authors. At resubmission of your final files, please be sure to provide your ORCID ID and those of all co-authors.
- 12) Thank you for providing the Source Data for your Western blots. However, please be sure to indicate above each Source Data image which figure and panel each full-length image represents (e.g. "Supplementary Figure 1f:").
- 13) Journal of Cell Biology now requires a data availability statement for all research article submissions. These statements will be published in the article directly above the Acknowledgments. The statement should address all data underlying the research presented in the manuscript. Please visit the JCB instructions for authors for guidelines and examples of statements at (<https://rupress.org/jcb/pages/editorial-policies#data-availability-statement>).

B. FINAL FILES:

****It is JCB policy that if requested, original data images must be made available to the editors. Failure to provide original images upon request will result in unavoidable delays in publication. Please ensure that you have access to all original data images prior to final submission.****

****The license to publish form must be signed before your manuscript can be sent to production. A link to the electronic license to publish form will be sent to the corresponding author only. Please take a moment to check your funder requirements before choosing the appropriate license.****

Thank you for your attention to these final processing requirements. Please revise and format the manuscript and upload materials within 7-14 days. If you need an extension for whatever reason, please let us know and we can work with you to determine a suitable revision period.

Thank you for this interesting contribution, we look forward to publishing your paper in Journal of Cell Biology.

Sincerely,

Ian Macara, PhD
Senior Editor
The Journal of Cell Biology

Tim Spencer, PhD
Executive Editor
Journal of Cell Biology

Reviewer #1 (Comments to the Authors (Required)):

The authors have extensively revised their manuscript in response to earlier feedback. They have reasonably addressed all the issues that I raised previously. This is a valuable paper that provides new insight into the molecular regulation of cell shape and its relationship to differentiation.

Reviewer #2 (Comments to the Authors (Required)):

The authors have made substantial improvements and addressed all of my initial concerns. No further experimental revisions necessary in my opinion.

One comment on the text: I would advise the authors to be cautious about claiming that all-spectrin influences differentiation or cell fate per se. The phenotype shows perdurance of K14 signal in basal cells, but it's not necessarily clear that this defect is cell autonomous. Defects in the supra basal layers could have non-autonomous effects, eg on the rate with which basal cells delaminate upwards. Likewise in supra basal cells, it would be more accurate to say that the mutants have defects in certain aspects of barrier formation (eg. TGM1 activity). This doesn't necessarily imply a broad cell fate change or a global inability to undergo terminal differentiation.

Reviewer #3 (Comments to the Authors (Required)):

The authors have satisfactorily addressed my previous concerns. However, the manuscript would still benefit from careful editing to improve the clarity and readability of the text.